# Super-additive cooperation

Charles Efferson[1✉], Helen Bernhard[2], Urs Fischbacher[3,4] & Ernst Fehr[2✉]

Repeated interactions provide an evolutionary explanation for one-shot human cooperation that is counterintuitive but orthodox[1–3]. Intergroup competition[4–7] provides an explanation that is intuitive but heterodox. Here, using models and a behavioural experiment, we show that neither mechanism reliably supports cooperation. Ambiguous reciprocity, a class of strategies that is generally ignored in models of reciprocal altruism, undermines cooperation under repeated interactions. This finding challenges repeated interactions as an evolutionary explanation for cooperation in general, which further challenges the claim that repeated interactions in the past can explain one-shot cooperation in the present. Intergroup competitions also do not reliably support cooperation because groups quickly become extremely similar, which limits scope for group selection. Moreover, even if groups vary, group competitions may generate little group selection for multiple reasons. Cooperative groups, for example, may tend to compete against each other[8]. Whereas repeated interactions and group competitions do not support cooperation by themselves, combining them triggers powerful synergies because group competitions constrain the corrosive effect of ambiguous reciprocity. Evolved strategies often consist of cooperative reciprocity with ingroup partners and uncooperative reciprocity with outgroup partners. Results from a behavioural experiment in Papua New Guinea fit exactly this pattern. They thus suggest neither an evolutionary history of repeated interactions without group competition nor a history of group competition without repeated interactions. Instead, our results suggest social motives that evolved under the joint influence of both mechanisms.

Although repeated interactions may seem like a paradoxical explanation for why humans cooperate in one-shot social dilemmas, the key claim is that people do not really have one-shot interactions. Instead, human psychology has evolved to treat interactions with first-time acquaintances as if they are the beginning of long-term relationships[1]. This hypothesis rests on two additional claims. First, ancestral groups were typically small and cohesive, and most relationships involved interacting repeatedly with group affiliates[6]. When an ancestral human interacted with someone in the same group, the pair were likely to interact again, and reputations were at stake. Second, uncertainty about whether an ancestral pair would interact again[2,3] involved a crucial asymmetry. Behaving badly and damaging one's reputation was an expensive error if the pair did interact again. Behaving well and needlessly protecting one's reputation was a cheap error if the pair did not. Selection favoured risking the cheap error[9].

By this hypothesis, even the most superficial indication that interactions might be repeated leads people to behave as if they are beginning a long-term relationship based on reciprocity and, if all goes well, mutual cooperation. When we observe one-shot cooperation, we are actually observing the evolutionary residue of individual selection under ancestral conditions[1] rather than a clear-eyed response to the explicit incentives at hand. A fundamental trigger for this ancestral psychology is shared group affiliation[10]. When interacting with someone who is ingroup, one should start nice and behave reciprocally because shared group affiliation in the ancestral past was an indication that interactions were likely to repeat[11–13]. When interacting with someone who is outgroup, one should behave selfishly. Selfish behaviour does not require an interest in derogating the outgroup; the ancestral ingroup psychology may simply be inactive[10].

Competition between groups represents a different hypothesis, prominent but controversial, about why humans cooperate in one-shot social dilemmas[4,5,7,14,15]. The principal claim is that ancestral competitions between groups ensured that selection at both the individual and group levels shaped evolution. Selfish people enjoyed an advantage over cooperative people within groups; groups with many cooperative people enjoyed an advantage over groups with many selfish people[16]. If the group selection effect was strong enough, populations would have evolved so that people were cooperative with ingroup members and selfish with outgroup members. The result would have been a parochial psychology that people retain today[17]. The controversy stems from the idea that selection within groups and migration between groups would have quickly made all groups similar in the ancestral past, and so the group selection effect would not have been strong enough[18].

Both hypotheses seem coherent, and they lead to overlapping predictions. If the setting has any features that lead people to see each other as group affiliates, people should behave cooperatively. Otherwise, people should behave selfishly[17]. The features in question can be implicit

[1]Faculty of Business and Economics, University of Lausanne, Lausanne, Switzerland. [2]Department of Economics, University of Zurich, Zurich, Switzerland. [3]Department of Economics, University of Konstanz, Konstanz, Germany. [4]Thurgau Institute of Economics, Kreuzlingen, Switzerland. ✉e-mail: charles.efferson@unil.ch; ernst.fehr@econ.uzh.ch

or explicit, subtle or conspicuous, a matter of conscious awareness or not[12]. Both hypotheses also rest on beliefs about ancestral social groups that will probably remain difficult or impossible to verify. Here we evaluate which of the two hypotheses captures the evolutionary mechanisms responsible for one-shot cooperation.

We do so with a large and comprehensive modelling project, and a closely related behavioural experiment based on a one-shot game with ingroup and outgroup pairings in Papua New Guinea. We show that neither hypothesis works. Neither repeated interactions alone nor intergroup competitions alone support ingroup cooperation in a meaningful way, and neither mechanism leads to ingroup and outgroup predictions consistent with behaviour observed in Papua New Guinea. Repeated interactions generate a cooperative equilibrium, but this equilibrium is exceedingly vulnerable to invasion by a class of mutations that we call 'ambiguous reciprocity'. Gratuitously assuming that such mutations are impossible eliminates the vulnerability, but this approach has no biological justification. Group competitions do not support ingroup cooperation because several mechanisms reduce both the variation between groups and the extent to which group selection can occur given the variation that exists.

Although the discussion of the two hypotheses often seems to treat them as strict alternatives, they are not. Repeated interactions within groups and competitions between groups can coincide[19]. We also show that combining the two mechanisms generates strong positive interactions. Positive interactions occur because intergroup competitions can stabilize ingroup cooperation against ambiguous reciprocity, and intergroup competitions often do this even when they do not support cooperation on their own. When the mechanisms interact, the result is the evolution of cooperative reciprocity with ingroup members, which amplifies cooperation within groups, and uncooperative reciprocity with outgroup members, which erodes cooperation between groups. This mix in which all equilibrium strategies are reciprocal, but not all reciprocal strategies are cooperative, is exactly what we observed among our participants in Papua New Guinea. Thus, an evolved psychology based on repeated interactions in the past may be necessary to explain contemporary one-shot cooperation with ingroup partners, but such a psychology is not sufficient. Intergroup competitions are also necessary but not sufficient. By contrast, the joint influence of the two mechanisms can provide a sufficient explanation for the evolution of one-shot cooperation.

## The two mechanisms in all combinations

Our models examine the evolution of strategies for a sequential social dilemma with a continuous action space[20,21]. The game is a theoretical version of the social dilemma we used in Papua New Guinea, and both our models and experiment rest on the same stage game (Methods). For the stage game, each player has an endowment. The first mover can transfer any amount from her endowment to the second mover, and the transfer is doubled. Conditional on the first mover's transfer, the second mover can transfer any amount from her endowment to the first mover, and this transfer is also doubled. A one-shot interaction consists of a single stage game. Repeated interactions consist of multiple stage games with new endowments for each interaction. An individual's strategy consists of an initial transfer and a response function (Methods). The initial transfer specifies what to transfer, if first mover, in the first interaction. For all subsequent transfers, the response function specifies what to transfer as a function of the partner's most recent transfer.

We model three scenarios in subdivided populations (Methods). The 'repeated interactions' scenario does not have competitions between groups. Individuals only play the social dilemma with ingroup partners, and these games can be one-shot or repeated. In the 'group competition' scenario, individuals play the social dilemma with ingroup partners and

outgroup partners. All games are one-shot, and competitions between groups occur. The 'joint' scenario is similar to the group competition scenario, but all ingroup games are repeated.

The strategies that evolve in the model provide predictions for our experiment. For the repeated interactions scenario, the strategies that evolve under repeated interactions provide predictions for the ingroup pairs in our experiment, whereas the strategies that evolve under one-shot play provide predictions for the outgroup pairs. This captures the hypothesis that ingroup interactions activate an ancestral psychology based on repeated play, whereas outgroup interactions—assumed to be rare and typically one-shot in the ancestral past—leave this psychology dormant[10–13]. For the group competition scenario, individuals play the social dilemma with both ingroup partners and outgroup partners, and strategies for doing so are explicitly conditional on group affiliation. Our experiment involved both ingroup and outgroup pairings, and so we derive ingroup and outgroup predictions directly from the ingroup and outgroup strategies that evolve in the model. For the joint scenario, individuals also play the social dilemma with ingroup and outgroup partners, and predictions for the experiment follow directly. However, because the joint scenario combines repeated interactions within groups and competition between groups into a novel selective regime, it can potentially support the evolution of strategies that differ from those that evolve when the two mechanisms operate in isolation.

## Framework for a wide range of conditions

Many details about human social life in the evolutionary past are unknown and are likely to remain that way[6]. We know little about how often people moved between ancestral groups, who exactly moved when someone did move, and how these characteristics varied across ancestral populations. Extrapolating from contemporary foragers is often the best we can do[22,23]. These limitations emphasize the importance of examining a wide range of conditions to identify settings that robustly support cooperation without being acutely sensitive to the details. Accordingly, we systematically manipulate the following six model characteristics (Methods). Together with our three scenarios, the result is a framework that handles uncertainty about ancestral conditions by considering an unusually comprehensive set of possibilities.

### The dimensionality of strategy space

The number of dimensions used to specify a strategy controls which strategies can and cannot arise via mutation. We vary the number of dimensions from two to four (Fig. 1). Two dimensions allow strategies that always escalate the degree of cooperation, strategies that always de-escalate, and perfect reciprocity. Perfect reciprocity starts generous, if first mover, and otherwise perfectly mimics the partner's most recent move. Three dimensions introduce the possibility of ambiguous reciprocity. Ambiguous strategies escalate cooperation when the partner is relatively uncooperative and de-escalate when the partner is relatively cooperative. Four dimensions add further possibilities, including de-escalation when the partner is uncooperative and escalation when the partner is cooperative.

### Cancellation effects at the individual level

Relatedness within groups can increase the probability that cooperators interact with other cooperators, which can support the evolution of ingroup cooperation. However, if these same cooperators compete against each other to reproduce at the individual selection stage, these competitions offset the effects of ingroup cooperation[24,25]. Offsetting effects of this sort are called cancellation effects, and we call them cancellation effects at the individual level to distinguish them from the group-level cancellation effects[8] discussed below. We modulate cancellation effects at the individual level by varying the life cycle.

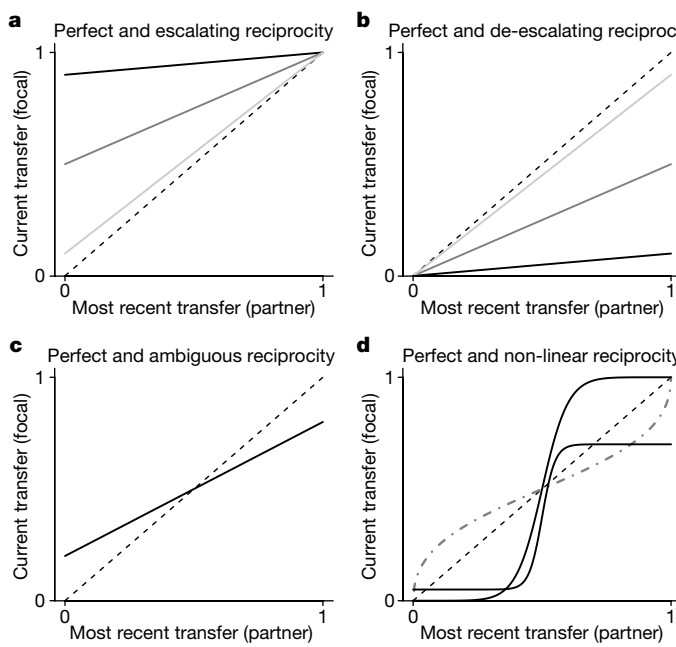

**Fig. 1 | Example response functions. a**, Escalating reciprocity means that the focal individual has a positively sloped response function and increases the transfer level when possible. Examples include weak (light grey), intermediate (grey) and strong (black), where weak escalation increases cooperation slowly and strong escalation does so quickly. **b**, De-escalating reciprocity means that the focal individual has a positively sloped response function and decreases the transfer level when possible. Examples include weak (light grey), intermediate (grey) and strong (black), where weak de-escalation reduces cooperation slowly and strong de-escalation does so quickly. **c**, Ambiguous reciprocity amounts to escalation in response to low transfers and de-escalation in response to high transfers. Ambiguous strategies, if allowed, arise readily from mutations of de-escalating and escalating strategies (supporting information section 1.2.12[26]). **d**, Non-linear forms of reciprocity allow non-linear analogues of ambiguous reciprocity (grey dash–dot line), as well as complex and flexible mixtures of de-escalation and escalation in response to both low and high transfers (black solid lines). Two dimensions are adequate for perfect, escalating and de-escalating forms of reciprocity, with one dimension for the initial transfer and another for the response function. Ambiguous reciprocity requires three dimensions, and non-linear strategies require four dimensions. **a–d**, The dashed line shows the response function for perfect reciprocity.

The decoupled life cycle places migration after game play but before ingroup competition to reproduce. This reduces cancellation effects at the individual level relative to the coupled life cycle, which places migration before game play.

### Cancellation effects at the group level
Cancellation effects can also operate at the group level[8]. If cooperative groups that win intergroup competitions go on to compete against their own, highly cooperative descendant groups, cancellation effects at the group level are high. Otherwise, these effects are low. We vary cancellation effects at the group level by manipulating the number of groups ($\Xi$) moving around in the meta-population each generation. At one extreme, groups move around a lot ($\Xi = 40$), and cancellation effects at the group level are as weak as possible. At the other extreme, groups do not move at all ($\Xi = 0$), and group-level cancellation effects are as strong as possible.

### The importance of differences in aggregate resources between groups
Conditional on a group competition occurring, the group with the most resources wins the competition with a probability more or less

sensitive to the difference in resources between the two groups. We capture greater sensitivity with larger values of the parameter $\lambda$.

### Migration rates
Either 8 or 16 out of 24 individuals migrate per group per generation ($m_j$). These values lead to relatively high or low relatedness within groups, respectively.

### Initial conditions
We use three different initial conditions. Focusing on two of the three conditions, seeding the population with perfect reciprocators means initial cooperation is as high as possible, whereas seeding the population with unconditionally selfish individuals means initial cooperation is as low as possible.

## The limits of repeated interactions
As a stand-alone mechanism, repeated interactions have an inexorable weakness. Cooperative strategies only evolve and persist if we arbitrarily restrict the set of possible strategies (Fig. 2). When using two dimensions to specify strategies, reciprocal strategies that support cooperation invade and persist under a wide range of conditions. With three or four dimensions, such strategies often invade, but they never persist. This result does not depend on whether the migration rate is low (Fig. 2a,b) or high (Fig. 2c,d), nor on a specific life cycle with associated cancellation effects at the individual level (Fig. 2a,c versus 2b,d). Increasing the number of interactions from 100 to 1,000 has little to no effect (supporting figures 15 and 16; supporting figures are available at Zenodo[26] (https://doi.org/10.5281/zenodo.10355347)). Quadrupling instead of doubling transfers also leaves cooperation at very low levels (supporting figure 16[26]). Finally, increasing relatedness by reducing migration rates to nearly zero, which is unrealistic for human populations[6,22], only supports small increases in cooperation (supporting figures 15 and 16[26]). Of note, the weakness of repeated interactions holds even though we limit attention to dyads, which are maximally conducive to the evolution of reciprocity[6,18].

The key distinction is between a two-dimensional strategy space (Fig. 1a,b) that precludes ambiguous reciprocity and a three-dimensional space that does not (Fig. 1c). A fourth dimension (Fig. 1d) has few additional consequences. The distinction between two and three dimensions is critical for the following reason. Regardless of dimensionality, repeated interactions often lead populations to evolve at first so that most individuals exhibit high initial transfers and escalating reciprocity. Once these strategies are common, variation among individuals in the degree of escalation (Fig. 1a) is not especially important. When initial transfers are high, players start cooperating at high levels, and they have little room to escalate. One degree of escalation is about as good as any other in terms of the behaviours generated. Selection on the exact degree of escalation is weak and drift correspondingly important.

However, in terms of susceptibility to invasion, one degree of escalation is not about as good as any other. Some forms of escalation are susceptible to invasion by ambiguous strategies, whereas others are not. Specifically, the equilibrium degree of escalation is often resistant to invasion by ambiguous strategies, but it is also extremely similar to other degrees of escalation that are susceptible to invasion (supporting information section 1.2.12; supporting information is available at Zenodo[26] (https://doi.org/10.5281/zenodo.10355347)). Consequently, after high initial transfers and escalation become common, only a tiny amount of drift makes the population vulnerable to ambiguous strategies. This vulnerability is irrelevant if we exclude ambiguous reciprocity by fiat, as we do when the strategy space is two-dimensional. Otherwise, this vulnerability dominates evolutionary dynamics (supporting information section 1.2.12[26]).

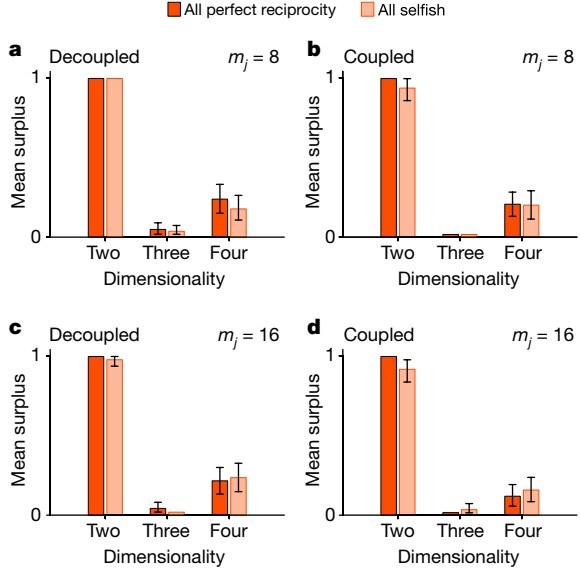

**Fig. 2 | Strategic flexibility hinders cooperative reciprocity under repeated interactions.** The graphs show the mean surplus per individual per ingroup interaction in the final generations of simulated evolution under $n = 100$ ingroup interactions. Error bars show 95% confidence intervals, which are calculated by bootstrapping over 50 independently simulated populations and omitted when extremely narrow. Strategies are defined in either two, three or four dimensions (Fig. 1). Initial conditions are either relatively favourable for the evolution of cooperation (all perfect reciprocity) or relatively unfavourable (all selfish). **a**–**d**, The life cycle either decouples game play and individual selection (**a**,**c**), which reduces individual-level cancellation effects and enhances the potential for relatedness to support cooperation, or it couples them (**b**,**d**). The number of migrants per group per generation ($m_j$) is either relatively low (**a**,**b**) or high (**c**,**d**). Cooperative reciprocal strategies evolve or persist only when strategies are two-dimensional, and thus ambiguous reciprocity (Fig. 1c) is not possible. When strategies are defined in three or four dimensions, with a relatively high degree of strategic flexibility as a result, cooperative reciprocal strategies invade, but they never persist.

Notably, when ambiguous strategies invade, they do not persist; de-escalating strategies displace them. We find no evidence for an equilibrium in which ambiguous strategies predominate, and so we should not expect to observe much ambiguous reciprocity empirically. This is no reason, however, to exclude the strategies theoretically. To do so would assume that ambiguous strategies are impossible for a human genome to encode or a human mind to imagine. Instead, ambiguous strategies represent a minimum degree of strategic flexibility. Mutations to the left intercept of de-escalating response functions necessarily generate ambiguous strategies, as do mutations to the right intercept of escalating response functions.

If we allow ambiguous strategies, dynamics take the following stylized form. Escalating strategies invade and then drift. Ambiguous strategies then invade, only to be displaced by de-escalating strategies (supporting figures 9–14[26]). Given a minimal degree of strategic flexibility, repeated interactions have no meaningful effect and do not provide a robust explanation for the evolution of cooperation. Both one-shot interactions and repeated interactions lead to the evolution of low initial transfers and de-escalating reciprocity (supporting information sections 2.1.10–2.1.12[26]). When we take the strategies that evolve under repeated interactions as a prediction for how people should play one-shot games with ingroup partners, we predict low initial transfers with de-escalation, an uncooperative form of reciprocity. When we take the strategies that evolve under one-shot interactions as a prediction for one-shot games with outgroup partners, we predict the same form of uncooperative reciprocity.

## The limits of group competition

As a stand-alone mechanism, intergroup competition also entails a fundamental weakness. The dimensionality of strategy space does not matter (supporting information section 3[26]). However, the life cycle, the importance of differences in resources between groups ($\lambda$), and migration ($m_j$) are all jointly critical. Intergroup competition supports the evolution of ingroup cooperation only if (1) the life cycle couples game play and individual selection (coupled), (2) $\lambda$ takes the highest value that we consider ($\lambda = 100$), and (3) the migration rate is relatively low ($m_j = 8$). Otherwise, ingroup strategies evolve to generate little or no cooperation (see GC(1) in Fig. 3 and Extended Data Fig. 1).

In effect, a delicate mix of at least three characteristics must hold. Migration must be sufficiently low to generate meaningful differences between groups. Intergroup competitions must have outcomes that are sufficiently sensitive to the group-level differences in resources that exist. Finally, the timing of events in the lives of individuals must take the correct form. Under repeated interactions, the decoupled life cycle is favourable for cooperation because it reduces cancellation effects at the individual level (supporting figures 15 and 16[26]). The group competition scenario is exactly the opposite. The decoupled life cycle is unfavourable for cooperation because migration occurs after game play but before intergroup competition, which separates a group's productivity from the group's ability to win intergroup competitions. During game play, a group with many cooperative individuals enjoys large gains because many group members cooperate. Immediately after game play, however, group members migrate, and they carry the gains from cooperation with them. This idea is relevant, for example, when the beneficiaries of cooperation accumulate embodied capital in the form of knowledge, skills, health and physical strength[27]. More broadly, when individuals carry the benefits of cooperation with them, any movement of individuals after game play but before group competition redistributes resources across groups in a way that must attenuate, all else equal, the bite of intergroup competition as a mechanism. The effect is weak if the migration rate is low and strong if high.

In sum, with a delicate three-part mix in place, intergroup cooperation supports the evolution of high initial transfers with escalation for ingroup play and low initial transfers with de-escalation for outgroup play. If we disturb this delicate mix, both ingroup and outgroup strategies evolve to exhibit low initial transfers with de-escalation (supporting information sections 2.1.13 and 2.1.14[26]). Thus, intergroup competition does not provide a robust explanation for the evolution of cooperation, and we predict uncooperative reciprocity for one-shot play with both ingroup and outgroup partners.

## Super-additive cooperation

In the joint scenario (supporting information section 2.1.6[26]), cooperative ingroup strategies often invade and persist under circumstances in which neither repeated interactions nor group competitions support cooperation on their own. Equivalently, the two evolutionary forces often interact strongly when combined. The result is a form of super-additive cooperation that far exceeds what we obtain by summing the cooperation levels from the two constituent mechanisms (Fig. 3 and Extended Data Fig. 1). The joint scenario, for example, can support the evolution of extreme super-additive cooperation even when initial conditions are unfavourable, strategy spaces are high-dimensional, individual-level and group-level cancellation effects are as strong as possible, and the migration rate is high (see Extended Data Fig. 1d and supporting figures 154d, 169d, 184d and 187d[26]). The joint scenario does not always lead to high cooperation, but it does so in a robust way that is not hypersensitive to the details. Unlike the repeated interactions scenario, the joint scenario does not require arbitrary restrictions on the strategy space for ingroup cooperation to evolve (supporting information section 3[26]). Unlike the group competition

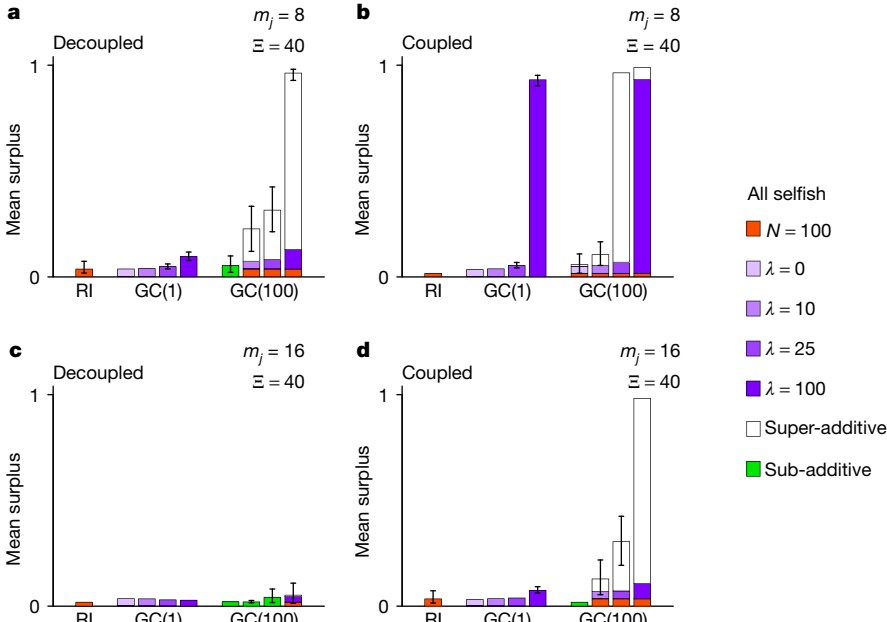

**Fig. 3 | Super-additive cooperation when initial conditions are relatively unfavourable and group mixing relatively favourable for the evolution of cooperation.** Graphs show the mean surplus per individual per ingroup interaction from the final generations of evolutionary simulations. Error bars show 95% confidence intervals, which are calculated by bootstrapping over 50 independently simulated populations and omitted when extremely narrow. RI signifies the repeated interactions scenario with $n = 100$ interactions per ingroup pair. GC(1) indicates the group competition scenario as competition outcomes vary in sensitivity to group differences ($\lambda$). GC(100) combines the two component mechanisms into the joint scenario, with bars from left to right corresponding to increasing $\lambda$. When the joint scenario is super-additive, the mean surplus is decomposed (supporting information section 3[26]) into the repeated interactions effect (orange), the group competition effect (purple), and the super-additive effect (white). Initial conditions consist of a population of unconditionally selfish individuals (all selfish), and group mixing is as high as possible ($\Xi = 40$), with the former unfavourable and the latter favourable for ingroup cooperation. That said, results are nearly identical when initial conditions consist of a population of perfect reciprocators (supporting figure 161[26]), which reveals that initial conditions actually have no meaningful effect. **a**–**d**, The life cycle either decouples (**a**,**c**) or couples (**b**,**d**) game play and individual selection. The number of migrants per group per generation ($m_j$) is either relatively low (**a**,**b**) or high (**c**,**d**). Initially, the population consists entirely of selfish individuals, and cooperative strategies must actually invade to become established. Nonetheless, repeated interactions and group competition interact strongly to produce large super-additive gains (**a**,**b**,**d**).

scenario, low migration, group competitions with outcomes sensitive to between-group differences ($\lambda$), and the coupled life cycle do not constitute a delicate mix in which all the pieces must be in place.

Super-additivity and high levels of ingroup cooperation evolve under a wide range of conditions that involve, among other sources of variation, (1) three- and four-dimensional strategy spaces (supporting figures 149–166[26]), (2) high migration rates (for example, Fig. 3d and Extended Data Fig. 1d; supporting information section 3[26]), (3) $\lambda$ values well below the maximum (for example, Fig. 3b; supporting information section 3[26]), and (4) the decoupled life cycle (for example, Fig. 3a; supporting information section 3[26]). Finally, cancellation effects at the group level can be quite detrimental to the evolution of ingroup cooperation. The joint scenario, however, leads to the evolution of ingroup cooperation far more robustly than group competition alone (supporting information section 2.1.18[26]), even when group-level cancellation effects reach their maximum value (for example, Extended Data Fig. 1d; supporting information section 3[26]). Finally, extensions of the model incorporate weak selection and individuals who make mistakes by deviating from the transfers their strategies specify. Results from both extensions are extremely similar to the results presented here (supporting information sections 4 and 5[26]).

Why do repeated interactions and group competition interact? As explained, under repeated interactions alone, a finite population drifts to regions of strategy space that are susceptible to invasion by ambiguous forms of reciprocity, with the collapse of ingroup cooperation the inevitable result. Before this collapse, most individuals have cooperative strategies with high initial transfers and escalating reciprocity (supporting information sections 1.2.12 and 2.1.12[26]). In this sense, repeated interactions support a kind of cooperative attractor that exists but is exceedingly fragile in finite populations. The joint scenario augments repeated interactions with intergroup competition, but intergroup competition does not create a cooperative equilibrium out of thin air. Rather, it helps stabilize a finite population of cooperative escalating reciprocators against the corrosive effects of drift and ambiguous reciprocity.

Intergroup competition functions as a kind of equilibrium selection device. Several mechanisms can transform a social dilemma into some other game with multiple equilibria. An aversion to inequality[28] is a proximate psychological mechanism that can render mutual cooperation an equilibrium in a one-shot prisoner's dilemma[29]. A psychology prone to internalize social norms and motivate people to punish norm violations can do the same[30,31]. Repeated interactions famously support many equilibria[32], and in our setting evolutionary dynamics under repeated interactions support two general classes of equilibria. One class is based on escalating reciprocity, and the other is based on de-escalating reciprocity. The cooperative escalating equilibria generate high payoffs. In the absence of group competitions, however, the fragility of these equilibria dominates, and the most probable outcome in the long run is an uncooperative de-escalating equilibrium.

Group competition can change the balance of forces by adding a mechanism that favours relatively cooperative groups. The higher payoffs associated with escalation can now dominate the fragility of escalation, with the final outcome a cooperative escalating equilibrium. When group competition shifts the balance in this way, the cooperative outcome does not require large differences between groups.

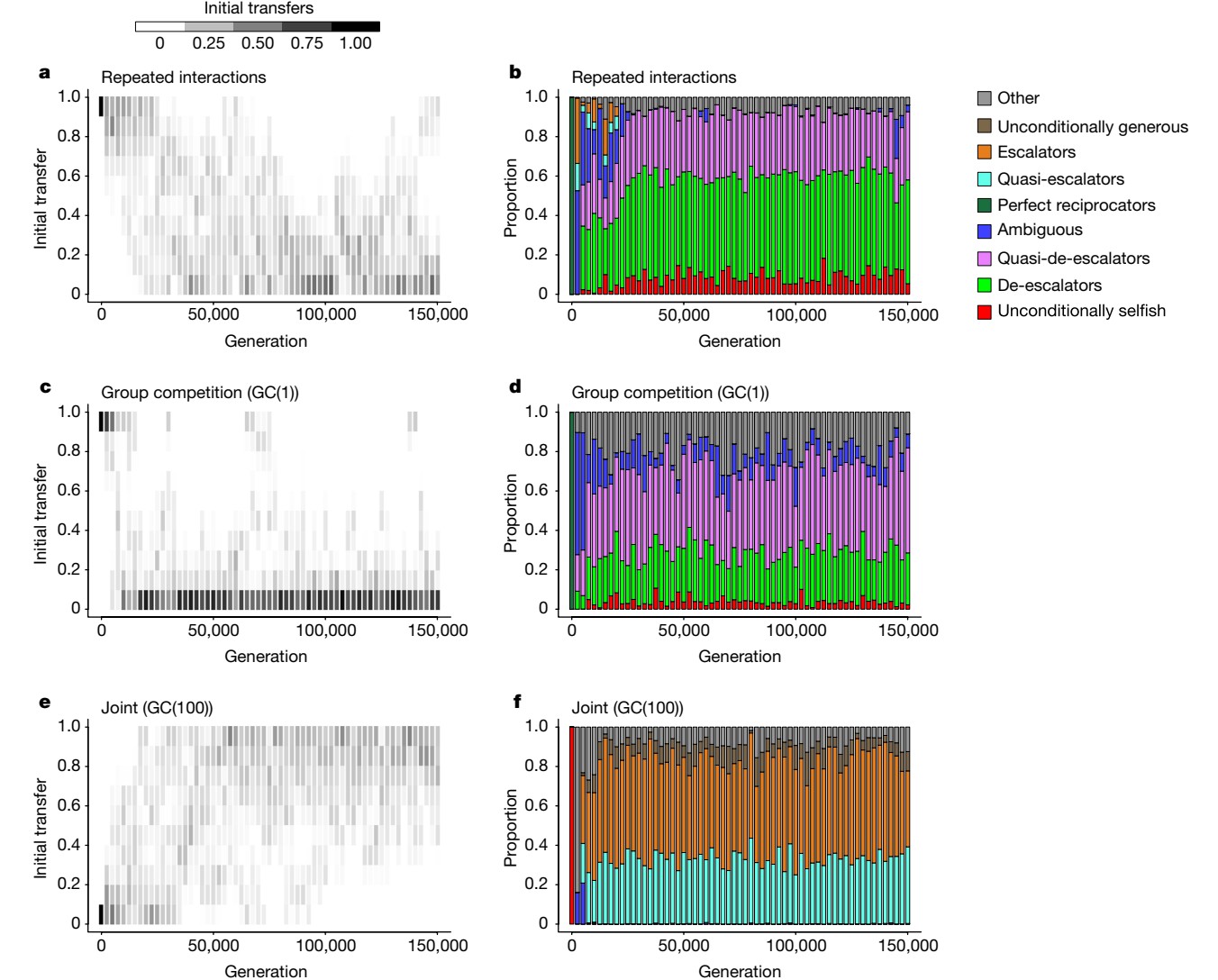

**Fig. 4 | Evolution of strategies under three scenarios. a–f**, Relative frequencies of initial transfers (**a**,**c**,**e**) and response functions (**b**,**d**,**f**). We bin initial transfer values and indicate relative frequencies by bin with a greyscale. We categorize response functions into discrete types. Because ambiguous strategies cover such a broad range, we separately identify extreme forms of ambiguous reciprocity (supporting information section 2.1.17[26]). Specifically, quasi-de-escalators are similar to de-escalating strategies in that they generate extremely low levels of cooperation in the long run. Similarly, quasi-escalators are similar to escalating strategies in that they generate extremely high levels of cooperation. For the results here, migration rates are high, group competitions have outcomes that are highly sensitive to differences between groups, and cancellation effects at the individual and group levels are as strong as possible (supporting information section 2.1.17[26]). Initial conditions favour cooperation in the repeated interactions (**a**,**b**) and group competition scenarios (**c**,**d**), but they disfavour cooperation in the joint scenario (**e**,**f**). Nonetheless, uncooperative forms of reciprocity prevail in the former two scenarios, whereas cooperative forms of reciprocity prevail in the latter scenario. For each scenario, results are based on five independently simulated populations.

The differences between groups in our simulated populations are limited[33], with the variation in strategies between groups constituting around 4–7% of the total variation in the population (supporting information section 2.1.19[26]). Any mechanism that increases variation among groups, with special forms of cultural transmission an obvious example[18], would presumably increase the scope for super-additive gains.

This summary does not mean that repeated interactions are the core mechanism, with group competitions having an ancillary role. Intergroup competition is essential. Repeated interactions without group competitions lead to the evolution of uncooperative reciprocity (Fig. 4a,b), just like group competitions without repeated interactions (Fig. 4c,d). When the two mechanisms are combined, however, outcomes can take an entirely different form (supporting information section 2.1.17[26]). When outcomes are super-additive, ingroup strategies

evolve to take a cooperative reciprocal form (Fig. 4e,f), namely high initial transfers and escalation. Evolved outgroup strategies take an uncooperative reciprocal form characterized by low initial transfers and de-escalation. Notably, de-escalation is at least partially owing to the fact that selection on the right intercepts of outgroup response functions should be weak in the joint scenario. Outgroup choices contribute little to fitness under the joint scenario, and so selection on outgroup strategies is weak in general. In particular, when initial transfers and left intercepts evolve to be low, as they do, selection on the right intercepts of response functions should be especially weak. Right intercepts then drift, which leads to intermediate values and de-escalating reciprocity. These outgroup strategies are more cooperative than the unconditional defection that represents the upper limit of feasible generosity towards outgroup interactions in many models[34], but they are less cooperative than the reciprocal escalation

that evolves to manage ingroup interactions in the joint scenario. The joint scenario is the only scenario that generates this ingroup–outgroup pattern with any regularity.

## Reciprocity among Ngenika and Perepka people

We implemented a two-person sequential social dilemma among people from Ngenika and Perepka groups in the Western Highlands of Papua New Guinea (Methods). As treatments, we manipulated the group affiliations of the two players to create ingroup and outgroup pairings. As explained, when super-additive cooperation occurs under the joint scenario, it rests on an evolved strategy profile consisting of high initial transfers with escalating reciprocity for ingroup partners and low initial transfers with de-escalating reciprocity for outgroup partners (Fig. 4 and supporting information section 2.1.17[26]). Perepekas and Ngenikas exhibited exactly this pattern. First movers (Extended Data Fig. 2a) exhibited high initial transfers with ingroup partners and low initial transfers with outgroup partners (ordinal logistic regression with ingroup dummy, d.f. = 67, odds ratio 6.273, $t$ = 3.973, two-sided $P = 1.76 \times 10^{-4}$). Among second movers, response functions were positively sloped (ordinal logistic regression with standard errors clustered on subject, d.f. = 408, odds ratio 2.090, $t$ = 5.698, two-sided $P = 2.33 \times 10^{-8}$, see supporting information section 9[26]) and uniformly more cooperative with ingroup partners than with outgroup partners (ordinal logistic regression with standard errors clustered on subject, d.f. = 408, odds ratio 4.744, $t$ = 3.518, two-sided $P = 4.83 \times 10^{-4}$, see supporting information section 9[26]). Of particular importance, second movers (Extended Data Fig. 2b) exhibited escalating reciprocity with ingroup partners and de-escalating reciprocity with outgroup partners. Only the joint scenario reliably predicts this strategy profile.

## Discussion

Repeated interactions alone cannot explain the evolution of one-shot cooperation because they cannot explain the evolution of repeated cooperation. Without unjustifiable restrictions on the strategy space, repeated interactions always lead to uncooperative forms of reciprocity. A few analogous results hold for the prisoner's dilemma in which players simply choose defect or cooperate[18,35]. The studies in question expand the strategy space by allowing strategies that condition the current choice on an increasing number of past interactions. Increasing strategic flexibility in this way undermines sustained cooperation[2,3,36], with even the most favourable conditions rarely generating cooperation rates in excess of 0.5 under repeated interactions alone[36].

One might object that expanding the strategy space in this way requires decision makers to have unreasonably long and detailed memories. This objection does not hold for our models because we only consider strategies that require people to remember a single interaction. Nonetheless, when cooperation varies continuously, strategic flexibility arises in other ways. A simple three-dimensional strategy that conditions only on the partner's most recent move is already enough to destabilize cooperative strategies, with little to no cooperation the final outcome. As a caveat, the results we present here are based on models without mistakes; individuals transfer exactly the amounts their strategies specify. Adding mistakes to our model (Methods) reinforces our findings. With mistakes, repeated interactions remain unable to support the evolution of cooperation, and mistakes actually expand the range of conditions under which super-additive cooperation appears (supporting information section 5[26]).

Intergroup competition also does not reliably support the evolution of one-shot cooperation, and the limitations of group competition could be even more serious than imagined. Group selection does not necessarily occur just because groups compete; cooperative groups must also win[18,37]. We have considered three reasons this may or may not happen. First, the timing of life events can affect the link between a group's productivity and its ability to win intergroup competitions. Under our coupled life cycle, a group with many cooperators produces large gains that remain in the group to help win competitions against other groups. Under our decoupled life cycle, migration exports the gains from cooperation before such competitions occur, which attenuates the link between productivity and winning. Migration does not hinder cooperation simply by making groups similar; migration makes groups similar at the worst possible time. Second, cancellation effects at the group level[8] undermine cooperation that would otherwise evolve. If a highly productive cooperative group ends up competing against its descendant groups, it enjoys little relative advantage because it competes against other highly productive cooperative groups. Third, even if a productive group competes with an unproductive group, the outcome is still uncertain because of any remaining forces ($\lambda$) that shape how differences between groups translate into probabilities of winning.

Many pieces have to come together for group competition to support cooperation as a stand-alone mechanism. As we know, groups must be different from each other for the selection of groups to be meaningful[18,37]. Equally critical, the gains from cooperation must stay in the group until group competitions occur, cooperative groups must compete specifically with uncooperative groups, and the outcomes of competitions must be sensitive to the differences between cooperative and uncooperative groups. The existence of so many conditions highlights a fundamental point. Estimating the frequency and lethality of ancestral wars[38–40], to take a contentious example, is not by itself decisive when evaluating the role of group selection. We would also need to know which specific groups fought against each other and who exactly died when wars occurred.

Because neither repeated interactions nor intergroup competitions support the evolution of cooperation by themselves, repeated interactions merit just as much infamy as group selection. Repeated interactions may seem more palatable because the effects operate via individual selection[1,12], but this is irrelevant if cooperation is not the result. The claim that we explain one-shot cooperation with an ancestral psychology based on repeated interactions is not theoretically viable. Empirically, people may be reciprocators who care about their reputations, and this may even be true in anonymous one-shot settings, but repeated interactions do not explain the evolution of such a psychology. Across taxa, in fact, and consistent with the theoretical weaknesses of repeated interactions, empirical examples of conventional reciprocal altruism in animal societies are surprisingly rare[41].

In spite of the weaknesses of the two mechanisms when separate, each can offset the weaknesses of the other. Repeated interactions create a cooperative attractor that is chronically fragile, and intergroup competitions control this fragility in situations where they do not support a cooperative attractor of their own. This super-additive mix produces the generic pattern observed in empirical studies, namely more cooperation with ingroup partners and less cooperation with outgroup partners[7]. More precisely, our joint scenario predicts the specific nuanced pattern we observed in Papua New Guinea, a pattern consisting of escalating reciprocity with ingroup partners and de-escalating reciprocity with outgroup partners. Future empirical research could examine how widely this mix of ingroup–outgroup reciprocity holds across societies, but we already know that ingroup favouritism in social dilemmas is widespread[10,42].

Our findings further suggest an important point about the evolution of cooperation. The mechanisms hypothesized to support the evolution of cooperation are rarely, if ever, mutually exclusive[19,36,43,44]. Current disputes about the evolution of human cooperation centre largely around whether some special or even unique mechanism has shaped human social evolution, with our extreme reliance on culture as a leading candidate[5,6]. However these disputes will be resolved, our results highlight the possibility that the combination of mechanisms responsible for human cooperation can also be special or even unique.

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

## Methods

### A sequential social dilemma with a continuous action space

In the models, pairs of players play a social dilemma. Players choose how much to cooperate from a continuous action space, and moves are sequential. The game is thus an apt description of many social dilemmas past and present. Food sharing[45,46] and alloparental care[47], for example, must be sequential social dilemmas with continuous action spaces. They are not simple prisoner's dilemma games in which players simultaneously decide to defect fully or cooperate fully. The emphasis on continuous action spaces is not trivial. As results from the repeated interactions scenario show, intuitions honed on the analysis of reciprocal altruism in repeated prisoner's dilemma games[48] do not extend to settings where cooperation can vary continuously. Moreover, by developing models and experiment (see below) with parallel designs, we recruit the complementary strengths of both methods in a way that renders the link between theory and empiricism transparent[49]. We do not need a vague intermediate step where we extrapolate from models based on one type of social interaction to experiments involving another type of social interaction, with misleading predictions as a result[44].

With respect to the stage game in the models (supporting information section 1[26]), each player has an endowment normalized to one. The first mover can transfer any amount up to and including her full endowment to the second mover, and the transfer is doubled. Then, the second mover can transfer any amount up to and including her full endowment to the first mover, and this transfer is also doubled. Because transfers are doubled, expected relatedness cannot explain cooperation. Given what we know about average relatedness within groups in small-scale societies[22], efficiency gains would have to be much higher than this for relatedness alone to be adequate.

A one-shot interaction is one stage game. Repeated interactions consist of repeated stage games, where each repetition involves new endowments. An individual's strategy has two parts, an initial transfer and a response function. The initial transfer specifies how much the individual transfers, if first mover, for the first interaction only. For all choices after the initial transfer, the response function specifies an individual's current transfer as a function of her partner's most recent transfer. Specifically, the second mover always responds to the first mover's transfer in the same interaction. If interactions are repeated, from the second interaction onward, the first mover responds to the second mover's transfer from the preceding interaction (supporting information sections 2.1.3, 2.2 and 2.3[26]).

### The three scenarios

The repeated interactions scenario consists of models of populations subdivided into 40 groups of 24 individuals each without any competition between groups. Individuals within groups pair off randomly to play the game. Individuals only play the social dilemma with ingroup partners, and we consider both one-shot games and repeated interactions (supporting information section 2.1.4[26]). Because individuals only play with ingroup partners, the repeated interactions scenario isolates the effects of repeated interactions and the reputational concerns they create from the effects of intergroup competition and more generally outgroup interactions of all sorts. We ignore uncertainty about whether a game is one-shot or repeated[2,3], which maximizes the scope for repeated interactions to support cooperation when relationships actually do last a long time.

The group competition scenario also consists of models in subdivided populations. In this scenario, however, groups compete, and games are always one-shot. Groups are paired within a generation (supporting information section 2.1.5[26]). Each individual plays both a one-shot social dilemma with a randomly selected ingroup partner and a one-shot social dilemma with a randomly selected outgroup partner from the paired group. The individual has separate strategies for ingroup versus outgroup interactions. The opportunity to cooperate with outgroup partners in our models is different from most evolutionary models of parochialism because most models limit attention to outgroup strategies that range from defection to outright aggression[34]. Defection in these models is the most generous feasible option for an outgroup interaction.

After game play, we model the occurrence of group competitions by assuming that paired groups compete against each other with relatively low probabilities (supporting information section 2.1.7[26]) that decrease as the groups become more similar (supporting information section 2.1.5[26]). This approach reflects the idea that paired groups assess each other and avoid competing when they have trouble identifying the probable winner, which is consistent with both past modelling work and ethnographic evidence[4,50]. We can think of a competition as a violent conflict, a competition for some limited resource, or a process where the culture of one group displaces the culture of another group[31]. In general, the group competition scenario isolates the effects of intergroup competition from the effects of repeated interactions and associated reputational concerns within groups. The joint scenario combines both repeated interactions within groups and competition between groups (supporting information section 2.1.6[26]). It is identical to the group competition scenario with one exception; ingroup interactions are always repeated.

### A framework for comprehensive variation in model structure

To develop a set of models that examine a wide range of potential ancestral conditions, we cross the six model characteristics below in all possible combinations.

**The dimensionality of strategy space (all scenarios).** We vary the dimensionality of the strategy space as a way of manipulating the set of possible strategies. When a strategy is two-dimensional, it consists of an initial transfer and a second quantity controlling the slope and location of a linear response function (supporting information sections 1.1 and 2.2[26]). Possible response functions include perfect reciprocity, escalating reciprocity, and de-escalating reciprocity. A perfectly reciprocal response function means a focal individual's transfer is exactly the same as her partner's most recent transfer (Fig. 1). When two perfect reciprocators interact, all transfers are identical to the initial transfer of the first mover. Escalating reciprocity means the focal player increases the degree of cooperation when possible (Fig. 1a), and unconditional full cooperation is an extreme case. When two escalators interact repeatedly, they converge on full cooperation, and in this sense escalation is a cooperative form of reciprocity. De-escalating reciprocity means the focal player decreases the degree of cooperation when possible (Fig. 1b), and unconditional full defection is an extreme case. When two de-escalators interact repeatedly, they converge on full defection, and thus de-escalation is an uncooperative form of reciprocity.

In a three-dimensional strategy space, a strategy consists of an initial transfer, as well as left and right intercepts for a linear response function (supporting information sections 1.2 and 2.1[26]). Three dimensions allow for all the strategies feasible in two dimensions, but with a number of additional possibilities. For example, three dimensions allow for ambiguous reciprocity. Ambiguous reciprocity means the focal player has a non-negatively sloped response that escalates low transfers and de-escalates high transfers (Fig. 1c). If an ambiguous reciprocator interacts repeatedly with a partner having any positively sloped response function, the players converge on intermediate levels of cooperation (supporting information section 1.2.8[26]). A four-dimensional strategy space adds strategies involving a wide range of non-linear response functions (supporting information section 2.3[26]). Some of the new possibilities include non-linear analogues of ambiguous reciprocity (Fig. 1d). New possibilities also include non-linear forms of reciprocity that do the opposite of ambiguous strategies by de-escalating low transfers and escalating high transfers (Fig. 1d). Such strategies punish

low transfers with even lower transfers and reward high transfers with even higher transfers.

**Cancellation effects at the individual level (all scenarios).** When a population is subdivided into groups and some individuals remain in the groups where they were born, relatedness within groups is present. When individuals play the social dilemma with ingroup partners, this relatedness allows cooperators to channel the benefits of cooperation towards other cooperators. Relatedness within groups can support the evolution of ingroup cooperation as a result, but it does not necessarily do so. Life history details, demography, and local ecological conditions can offset the effects of related individuals playing the game together[51]. Offsetting effects of this sort are cancellation effects at the individual level. Our models vary these cancellation effects by relying on two different life cycles (supporting information section 2.1.2[26]). In one case, the order of events in the life cycle is birth, game play, migration, group competition when relevant, and finally individual selection within groups. Game play and individual selection are decoupled. Individuals play the ingroup social dilemma with partners who are on average related to some extent. Relatedness increases the probability that cooperators end up playing together, which supports mutual cooperation. However, when individuals later compete within the group to reproduce, they compete against a different set of individuals precisely because migration occurs after game play but before individual selection. The timing of migration decouples the patterns of relatedness that hold when individuals play the social dilemma from the patterns of relatedness that hold when individuals compete to reproduce. As a result, related cooperators impose the gains from mutual cooperation as a relative advantage on others who are unrelated.

In the other case, the life cycle is birth, migration, game play, group competition when relevant, and individual selection within groups. Under this life cycle, game play and individual selection are coupled. Relatedness within groups ensures that cooperators are relatively likely to play with other cooperators. However, because migration occurs before game play, not after, cooperators who play together also end up competing against each other to reproduce. This cancels, to some extent, the degree to which relatedness supports the evolution of cooperation[24,25]. In our case, this cancellation effect at the individual level does not completely offset the value of playing the social dilemma with relatives. Under both life cycles, the evolution of cooperation increases with relatedness, though the effect is weak. Playing the game with related partners thus provides some limited support for the evolution of cooperation (supporting figures 15 and 16[26]). That said, cancellation effects at the individual level also play a role in the following precise sense. In models without group competition, the decoupled life cycle supports more cooperation than the coupled life cycle (supporting figures 15 and 16[26]).

Importantly, in terms of the link between game play and individual selection, decoupling is a relative concept. Under the decoupled life cycle, related cooperators who play the social dilemma together might still end up competing against each other at the selection stage. This outcome is possible simply because, even when migration rates are high, some individuals remain in the natal group. Thus, two individuals who play the game together may both stay in the same group and end up competing to reproduce later. The timing of migration does not completely eliminate this possibility because not everyone migrates. Instead, the decoupled life cycle ensures that individuals who play the social dilemma together are less likely to compete against each other than they would be under the coupled life cycle.

**Cancellation effects at the group level (group competition and joint scenarios).** Cancellation effects can also operate at the group level[8], and the intuition parallels that at the individual level precisely. Imagine a competition between two groups, one group composed of cooperative individuals and the other of uncooperative individuals. The cooperative group wins and replaces the losing group with a descendant group that is also relatively cooperative. If the parent and descendant groups go on to compete with two entirely different groups in the subsequent generation, both groups are relatively likely to compete against less cooperative groups and thus win their respective competitions. This maximizes the extent to which the group-level benefits of cooperation support the evolution of cooperation via group selection. In contrast, if the parent and descendant groups go on to compete against each other, then two cooperative groups compete against each other, with neither enjoying a relative advantage. This cancels the effects of the group-level benefits that result from both groups having many cooperative individuals.

Apart from a recent and important exception[8], multi-level selection models are like the former example. However, if ancestral human groups did not rove freely across the landscape in search of new competitions, which seems entirely plausible, ancestral conditions were at least somewhat like the latter example. To examine this distinction, we use a novel approach to manipulate cancellation effects at the group level (supporting information section 2.1.2[26]). The 40 groups in a population constitute a population of groups. In each generation groups are paired and have a competition with positive probability. We can interpret this setting as one in which paired groups occupy adjacent territories that place the two groups in close contact. At the beginning of each generation, $\Xi \in \{0, 20, 40\}$ groups are randomly selected to enter a pool of migrating groups that move around in space. These migrating groups are randomly redistributed to the open territories. The population of groups is well mixed when $\Xi = 40$. Groups move around a lot, and groups that win intergroup competitions are relatively unlikely to compete against their descendant groups in the subsequent generation. This minimizes cancellation effects at the group level. Anchoring the opposite extreme, $\Xi = 0$, which means groups never move. This maximizes group-level cancellation effects.

**The importance of differences in aggregate resources between groups (group competition and joint scenarios).** If paired groups engage in a group competition, as explained above, the group with more resources may or may not win the competition. Specifically, the probability of winning can be more or less sensitive to the difference in total resources between the two groups. We consider four levels of sensitivity (supporting information section 2.1.5[26]) controlled by the parameter $\lambda \in \{0, 10, 25, 100\}$. If $\lambda = 0$, which group wins is unrelated to the difference in total resources. Groups compete in this case, but outcomes are unsystematic. Therefore, group selection cannot occur, and in this sense $\lambda = 0$ is effectively like the repeated interactions scenario. As $\lambda$ values increase, the group with more resources is increasingly likely to win, and the group competition and joint scenarios are increasingly different from the repeated interactions scenario.

**Migration rates (all scenarios).** We vary the migration rate and by extension the relatedness within groups by allowing either 8 or 16 out of 24 individuals to migrate ($m_j$) per group per generation (supporting information sections 2.1.4–2.1.6 and 2.1.19[26]).

**Initial conditions (all scenarios).** In the initial generation, we seed the population with either (1) perfect reciprocators who initially transfer the full endowment, (2) unconditionally selfish individuals, or (3) individuals having random strategies drawn from a uniform distribution over the strategy space (supporting information section 2.1.8[26]). Perfect reciprocators start by transferring the maximum possible amount, if first mover, in the first interaction. For all subsequent choices, perfect reciprocators do exactly what their partners just did. In other words, they match the most recent transfers of their partners measure for measure. Seeding the population with perfect reciprocators represents initial conditions that are favourable for the evolution of cooperation,

while seeding the population with unconditionally selfish individuals represents initial conditions that are unfavourable.

Altogether, the three scenarios and six model characteristics yield 936 combinations. For each combination, we simulated 50 independent populations. In the main paper we focus on simulation results based on three-dimensional strategies. We occasionally discuss analytical results and simulation results based on two- and four-dimensional strategies. We especially do so for the repeated interactions scenario, where the dimensionality of the strategy space is decisive (supporting information section 1.2.12[26]). The supporting information[26] includes additional results and analyses, including those that go beyond the core project outlined here, and we also mention these results in the main paper as appropriate.

### Adding mistakes

The main paper presents results based on models that assume individuals never make mistakes. Theory based on repeated play of the standard prisoner's dilemma suggests this may not be an innocent assumption. Without mistakes, different cooperative strategies can drift in and out of the population because the strategies in question lead to identical choices[36,52]. The population eventually drifts towards some mix of cooperative strategies that is vulnerable to invasion by an uncooperative strategy, and cooperation collapses. With mistakes, however, these same cooperative strategies no longer generate identical choices. Drift accordingly plays a reduced role, and mistakes can stabilize a specific cooperative strategy from among a glut of cooperative strategies[52].

Because of the potential importance of mistakes, we added mistakes and repeated our entire simulation study. A mistake occurs when an actual transfer deviates from the transfer stipulated by an individual's strategy. We implemented mistakes by distributing actual transfers around the stipulated transfer (supporting information section 5[26]). Mistakes are thus common, but they vary in magnitude. For three-dimensional and four-dimensional strategies, results remain, in effect, exactly the same. In the two-dimensional case, under repeated interactions as a stand-alone mechanism, mistakes dramatically slow down the invasion of cooperative strategies compared to otherwise identical situations without mistakes. As a result, over long but finite time scales, repeated interactions cannot support the evolution of cooperative strategies even when strategies are two-dimensional. This limitation opens the door for group competitions to interact positively with repeated interactions, which is exactly what happens (supporting information section 5.3[26]). Mistakes thus expand the range of conditions that lead to the evolution of super-additive cooperation. Future research could vary the structure of mistakes when actions are continuous to see how robust this conclusion is.

### Sequential social dilemma in Papua New Guinea

We conducted our experiment with members of Perepka and Ngenika groups, two horticultural groups in the Western Highlands of Papua New Guinea (supporting information section 6[26]). The Western Highlands are an ideal place to evaluate evolutionary theories of human cooperation because the people who live there, relatively speaking, are beyond the reach of state institutions. Social preferences, local norms, reciprocity and group affiliation are the main forces that govern social life. These forces were probably pervasive for much of the human evolutionary past, and so they are the primary points of contention with respect to the evolution of human cooperation. By contrast, the enforceable contracts and legal institutions of contemporary large-scale societies introduce additional forces that are recent in evolutionary terms. This can confuse the interpretation of empirical findings by confounding ancestral psychologies with incentives, norms, and expectations tied to contemporary institutions.

At the time of the experiment, the Ngenika and Perepka groups inhabited territories separated by about 30 km in the Western Highlands. Although each group was aware of the other's existence, no one had any memory of hostilities between the two groups. With adult participants, we implemented a sequential social dilemma that included both ingroup and outgroup pairings (supporting information sections 7 and 8[26]). One author (H.B.) grew up and lived in the local area for 15 years, speaks the local language (Tok Pisin) fluently, and has a detailed knowledge of the values and cultural practices of local populations. This knowledge ensured that the experiments could be conducted in the local language and in a manner respectful of local cultures. Participants provided informed consent verbally. The Internal Review Board of the Faculty of Business, Economics and Informatics at the University of Zurich approved the study.

The players in a pair were each provided with an endowment of five Papua New Guinean Kina. This endowment was roughly half of a high daily wage for a labourer in the informal sector of the local workforce. Most participants earned less than this daily wage on average because they were not working for money on a regular basis. After receiving the endowment, the first mover in a pair transferred some amount between zero and five Kina, in increments of one Kina, to the second mover. The experimenter doubled this transfer. Before learning the amount actually transferred, the second mover specified an amount she wished to back transfer to the first mover for each of the first mover's possible transfer levels, yielding six observations per second mover. This is the strategy method of eliciting second mover responses, and previous research has shown it to be a reliable method for measuring behavioural strategies[53]. After eliciting the second mover's strategy, the experimenter revealed the amount actually transferred by the first mover and implemented the appropriate back transfer. The experimenter also doubled the back transfer.

Using a between-subjects design, we implemented four treatments that differed in terms of the group affiliations of the two players. We varied affiliations in all combinations, which yielded two ingroup treatments (Ngenika/Ngenika and Perepka/Perepka) and two outgroup treatments (Ngenika/Perepka and Perepka/Ngenika). We used no statistical methods to pre-determine sample size (see Reporting Summary). All players knew the rules of the game. Each player also knew the group affiliation of her partner. The experimenter mediated all interactions in private, and so interactions were anonymous apart from information about group affiliations.

### Reporting summary

Further information on research design is available in the Nature Portfolio Reporting Summary linked to this article.

## Data availability

Data from the experiment in Papua New Guinea and supporting information are available at https://doi.org/10.5281/zenodo.10355347. Source data are provided with this paper.

## Code availability

Simulation code is at www.github.com/cmefferson/superAdditive-Cooperation in the directories {two,three,four}DimSimFiles. These three directories correspond to two-, three-, and four-dimensional models, respectively.

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

**Acknowledgements** The authors thank M. Sirdey and F. Calvo, and seminar participants at the Institute for Advanced Study Toulouse, the Sante Fe Institute, Harvard University, and the universities of Amsterdam, Lausanne, Konstanz and Zurich. C.E. acknowledges the support of the Swiss National Science Foundation (grant nos. 100018_185417 and 100018_215540).

**Author contributions** C.E. and U.F. designed the simulations and coded them independently. C.E. analysed the data produced by the simulations and interpreted the results with input from U.F. and E.F. C.E. developed the associated analytical models. H.B., U.F. and E.F. designed the experiment, and H.B. organized and conducted the experiment in Papua New Guinea. C.E. and H.B. analysed the data from the experiment. C.E. synthesized results from the simulations and experiment. C.E. and E.F. wrote the paper with input from H.B. and U.F.

**Funding** Open access funding provided by University of Zurich.

**Competing interests** The authors declare no competing interests.

**Additional information**
**Correspondence and requests for materials** should be addressed to Charles Efferson or Ernst Fehr.

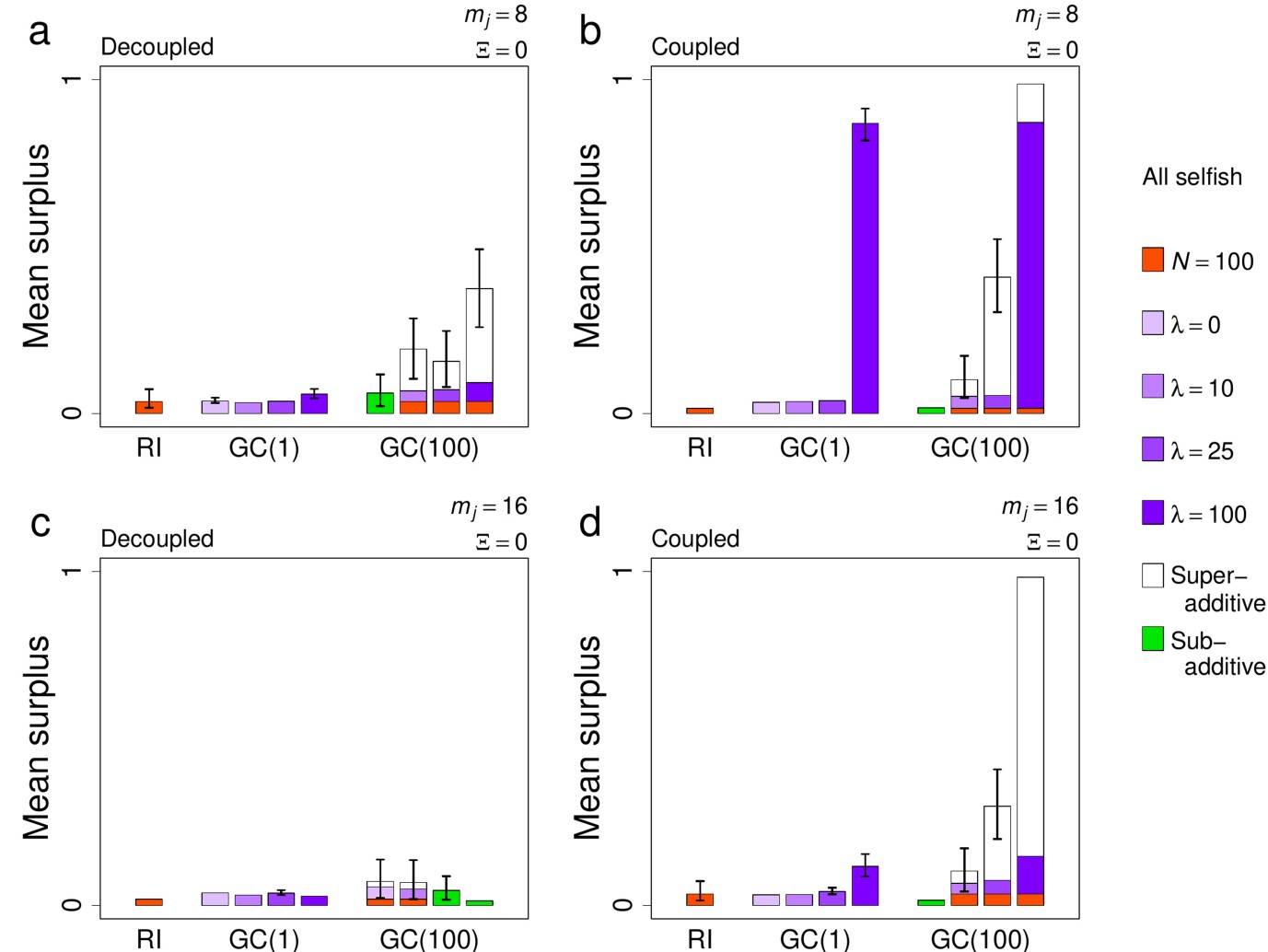

**Extended Data Fig. 1 | Super-additive cooperation when initial conditions and group mixing are relatively unfavourable for the evolution of cooperation.** The graphs show the mean surplus per individual per ingroup interaction from the final generations of evolutionary simulations. Error bars show 95% confidence intervals, which are calculated by bootstrapping over 50 independently simulated populations and omitted when extremely narrow. RI signifies the repeated interactions scenario with $n = 100$ interactions per ingroup pair. GC(1) indicates the group competition scenario as competition outcomes vary in sensitivity to group differences ($\lambda$). GC(100) combines the two component mechanisms into the joint scenario, with the bars from left to right corresponding to increasing $\lambda$. When the joint scenario is super-additive, the mean surplus is decomposed (supporting information section 3[26]) into the repeated interactions effect (orange), the group competition effect (purple),

and the super-additive effect (white). Initial conditions consist of a population of unconditionally selfish individuals (All selfish), and group mixing is as low as possible ($\Xi = 0$), both of which are relatively unfavourable for ingroup cooperation. That said, results are nearly identical when initial conditions consist of a population of perfect reciprocators (supporting figure 163[26]), which reveals that initial conditions actually have no meaningful effect. The life cycle either decouples (**a, c**) or couples (**b, d**) game play and individual selection. The number of migrants per group per generation ($m_j$) is either relatively low (**a, b**) or high (**c, d**). Cancellation effects at the group level[8] hinder cooperation in general. Moreover, the initial population consists entirely of selfish individuals, and cooperative strategies must actually invade to become established. Super-additivity is nonetheless common (**a, b, d**) and in some cases the result of extremely strong positive interactions (**d**).

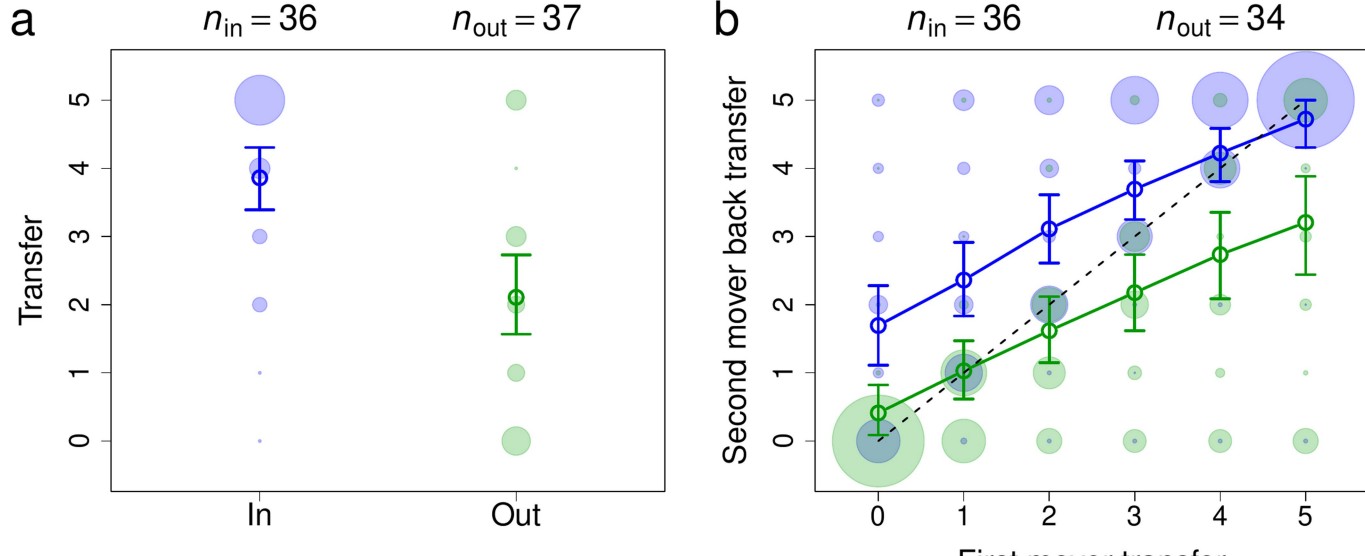

**Extended Data Fig. 2 | First mover transfers and second mover back transfers.**
**a**, The mean transfers of first movers with ingroup partners (36 participants) were relatively high and with outgroup partners (37 participants) relatively low. Error bars are 95% bootstrapped confidence intervals. Bubble plots show the distribution of transfers conditional on ingroup versus outgroup. **b**, Second movers with ingroup partners (36 participants, blue) exhibited escalating reciprocity, and second movers with outgroup partners (34 participants, green) exhibited de-escalating reciprocity. Point estimates show mean transfers, and error bars are 95% bootstrapped confidence intervals clustered on second mover (six choices per second mover). Bubble plots show the distribution of back transfers jointly conditional on first mover transfer and ingroup versus outgroup. A minor qualification concerns second mover outgroup strategies (**b**, green) conditional on a first mover transfer of zero. Bubbles show that in this situation almost all second movers also chose zero. A handful, however, chose positive amounts, and the 95% confidence interval does not quite overlap zero. This pattern is consistent with our simulations in the sense that mutations and demographic stochasticity ensure that some agents have strategies dictating small but positive transfers in response to transfers of zero from an outgroup partner. The average outgroup response function is thus a form of quasi-de-escalation (Fig. 4).

| | |
|---|---|

# Reporting Summary

## Statistics

For all statistical analyses, confirm that the following items are present in the figure legend, table legend, main text, or Methods section.

| n/a | Confirmed | |
|---|---|---|
| ☐ | ☒ | The exact sample size (*n*) for each experimental group/condition, given as a discrete number and unit of measurement |
| ☐ | ☒ | A statement on whether measurements were taken from distinct samples or whether the same sample was measured repeatedly |
| ☐ | ☒ | The statistical test(s) used AND whether they are one- or two-sided<br>*Only common tests should be described solely by name; describe more complex techniques in the Methods section.* |
| ☐ | ☒ | A description of all covariates tested |
| ☐ | ☒ | A description of any assumptions or corrections, such as tests of normality and adjustment for multiple comparisons |
| ☐ | ☒ | A full description of the statistical parameters including central tendency (e.g. means) or other basic estimates (e.g. regression coefficient) AND variation (e.g. standard deviation) or associated estimates of uncertainty (e.g. confidence intervals) |
| ☐ | ☒ | For null hypothesis testing, the test statistic (e.g. *F*, *t*, *r*) with confidence intervals, effect sizes, degrees of freedom and *P* value noted<br>*Give P values as exact values whenever suitable.* |
| ☒ | ☐ | For Bayesian analysis, information on the choice of priors and Markov chain Monte Carlo settings |
| ☒ | ☐ | For hierarchical and complex designs, identification of the appropriate level for tests and full reporting of outcomes |
| ☐ | ☒ | Estimates of effect sizes (e.g. Cohen's *d*, Pearson's *r*), indicating how they were calculated |

*Our web collection on statistics for biologists contains articles on many of the points above.*

## Software and code

Policy information about availability of computer code

| Data collection | No software was used for collection of experimental data. |
|---|---|
| Data analysis | Data were analyzed with R (4.1.3). |

For manuscripts utilizing custom algorithms or software that are central to the research but not yet described in published literature, software must be made available to editors and reviewers. We strongly encourage code deposition in a community repository (e.g. GitHub). See the Nature Portfolio guidelines for submitting code & software for further information.

## Data

Policy information about availability of data

All manuscripts must include a data availability statement. This statement should provide the following information, where applicable:
- Accession codes, unique identifiers, or web links for publicly available datasets
- A description of any restrictions on data availability
- For clinical datasets or third party data, please ensure that the statement adheres to our policy

The experimental data and code for analyses are available at www.github.com/cmefferson/superAdditiveCooperation in the directory "data".

## Human research participants

Policy information about studies involving human research participants and Sex and Gender in Research.

| | |
|---|---|
| Reporting on sex and gender | At the time of data collection, the difference between sex and gender was not a salient distinction among Perepkas and Ngenikas. Before participating in the experiment proper, each participant responded to a short questionnaire. We collected data on the gender/sex of participants at this time. The data are used as controls in statistical analyses and are available with the publicly posted raw data. However, the data are fully anonymized, and individuals are not identifiable. 37% of participants were female. |
| Population characteristics | See above. |
| Recruitment | See below under "Sampling strategy" for a complete description of recruitment and sampling. As is typically true for behavioral experiments, recruitment into the study was not representative. Conditional on participating, however, assignment to treatment was random, and assignment to role as Player 1 or Player 2 was random. |
| Ethics oversight | IRB of the Faculty of Business, Economics and Informatics at the University of Zurich. |

Note that full information on the approval of the study protocol must also be provided in the manuscript.

## Field-specific reporting

Please select the one below that is the best fit for your research. If you are not sure, read the appropriate sections before making your selection.

☐ Life sciences    ☒ Behavioural & social sciences    ☐ Ecological, evolutionary & environmental sciences

For a reference copy of the document with all sections, see nature.com/documents/nr-reporting-summary-flat.pdf

## Behavioural & social sciences study design

All studies must disclose on these points even when the disclosure is negative.

| | |
|---|---|
| Study description | The empirical study was a standard behavioral experiment in the tradition of experimental economics (e.g. no deception of participants, choices were incentivized). Specifically, the experiment was a one-shot sequential social dilemma with a continuous action space that is essentially a one-shot symmetric trust game. |
| Research sample | Experimental subjects were adult Perepkas and adult Ngenikas from the western Highlands of Papua New Guinea. The main paper discusses at length the rationale for using this sample when examining the evolution of cooperation. |
| Sampling strategy | The sample was a convenience sample. Specifically, Helen Bernhard visited each of the two groups on two separate days to recruit participants, which meant a total of four days recruiting. On a given day, she walked through the settlements and talked to adults present and invited them to the study in a few days time. No statistical methods were used to pre-determine sample size, and practical concerns (e.g., time in the field) were critical. However, with two treatments, our sample size would provide adequate power (alpha = 0.05, beta = 0.8), given an OLS model fully saturated with respect to experimental design, for an approximate effect size associated with $R^2 = 0.9$, i.e. Cohen's $f^2 = 0.111$. In practice, this is a lower bound for Players 2 because we have multiple observations per player. |
| Data collection | Data were recorded with pen and paper. Helen Bernhard conducted the experiment in-person. Her spouse, who does not speak Tok Pisin, was nearby to pay participants their show-up fees and provide participants with refreshments. |
| Timing | Data were collected in July 2004. |
| Data exclusions | No data were excluded. |
| Non-participation | No participants dropped out or declined participation. Some participants could not participate because they did not correctly answer a series of questions that tested comprehension of the game. To proceed to the experiment proper, a person had to answer all of these questions correctly. This was a pre-determined criterion in the following. People who did not meet the criterion received a show-up fee but did not participate in the experiment itself. |
| Randomization | Treatments and role (Player 1 vs Player 2) were first assigned to specific numbers. Participants were randomly assigned to treatment and role by blindly drawing numbers written on small pieces of paper from a bowl. |

## Reporting for specific materials, systems and methods

We require information from authors about some types of materials, experimental systems and methods used in many studies. Here, indicate whether each material, system or method listed is relevant to your study. If you are not sure if a list item applies to your research, read the appropriate section before selecting a response.

## Materials & experimental systems

| n/a | Involved in the study |
|-----|------------------------|
| ☒ | ☐ Antibodies |
| ☒ | ☐ Eukaryotic cell lines |
| ☒ | ☐ Palaeontology and archaeology |
| ☒ | ☐ Animals and other organisms |
| ☒ | ☐ Clinical data |
| ☒ | ☐ Dual use research of concern |

## Methods

| n/a | Involved in the study |
|-----|------------------------|
| ☒ | ☐ ChIP-seq |
| ☒ | ☐ Flow cytometry |
| ☒ | ☐ MRI-based neuroimaging |

