## [Peer Review File · Nature]

Manuscript Title: Super-additive cooperation

Editorial Notes:

Reviewer Comments & Author Rebuttals

Reviewer Reports on the Initial Version:

Referees' comments:

Referee #1 (Remarks to the Author):

Super-additive cooperation

By Charles Efferson, Helen Bernhard, Urs Fischbacher & Ernst Fehr

In this paper, the authors show that repetition and group structure together can produce levels of cooperation that are significantly higher than what you would expect if you just take the level of cooperation that you get with repetition, and the level of cooperation that you get with group selection, and add them up. The argument is somewhat complex to make, because they also have some things to say about the benchmarks of these ingredients on their own. They show that the repetition benchmark from the literature is rather optimistic and follows from the choice of details of the game. The standard stage game typically is binary and simultaneous; it only allows for cooperation and defection in the PD (and, with repetition, for strategies that condition those binary choices on the past) and assumes that players decide simultaneously. If one allows for intermediate levels and assumes that players take turns, and if one lets the ways in which players can respond be sufficiently flexible functions of what the other does, then that undermines the level of cooperation that evolves.

While the literature benchmark on repetition is more uniform (but, as the authors argue, too optimistic), the literature benchmark on group selection is more diffuse, and somewhat polarized, with what I guess is a majority position that sees no role for group selection at all. This complicates the description of the benchmark in the literature a bit, but of course what one can do, and what the authors did, is show by simulations that the levels of cooperation in their model are not at all spectacular, unless the intensity of between-group competition is really ramped up enormously. Also important: they take into account the cancellation effect at the group level, which is typically not done in the literature. While some of the literature could not have lower expectations of group selection to begin with, taking the cancellation effect at the group level into account does adjust the benchmark downwards for papers that base their optimism on models that ignore the cancellation effect at the group level. With these re-calibrated benchmarks, they show that the interaction effect between those is more than substantial. (Note that the spectacular interaction effect is not the result of their recalibrated benchmarks, because those are just what they are in isolation in their setup). They end with an experiment in Papua New Guinea, the findings of which match the model predictions.

I am very enthusiastic about this paper. Progress in the literature on the evolution of (human) cooperation has slowed down in recent years, and not because we are so close to the answer. This is for the first time in quite a while that I have seen a paper that does bring something new to the table. The content of their message is unavoidably somewhat complex, but the authors do a good job at communicating it (there might be some room for improvement, though, that I think the authors would welcome; see the bit on **initial conditions** below). Also, the existing literature is regularly a bit weak on the link between theoretical results and experiments; these parts of the literature have developed somewhat independently with loose reference to each other. This paper is a more than welcome exception to this.

Of course, one could have other ideas on how to optimally present this; others would perhaps cut the findings up into different papers; one about the benchmark for repetition; one for the interaction effect; and maybe even a separate paper with the experiment. That would also be a legitimate preference, but so is the preference to put all three in one combined argument, given that the main point really is the super-additive way in which the two mechanisms combine, and the fact that one cannot make that point without getting the benchmarks right (I always hate it if reviewers upgrade considerations between different legitimate alternatives on side issues to reasons for not publishing the results).

Initial conditions There is a way in which I think the authors could kill two birds with one stone and improve the simulation results both in accuracy and with respect to ease of presentation. Usually there is a tradeoff between the two, but here there might not be, and both could perhaps be improved.

If we look at figure 2, then the results there depend on the initial conditions. If I think of the simulations as a way to estimate properties of the invariant distribution of a Markov chain, then dependence on initial conditions would suggest that there are two parts of the state space that are (practically) disconnected, and that one set of initial conditions make the system likely to go to one, and another set of initial conditions makes the system go to the other. The final averages then are averages of the system stochastically rummaging around what would be one attractor in the absence of noise, starting from one place, versus stochastically rummaging around another, starting from another place.

Now when I first saw that in Fig 2, I was a bit surprised, and curious why this system would have two somewhat separate, relatively stable sets of states (notice that everything other than the initial conditions is the same). Looking at the results in the Supplementary Information, however, I think there might be another, more trivial reason why the simulations come out this way. The starkest indication is in the panels on the right-hand side of figures 108 and 109 of the SI. There, I assume that each line represents a simulation run (I could not find where that is stated explicitly, but this makes the most sense, and if they would represent different groups within a population, what follows also applies, but with a little rephrasing). These panels suggest that there is only traffic from negative degrees to positive ones; none of them have gone from negative to positive and back, but many have gone just from negative to positive. This suggest that it might only be a matter of time that all of them will be positive, and the simulations have just not run long enough for all runs to get there (*when* a run passes from negative to positive is a random variable, and the figure suggests that for the current run length, there is a substantial probability that that has not happened yet by the end of a run). The average over some interval at the end, averaged over the runs, therefore is still not reflective of the properties of the invariant distribution, because a bunch of runs haven't gone positive yet.

The simulation approach of doing very many runs, and then taking averages over them, I think is inefficient, in the sense that for every run, you have to wait for the dependency on the initial state to wear off, and the computation time you waste on letting the initial condition wear off, you now waste 100 times, if you do 100 repeats. It is much better to do far fewer runs (if there is no indication that the system is not ergodic, and unless I am missing something, I don't think there is here) and let them run for much longer. In that

sense, it is not a surprise that with this approach, it seems that things depend on the initial conditions, where that is really just a consequence of the choice how often and how long to simulate; the averages as they are, still contain too many transient population states (this is not very precise, but anyway, states that every run passes through on their way from the initial condition towards the region of attraction, and that are thereby overrepresented).

So, in order to check if this might be an explanation, I would suggest to take one combination of choices for the parameters of the model, repeat this, but with far fewer, and much longer runs.¹ If that makes the dependence on initial conditions go away, then that is good news on all fronts; the Markov chain is “normal” (not non-ergodic, and, more importantly, also not almost non-ergodic, with multiple sets of states with super little traffic between them, relative to the length of the run) after all, and the authors can save a lot of presentation space, because now the variable “initial conditions” does not need to be taken into consideration anymore. This will really help, given the number of figures they have now. Also, the results will gain in precision, because if the results presented depend on initial conditions for the wrong reasons, and the invariant distribution does not, then that implies that at least one of them is not 100% spot on.

I am not an expert on statistical properties of simulation models, but whatever the error bars capture, if my hunch is correct, they miss out on something relevant, which would be picked up if one would look at what happens, averaged over the simulation, over time; this would then probably still detect an (upward) trend in the degree (or another variable), averaged over runs, towards the end of the run time. Also, if the error bars become smaller with fewer, but longer simulation runs (cutting off a longer initial stretch, but also keeping a longer final stretch to average an individual run over), that could be in line with my hunch. If you would establish that the run time is long enough for the population to very, very likely not get stuck anywhere, I think I don't even care all that much about the error bars, as we don't have to worry that the results are created by a fluke.

None of this undercuts the findings in the paper in any way, shape, or form; it's just that checking this would be worth it, because if my hunch is correct, then that would save a lot on figure space, and make presenting the results easier.

Repeated games Given that the message is already quite complicated, maybe it is best not to complicate the discussion of the benchmarks further, but also in the classic setting, with simultaneous, binary repeated prisoners' dilemmas, the literature is biased towards thinking that evolution favors cooperation more than it really does. The most neutral results are from van Veelen, García, Rand & Nowak, 2012 (and some econ publications by the same authors) where they find that even for very high repeat probabilities, the average level of cooperation typically does not go over 50%, even for quite favorable b/c ratios. That is not as bleak as what the authors find with a 3-

¹ I expect computation time is an issue, so therefore I could also imagine that it is worth playing around with the population size, mutation probability, and perhaps selection strength to max out efficiency. If the results are not too sensitive to the population size, I would trade pop size for more simulation length, especially because the computation time might not be linear in pop size. Also increasing the mutation rate a bit could perhaps speed things up a little without changing the results. I am sure the authors thought of this, but I just want to be complete.

dimensional strategy space, but also not quite as high as what one would think if one reads the literature.

Group competition Simon, Fletcher & Doebeli, 2013, (and some bio publications by the same authors) is a good reference for technically advanced group selection modeling. Given these models, I am curious about details of the dynamics in the manuscript. Individuals play in groups, but there is migration, so therefore I am curious if the population as a whole transitions between equilibria, and the population is close to monomorphic at any point in time (as in van Veelen, García, Rand & Nowak, 2012, which is also a paper about the interaction between two mechanisms, one of which is repetition) or if the “equilibrium state” of the population is more like a dynamic equilibrium, with groups, within which individuals are more or less doing the same, but in which there is an equilibrium amount of heterogeneity between groups (this would be group selection that is more in the style of Simon et al., 2013). Both would be OK, I’m just curious.

The authors moreover cite Price (1970), which they may do intentionally, but just in case, it is Price (1972) that deals with group selection specifically.

Precise / concise There are some things that are precise in the SI, but, understandably, a bit less so in the main text. For instance, “decoupled” and “coupled” are described as relative terms in the SI, but in the main text, one can infer that they must be relative from the setup of the model, and while reading I was wondering if I missed something, because coupled and decoupled sounded a bit absolute there. Another example is that I was a bit confused how the repeated version works; what I guessed it was, is what it is indeed, but because of the strategy set of the one-shot game, which specifies something for the 1st mover in a quite different way than it does for the 2nd mover, there is (a tiny bit of) space for uncertainty as to how the 2nd round 1st mover determines the level of cooperation (and every subsequent 1st mover).

The experiment I didn’t focus much on the experiment, but if one has the opportunity to test the predictions of the model with a non-WEIRD population for whom we don’t have to rely on assumptions that would justify assuming that something in the lab is a good proxy for ingroup/outgroup in population structures that we don’t live in anymore, then why not? That makes it not just cute to test it in Papua New Guinea. We should not make it a habit to test everything evolutionary in some small-scale society, but here it does have value added in making the experiment relatively clean.

I would love to think more about this! and there are some details that I hope to understand better at some point, but time is limited, and it is safe to say that this is really interesting, so I would suggest a revision, where the only real change would be to investigate my initial conditions hunch, and change the presentation of the findings accordingly if that checks out.

Small things:

Page 13: the group that wins is unrelated to the difference in total resources → which group wins is unrelated to the difference in total resources?

References

- Price, G. R. (1970). Selection and covariance. *Nature*, *227*, 520-521.
- Price, G. R. (1972). Extension of covariance selection mathematics. *Annals of human genetics*, *35*(4), 485-490.
- Simon, B., Fletcher, J. A., & Doebeli, M. (2013). Towards a general theory of group selection. *Evolution*, *67*(6), 1561-1572.
- Van Veelen, M., García, J., Rand, D. G., & Nowak, M. A. (2012). Direct reciprocity in structured populations. *Proceedings of the National Academy of Sciences*, *109*(25), 9929-9934.

Referee #2 (Remarks to the Author):

There is a lot to like in this manuscript, including the integration of evolutionary modelling and simulations, and behavioral experiments with participants from non-western populations. The conclusion that propensity to cooperate evolved in the joint context of repeated (in-group) interactions and intergroup competitions makes sense, and the paper offers useful 'boundary conditions' on when and how these mechanisms selected for cooperation. Below are specific comments on the paper, some more general and some more specific.

General Comments

1. The abstract is too technical and difficult to understand. It was not until I was well into the paper that it started to make sense. A more accessible abstract can make the paper more impactful. (this applies at times to other aspects of the paper, and some editing may help)
2. As a reader, it was not fully clear what the authors' main aim is. Is it that contemporary theories on the evolution of cooperation are wrong, and some integration actually better? It is that one-shot cooperation (more with partners from one's own community) is puzzling? Perhaps a combination, or even something else? In its current form, the paper touches on many issues, and provides insights both small and big. But the take-away in terms of its implications for any of the above questions could be anticipated and spelled out more clearly.
3. As an aside – I agree that some have tried to explain cooperation in one-shot games with reference to

ancestral environments (putatively) marked by repeated interactions and reputational concerns alongside 'priming.' Since such hard-to-falsify claims have been made, sophisticated alternatives have been offered that need no reference to evolutionary selection pressures (e.g., institutions, norms, socialization/social preferences). Is there reason why these are ignored in the current reporting?

4. The experiments are about one-shot interactions with an in- and an out-group member. Results are quite straightforward when other literatures are taken into account (a meta-analytic summary can be found in Balliet, Psychol Bull. 2014). Some details in the form of reciprocity are not, however, identified previously and this is a novel and interesting aspect of the experiments.

5. Simulations are used to identify when and why the reciprocation strategies 'survive' and may (have) evolve(d). This assumes, of course, that the forms observed in the experiments are empirically robust, replicable, and generalizable. Knowing this is needed to have the simulation results being general rather than limited to a very specific case, and perhaps one that exists in theory without general links to human behavior. I'm not sure, but perhaps the work by Romano and Sutter (Nat Comm 2021) may be of help here?

6. Simulations provide a hypothesized mechanism producing a particular outcome. The experimental data reveal that particular outcome, and are inferred to support the hypothesized mechanism. This 'reverse inference' may be problematic – how can we know that it is the joint mechanism of repeated interaction and intergroup competition that explains the behavioral data? I have two questions: (i) could any of the mechanisms referred to under [3] not (also) play a role; and (ii) were simulations blind to or (implicitly) informed by the results from the behavioral experiment? If informed, how could this have affected parameter choices and foci and, inadvertently perhaps, caused a particular focus on a subset of mechanisms and boundary conditions?

7. Group-selection is debated and the field converges on it being a limited explanation for (group-level) cooperation. The paper mentions group-selection in passing, without fully developing it. Is it needed? Do (simulation or experimental) results inform the debate, and how?

Minor Comments

8. P. 3/2nd para. Sentence "The hypothesis is not paradoxical...anything else" is hard to understand and feels not needed?

9. My reading of some of the parochial altruism literature always was that intergroup competition ups within-group cooperation because cooperation helps to win the intergroup competition. If people can migrate with their earnings after cooperating but before confronting hostile out-groups, this "classic" mechanism no longer applies. Perhaps I misunderstood, but is this part of the simulation 'fair' towards the intergroup competition mechanism? And even if so, are results then not revealing (indirect) support for the mechanism – once migration is allowed, intergroup competition no longer has meaningful effects on cooperation?

10. The term 'super-additive' can be easily misunderstood as suggesting that both mechanisms do the trick, but the combination over and beyond that (an ordinal interaction); However, this seems not what the authors find – neither is sufficient and both are needed seems to capture results better?

Referee #3 (Remarks to the Author):

This paper reports on an important and counterintuitive result — the ineffectiveness of repetition, or between-group selection on their own to generate cooperation, and the qualitatively different levels of cooperation that can be attained when both these mechanisms are combined. The authors have mostly addressed this using a theoretical model in which a wide range of potentially relevant parameters can be varied. They have found an extremely robust set of conditions for cooperation to evolve, which has key implications for understanding the origins of human cooperation.

I am thrilled that this study critically examines the robustness of repetition as a solution to cooperation. In the evolutionary literature, it is widely assumed that repeated interactions can solve the problem of cooperation. However, there are reasons to doubt that this is the whole story. On theoretical grounds, previous work has shown that when the range of possible strategies allowed in the evolutionary mix is broad, conditionally cooperative strategies can be destabilized, and populations may cycle between high and low levels of cooperation. Ref 35 (van Veelen et al 2012) is a good example illustrating this effect, as is an old article in this very journal from 1987 (“No pure strategy is evolutionarily stable in a repeated prisoner’s dilemma game” by Boyd and Lorberbaum). This paper adds substantially to these earlier works by examining a continuous sequential PD, and shows that when the strategy space is not artificially restricted—cooperation cannot persist. This should catch people’s attention in and of itself, but the paper offers us even more. Specifically, the paper convincingly reveals a robust set of conditions in which repeated interactions can indeed sustain cooperation — intergroup competition, and then goes on to show empirical results that align nicely with the predictions of the model. Taken together, it is an impressive study, and a result that should make people seriously reconsider how reciprocity-based cooperation can evolve.

This paper identifies inter-group competition as the solution, and finds that relatedness is not effective at generating cooperation. This differs from the finding in van Veelen et al where relatedness and repetition together can yield cooperation. However, the model set up here is different with cooperation choices being continuous. Realistic cooperative problems that humans solved are likely to have been continuous, so this extension is critical. Another solution (e.g. Boyd 1989) is that mistakes can allow cooperation to be stable in the repeated Prisoner’s Dilemma game. In the current paper the authors suggest that not allowing for mistakes gives the best chance for cooperation. This is true when examining how particular strategies handle errors, because many common and popular strategies don’t handle errors well. But, when a wide mix of strategies are allowed in an evolving population, mistakes can help the cooperative equilibrium to not be destabilized. If it is possible for the authors to

incorporate mistakes into the model, it would allow us to know what other alternatives, if any, to intergroup competition can play help stabilize cooperation. This extension could greatly increase the impact of the paper. If it is beyond the scope of this study, I think it will be good to at least engage with that literature in the Discussion, and highlight the possibility that mistakes might offer a similar relief for cooperation as intergroup competition does in the current model.

On empirical grounds, one puzzle of reciprocity is why it is not more common across animals that live in stable social groups. The evidence for reciprocity is sparse, and most examples of it are in low-stakes contexts like grooming. But the theory predicts we should see high-cost/high-benefit forms of exchange regularly evolving in species where the chances of interacting with group mates again are high, which is easily met in many vertebrate societies. While relatedness offered a magical solution to cooperation, with concomitant pervasive empirical evidence for it across nature, repetition offered a magical solution with nothing much to show for it in nature. Humans on the other hand are able to manage high-stakes exchange on the basis of repeated interactions. Models like the current one are absolutely essential for us to understand how the conditions for reciprocity to evolve are more easily met in humans than other animals. For instance, this work tells us that if between-group competition is a more important force in humans than other animals, it could help explain the discrepancy in reciprocal cooperation between humans and other animals! That is possible if cultural differentiation, which only humans have, allows for much stronger between-group selection, as has been argued by many. There is also a neat analogy between this result and the literature on cultural group selection being an equilibrium selection process — mostly talked about in the context of favoring group-beneficial norms over a host of other norms; in the current model it provides cooperative reciprocal strategies, not norms, that little extra help. In both cases the group-selection force can be effective because the behaviors in question are not costly to individuals (due to norm enforcement, or in this case due to repeated interactions). The authors don't venture into this space, and stay pretty close to the model results in the Discussion. But there are indeed much broader implications.

The consideration of the cancellation effects and the individual and group level is valuable, as these are important evolutionary forces that are often ignored in models of cooperation, especially the group-level cancellation effect.

Two minor comments:

1. Why are one-shot interactions in the group competition scenario not unconditional defection, but rather de-escalating strategies. Can you provide the readers some intuition for this result, especially because the intro focused on the puzzle of one-shot cooperation.
2. Regarding the framing: While I see the original motivation for the study had to do with the one-shot versus repeated interactions debate about human cooperation in transient interactions, it doesn't seem like the model ended up speaking too much to this question. It speaks to a more interesting question — i.e. can repeated interactions favor cooperation in repeated interactions, as the authors say in the

Discussion. So, in terms of the write-up, there is a bit of a mismatch between what the Intro suggest will be the meat of the paper, and what the meat of the paper turns out to be.

Author Rebuttals to Initial Comments: Response to Referee #1

We'd like to thank you for your astute and extremely helpful report on the first submission of our paper, "Super-additive cooperation." We are truly delighted to know that you are "*very enthusiastic*" about the paper, and that you believe it "*bring[s] something new to the table*" for the "*first time in quite a while.*"

Equally important, however, your detailed comments and questions pushed us to include a number of additions to the revised paper. Specifically, ***in light of your comments, we reran all simulations over a much longer time frame than before. We did this both for the original models and for a new set of models that include implementation errors (i.e. choice mistakes).*** In addition, again in response to your insights, we analysed heterogeneity and population structure for all simulations, and we include these results in the revised supplement. Finally, we have rewritten and expanded various sections of the main paper to clarify key points and answer important questions. Many of these revisions were due to your constructive comments.

We feel the paper has improved greatly as a consequence of the important insights and suggestions offered by you, the editor, and your fellow referees. We hope you will agree and support publication of the revised version in *Nature*. For your convenience, we now summarize the changes we made specifically in response to your comments and questions.

Are model behaviours really sensitive to initial conditions, or did we simply not run simulations long enough? To find out, run the simulations longer. Reduce the number of simulated populations per parameter combination to keep the requirements for computing power under control.

We owe you a debt of gratitude for your observations and suggestions about initial conditions. You suggested that model results, especially with respect to the repeated interactions scenario, were probably not sensitive to initial conditions. Instead, we had simply not run all simulations long enough for the system to stabilize. You also proposed that, if computing resources were an issue, we should reduce the number of simulated populations per combination of parameter values.

Indeed, we were concerned about computing resources after reading your comments and the comments of Referee #3. Referee #3 indicated that we should add choice mistakes and rerun simulations with this addition in place. Together with your recommendations, we estimated a dramatic increase in the scale of the project.

Happily, however, the University of Lausanne has recently invested heavily in high-performance computing. We did not use these resources for our original submission, but we did for the revision. ***As a result, we were able to do everything you and Referee #3 asked without any meaningful trade-offs. We increased all simulations from 50,000 generations to 150,000 generations,*** and we were able to do so without reducing the number of simulated populations per parameter combination. In addition, ***we were able to add choice mistakes and repeat the entire project,*** again with 50 populations per parameter combination and 150,000 generations per population.

Your intuition was correct! Long-run results are not sensitive to initial conditions. The revision includes a new Fig. 2 that shows exactly this. For your convenience, we show this figure below. Initial

conditions matter for neither the persistence nor the invasion of cooperative strategies, but the dimensionality of strategies is paramount.

Figure 2 | Strategic flexibility hinders cooperative reciprocity under repeated interactions.

The graphs show the mean surplus per individual per ingroup interaction in the final generations of simulated evolution under $N = 100$ ingroup interactions. Error bars show 95% confidence intervals, which are calculated by bootstrapping over populations and omitted when extremely narrow. Strategies are defined in either two, three, or four dimensions (Fig. 1). Initial conditions are either relatively favourable for the evolution of cooperation (All perfect reciprocity) or relatively unfavourable (All selfish). The life cycle either decouples game play and individual selection (a, c), which reduces individual-level cancellation effects and enhances the potential for relatedness to support cooperation, or it couples them (b, d). The number of migrants per group per generation (m_j) is either relatively low (a, b) or high (c, d). Cooperative reciprocal strategies only evolve or persist when strategies are two-dimensional, and thus ambiguous reciprocity (Fig. 1c) is not possible. When strategies are defined in three or four dimensions, with a relatively high degree of strategic flexibility as a result, cooperative reciprocal strategies sometimes invade, but they never persist.

Moreover, you helpfully pointed out that we could simplify the paper if we were able to confirm that initial conditions are not very interesting. Thank you very much for this excellent comment. In response, we have reduced the number of super-additivity graphs in the main paper from four to two. Figs. 3 – 6 in the original submission have simply become Figs. 3 – 4 in the revision. Because initial conditions do not matter, the revision only shows a selection of super-additivity results when initial conditions consist of unconditional defection. Fig. 3 (see below) shows outcomes when cancellation effects at the group level are as weak as possible, and Fig. 4 (see below) when these cancellation effects are as strong as possible. Figure captions point the reader to the supplement for nearly identical results from different initial conditions.

Figure 3 | Super-additive cooperation when initial conditions are relatively unfavourable and group mixing relatively favourable for the evolution of cooperation. The graphs show the mean surplus per individual per ingroup interaction from the final generations of evolutionary simulations. Error bars show 95% confidence intervals, which are calculated by bootstrapping over populations and omitted when extremely narrow. RI signifies the repeated interactions scenario with $N = 100$ interactions per ingroup pair. GC(1) indicates the group competition scenario as competition outcomes vary in sensitivity to group differences (λ). GC(100) combines the two component mechanisms into the joint scenario, with the bars from left to right corresponding to increasing λ . When the joint scenario is super-additive, the mean surplus is decomposed (Supplementary Information § 3) into the repeated interactions effect (orange), the group competition effect (purple), and the super-additive effect (white). Initial conditions consist of a population of unconditionally selfish individuals (All selfish), and group mixing is as high as possible ($\Xi = 40$), with the former unfavourable and the latter favourable for ingroup cooperation. That said, results are nearly identical when initial conditions consist of a population of perfect reciprocators (Supplementary Figure 160), which reveals that initial conditions actually have no meaningful effect. The life cycle either decouples (a, c) or couples (b, d) game play and individual selection. The number of migrants per group per generation (m_j) is either relatively low (a, b) or high (c, d). Initially, the population consists entirely of selfish individuals, and cooperative strategies must actually invade to become established. Nonetheless, repeated interactions and group competition interact strongly to produce large super-additive gains (a, b, d).

Figure 4 | Super-additive cooperation when initial conditions and group mixing are relatively unfavourable for the evolution of cooperation. The graphs show the mean surplus per individual per ingroup interaction from the final generations of evolutionary simulations. Error bars show 95% confidence intervals, which are calculated by bootstrapping over populations and omitted when extremely narrow. RI signifies the repeated interactions scenario with $N = 100$ interactions per ingroup pair. GC(1) indicates the group competition scenario as competition outcomes vary in sensitivity to group differences (λ). GC(100) combines the two component mechanisms into the joint scenario, with the bars from left to right corresponding to increasing λ . When the joint scenario is super-additive, the mean surplus is decomposed (Supplementary Information § 3) into the repeated interactions effect (orange), the group competition effect (purple), and the super-additive effect (white). Initial conditions consist of a population of unconditionally selfish individuals (All selfish), and group mixing is as low as possible ($\Xi = 0$), both of which are relatively unfavourable for ingroup cooperation. That said, results are nearly identical when initial conditions consist of a population of perfect reciprocators (Supplementary Figure 162), which reveals that initial conditions actually have no meaningful effect. The life cycle either decouples (a, c) or couples (b, d) game play and individual selection. The number of migrants per group per generation (m_j) is either relatively low (a, b) or high (c, d). Cancellation effects at the group level⁸ hinder cooperation in general. Moreover, the initial population consists entirely of selfish individuals, and cooperative strategies must actually invade to become established. Super-additivity is nonetheless common (a, b, d), and in some cases the result of extremely strong positive interactions (d).

Given that group competitions play such a central role in the modelling project, what degree of structure do simulated populations exhibit?

Thank you for encouraging us to say more about population structure in our revision. This is particularly relevant in view of the fact that variation between groups plays such a prominent role in theories of group selection. To tackle this issue in our revision, we conducted a number of new analyses on our simulation data, and we added a new section presenting the results (i.e. § 2.1.19) to the Supplementary Information (updated on github.com/cmefferson).

The idea behind the new analyses is as follows. We first analysed the overall degree of heterogeneity in simulated populations through time, and then we identified how much of the variation is between groups. These analyses show that heterogeneity and population structure are both generally present but quite limited. We refer to this conclusion in the revised version of the main paper in the following way.

Put differently, intergroup competitions function as a kind of equilibrium selection device. Several mechanisms can transform a social dilemma into some other game with multiple equilibria. An aversion to inequitable outcomes³⁵, for example, is a proximate psychological mechanism that can render mutual cooperation an equilibrium in a one-shot prisoner's dilemma³⁶. A psychology prone to internalise social norms and motivate people to punish norm violations can do the same^{29, 37}. Repeated interactions famously support many equilibria³⁸, and in our setting evolutionary dynamics under repeated interactions support two general classes of equilibrium. One class is based on escalating reciprocity, and the other is based on de-escalating reciprocity. The cooperative escalating equilibria generate high payoffs. In the absence of group competitions, however, the fragility of these equilibria dominates evolutionary dynamics, and the most likely outcome in the long run is an uncooperative de-escalating equilibrium.

Group competitions, however, can change the balance of forces by adding a mechanism that favours relatively cooperative groups. The higher payoffs associated with escalation can now dominate the fragility of escalation, with the final outcome a cooperative escalating equilibrium. Importantly, when group competitions shift the balance of forces in this way, the cooperative outcome does not require large differences between groups. Indeed, the differences between groups in our simulated populations are limited³⁹, with the variation in strategies between groups constituting around 4 – 7% of the total variation in the population on average (Supplementary Information § 2.1.19). Any mechanism that increases variation among groups, with special forms of cultural transmission an obvious example¹⁸, would presumably increase the scope for super-additive gains when repeated interactions and group competitions coincide.

pp. 22-23

In short, these new analyses (Supplementary Figures 109 – 118) show results that are highly consistent across our three different scenarios and all of parameter space. The only parameter that has a clear, if small, effect is the migration rate. Our low migration rate leads to limited structure, with approximately 7% of the total strategy variation in the population between groups, and our high migration rate leads to values of 4-5%. For reference, this is less than most of the cultural F_{ST} values in the well-known papers by Bell et al. (2009, *PNAS*) and Muthukrishna et al. (2020, *Psychological Science*). Our result is significant because much of the debate about cultural group selection has centred around how much between-group variation were actually created by genes versus culture in the evolutionary past.

How does our benchmark under repeated interactions compare to other studies that have worked with the binary prisoner's dilemma and large strategy spaces?

In our original submission, we discussed a handful of studies showing that repeated interactions struggle to support cooperation in the binary prisoner's dilemma when the strategy space is large. You rightly pointed out that our results for a continuous version of the game are even more pessimistic about the potential for repeated interactions to support cooperation. You suggested that we could do a better job highlighting these issues, which in turn would help orient readers who may be unfamiliar with the theoretical challenges posed by large strategy spaces. This was a really great suggestion, and we have expanded our discussion of the benchmarks provided by both the binary prisoner's dilemma and our continuous sequential game. The revised main paper reads as follows.

Repeated interactions alone cannot explain the evolution of one-shot cooperation because they cannot explain the evolution of repeated cooperation. Without unjustifiable restrictions on the strategy space, repeated interactions always lead to the evolution of uncooperative forms of reciprocity. A few analogous results hold for the standard prisoner's dilemma in which players simply choose defect or cooperate^{18, 42}. The studies in question expand the strategy space by allowing strategies that condition the current choice on an increasing number of past interactions. Instead of only allowing strategies that condition on the most recent interaction, with tit-for-tat a prominent example, they also allow strategies that condition on the last two interactions, the last three, or many more. Increasing strategic flexibility in this way undermines sustained cooperation^{2, 3, 43}, with even the most favourable conditions rarely generating cooperation rates in excess of 0.5 when repeated interactions act alone⁴³.

One might object, however, that expanding the strategy space in this way requires decision makers to have unreasonably long and detailed memories. This objection does not hold for our models because we only consider strategies that require people to remember a single event. Nonetheless, when cooperation varies continuously, strategic flexibility readily arises in other ways. If we only allow escalating and de-escalating strategies, as with our two-dimensional models, we make the continuous game similar to the standard prisoner's dilemma. Specifically, as escalating or de-escalating reciprocity become common, interacting players tend to converge quickly on full cooperation or full defection respectively. Players do not converge on intermediate levels of cooperation, which minimizes the role of continuous actions by making the continuous game similar to the standard game. Without this restriction, repeated interactions do not reliably support the evolution of cooperation, and our repeated interactions scenario paints an even more dismal picture than studies relying on the standard prisoner's dilemma^{42, 43}. In our case, a simple three-dimensional strategy that conditions on one move only is already enough to destabilize cooperative strategies, with little to no cooperation the final outcome.

pp. 26-27

“Decoupled” is a relative concept, which was not clear in the original submission.

Thank you for raising this point. Indeed, we made this point ourselves in the original supplement, but not in the main paper. Clearly, this was the wrong way around, and we are grateful to you for alerting us to this problem. In particular, we are asking the word “decoupled” to do double duty. On the one hand, it refers to one of two life cycles we implement in our models. In this sense, the term is in fact categorical. On the other hand, however, it refers to the degree to which population structure at the game play stage is separate from population structure at the selection stage. To clarify these issues, we have added the following paragraph to the revised main paper.

Importantly, in terms of the link between game play and individual selection, decoupling is a relative concept. Under the decoupled life cycle, related cooperators who play the social dilemma together might still end up competing against each other at the selection stage. This outcome is possible simply because, even when migration rates are high, some individuals remain in the natal group. Thus, two individuals who play the game together may both stay in the same group and end up competing later to reproduce. The timing of migration does not completely eliminate this possibility because not everyone migrates. Instead, the decoupled life cycle ensures that individuals who play the social dilemma together are less likely to compete against each other than they would be under the coupled life cycle.

p. 13

Better explain in the main paper how strategies actually work.

You were also kind enough and astute enough to see that our description of strategies in the main paper was too terse. Our brief explanation was obscuring the all-important difference in one-shot games between how first movers use their strategies and how second movers use their strategies. Thank you for bringing this source of confusion to our attention. To correct the problem, we have expanded to relevant paragraph in the main paper. The revised version now reads as follows.

A one-shot interaction consists of a single stage game. Repeated interactions consist of multiple stage games, with new endowments for each interaction. An individual's strategy consists of an initial transfer and a response function. The initial transfer specifies how much the individual transfers, if first mover, for the first interaction only. For all subsequent transfers, however many, the response function specifies an individual's current transfer as a function of her partner's most recent transfer. Specifically, the second mover always responds to the first mover's transfer in the same interaction. If interactions are repeated, from the second interaction onward, the first mover responds to the second mover's transfer from the preceding interaction (Supplementary Information § 2.1.3, 2.2, and 2.3).

p. 7

Thank you!

Finally, we'd like to thank you for the time and energy you devoted to a constructive and insightful review of our paper. We were delighted to read your enthusiastic reaction and your extremely useful comments and suggestions, all of which we incorporated into the revised version of the paper. We believe the paper is vastly improved because of your input. We hope you will agree and support publication of the revised version of "Super-additive cooperation" in *Nature*.

Sincerely,

Charles Efferson and Ernst Fehr

References from Main Paper

1. Hagen, E. H. & Hammerstein, P. Game theory and human evolution: A critique of some recent interpretations of experimental games. *Theoretical Population Biology* 69, 339–348 (2006).
2. Zefferman, M. R. Direct reciprocity under uncertainty does not explain one-shot cooperation, but demonstrates the benefits of a norm psychology. *Evolution and Human Behavior* 35, 358–367 (2014).
3. Jagau, S. & van Veelen, M. A general evolutionary framework for the role of intuition and deliberation in cooperation. *Nature Human Behaviour* 1, 1–6 (2017).
4. Choi, J.-K. & Bowles, S. The coevolution of parochial altruism and war. *Science* 318, 636–640 (2007).
5. Richerson, P. et al. Cultural group selection plays an essential role in explaining human cooperation: A sketch of the evidence. *Behavioral and Brain Sciences* 39 (2016).
6. Boyd, R. & Richerson, P. J. Large-scale cooperation in small-scale foraging societies. *Evolutionary Anthropology: Issues, News, and Reviews* (2022).
7. De Dreu, C. K., Fariña, A., Gross, J. & Romano, A. Prosociality as a foundation for intergroup conflict. *Current Opinion in Psychology* 44, 112–116 (2022).
8. Akdeniz, A. & van Veelen, M. The cancellation effect at the group level. *Evolution* (2020).
9. Haselton, M. G., Nettle, D. & Murray, D. R. The evolution of cognitive bias. *The Handbook of Evolutionary Psychology* 1–20 (2015).
10. Balliet, D., Wu, J. & De Dreu, C. K. Ingroup favoritism in cooperation: a meta-analysis. *Psychological Bulletin* 140, 1556 (2014).
11. Yamagishi, T., Jin, N. & Kiyonari, T. Bounded generalized reciprocity: ingroup boasting and ingroup favoritism. *Advances in Group Processes* 16, 161–197 (1999).
12. Haley, K. J. & Fessler, D. M. T. Nobody's watching?: subtle cues affect generosity in an anonymous economic game. *Evolution and Human Behavior* 26, 245–256 (2005).
13. Yamagishi, T. & Mifune, N. Parochial altruism: does it explain modern human group psychology? *Current Opinion in Psychology* 7, 39–43 (2016).
14. Alger, I., Weibull, J. W. & Lehmann, L. Evolution of preferences in structured populations: genes, guns, and culture. *Journal of Economic Theory* 185, 104951 (2020).
15. Handley, C. & Mathew, S. Human large-scale cooperation as a product of competition between cultural groups. *Nature Communications* 11, 1–9 (2020).
16. Wilson, D. S. & Wilson, E. O. Rethinking the theoretical foundation of sociobiology. *The Quarterly Review of Biology* 82, 327–348 (2007).
17. Gross, J. & De Dreu, C. K. The rise and fall of cooperation through reputation and group polarization. *Nature Communications* 10, 1–10 (2019).
18. Henrich, J. Cultural group selection, coevolutionary processes and large-scale cooperation. *Journal of Economic Behavior and Organization* 53, 3–35 (2004).

19. Muthukrishna, M. & Henrich, J. A problem in theory. *Nature Human Behaviour* 3, 221–229 (2019).
20. Akdeniz, A. & van Veelen, M. The evolution of morality and the role of commitment. *Evolutionary Human Sciences* 3 (2021).
21. Henrich, N. & Henrich, J. *Why Humans Cooperate: A Cultural and Evolutionary Explanation* (Oxford: Oxford University Press, 2007).
22. Wahl, L. M. & Nowak, M. A. The continuous prisoner's dilemma: I. linear reactive strategies. *Journal of Theoretical Biology* 200, 307–321 (1999).
23. Le, S. & Boyd, R. Evolutionary dynamics of the continuous iterated prisoner's dilemma. *Journal of Theoretical Biology* 245, 258–267 (2007).
24. Gurven, M. To give and to give not: the behavioral ecology of human food transfers. *Behavioral and Brain Sciences* 27, 543–583 (2004).
25. Kaplan, H. & Gurven, M. The natural history of human food sharing and cooperation: a review and a new multi-individual approach to the negotiation norms. In Gintis, H., Bowles, S., Boyd, R. & Fehr, E. (eds.) *Moral Sentiments and Material Interests: The Foundation of Cooperation in Economic Life*, 75–113 (Cambridge: The MIT Press, 2005).
26. Hrdy, S. B. *Mothers and Others: The Evolutionary Origins of Mutual Understanding* (Cambridge, MA: Harvard University Press, 2011).
27. Axelrod, R. M. *The Evolution of Cooperation* (New York: Basic Books, 1984).
28. Hill, K. R. et al. Co-residence patterns in hunter-gatherer societies show unique human social structure. *Science* 331, 1286–1289 (2011).
29. Henrich, J. & Muthukrishna, M. The origins and psychology of human cooperation. *Annual Review of Psychology* 72, 207–240 (2021).
30. Rusch, H. The evolutionary interplay of intergroup conflict and altruism in humans: a review of parochial altruism theory and prospects for its extension. *Proceedings of the Royal Society B: Biological Sciences* 281, 20141539 (2014).
31. Apicella, C. L., Marlowe, F. W., Fowler, J. H. & Christakis, N. A. Social networks and cooperation in hunter-gatherers. *Nature* 481, 497 (2012).
32. Lehmann, L. & Rousset, F. How life history and demography promote or inhibit the evolution of helping behaviours. *Philosophical Transactions of the Royal Society B: Biological Sciences* 365, 2599–2617 (2010).
33. Taylor, P. D. Altruism in viscous populations: an inclusive fitness model. *Evolutionary Ecology* 6, 352–356 (1992).
34. Kaplan, H., Lancaster, J. & Robson, A. Embodied capital and the evolutionary economics of the human life span. *Population and Development Review* 29, 152–182 (2003).
35. Fehr, E. & Schmidt, K. M. A theory of fairness, competition, and cooperation. *Quarterly Journal of Economics* 114, 817–868 (1999).
36. Bowles, S. *Microeconomics: Behavior, Institutions, and Evolution* (New York: Russell Sage,

2004).

37. Chudek, M. & Henrich, J. Culture–gene coevolution, norm-psychology and the emergence of human prosociality. *Trends in Cognitive Sciences* 15, 218–226 (2011).
38. Gibbons, R. S. *Game Theory for Applied Economists* (Princeton University Press, 1992).
39. Bell, A. V., Richerson, P. J. & McElreath, R. Culture rather than genes provides greater scope for the evolution of large-scale prosociality. *Proceedings of the National Academy of Sciences* **106**, 17671–17674 (2009).
40. Brandts, J. & Charness, G. The strategy versus the direct-response method: a first survey of experimental comparisons. *Experimental Economics* 14, 375–398 (2011).
41. [A note. Not a citation.]
42. Boyd, R. & Lorberbaum, J. P. No pure strategy is evolutionarily stable in the repeated prisoner's dilemma game. *Nature* 327, 58–59 (1987).
43. van Veelen, M., García, J., Rand, D. G. & Nowak, M. A. Direct reciprocity in structured populations. *Proceedings of the National Academy of Sciences* 109, 9929–9934 (2012).
44. Boyd, R. Mistakes allow evolutionary stability in the repeated prisoner's dilemma game. *Journal of Theoretical Biology* 136, 47–56 (1989).
45. Price, G. R. Extension of covariance selection mathematics. *Annals of Human Genetics* 35, 485–490 (1972).
46. Gat, A. *War in Human Civilization* (Oxford: Oxford University Press, 2006).
47. Bowles, S. Did warfare among ancestral hunter-gatherers affect the evolution of human social behaviors? *Science* 324, 1293–1298 (2009).
48. Fry, D. P. & Söderberg, P. Lethal aggression in mobile forager bands and implications for the origins of war. *Science* 341, 270–273 (2013).
49. Romano, A., Sutter, M., Liu, J. H., Yamagishi, T. & Balliet, D. National parochialism is ubiquitous across 42 nations around the world. *Nature Communications* 12, 4456 (2021).
50. Axelrod, R. & Hamilton, W. D. The evolution of cooperation. *Science* 211, 1390–1396 (1981).

Response to Referee #2

We would like to thank you for your constructive and insightful report on our submission, “Super-additive cooperation,” earlier this year. We are delighted that you discovered “*a lot to like in [our] manuscript, including the integration of modelling, simulations, and behavioral experiments with participants from non-Western populations.*” We are also quite happy to know that you found our basic conclusions compelling, specifically that the key message “*makes sense.*”

We would also like to thank you for your excellent feedback, which provided several important suggestions for how to improve the paper. In response to your comments, as well as those of the editor and your fellow referees, we have revised and expanded several key passages in the main paper. We have also dramatically **extended the models themselves by extending the simulations from 50'000 to 150'000 generations** (suggested by Referee #1) and by **allowing for implementation errors** in strategy choices (i.e., choice mistakes, as suggested by Referee #3). These extensions generated important new insights. In particular, they show that **our results are robust to the existence of choice mistakes, and that variation in initial conditions (e.g., whether the simulations start with all subjects unconditionally selfish versus cooperatively reciprocal) also plays no role.** As a result, we believe this project may now constitute one of the most comprehensive modelling projects on the evolution of cooperation to date.

Because of all this outstanding feedback, we believe the revised version of the paper and supplement are much improved over the original submission. We hope you will agree and support publication of the revision in *Nature*. For your convenience, we now summarize the revisions we made in response to your feedback.

Make the abstract less technical and less difficult to understand

We followed your advice and made the abstract accessible for a wider audience. The abstract now makes clear that our paper deals with **evolutionary** explanations of cooperation, i.e., we are interested in the **ultimate**, evolutionary mechanisms that give rise to cooperation. The abstract is now also more explicit about why the findings from our simulations not only undermine repeated interactions as a stand-alone evolutionary explanation, but why they also **undermine the evolutionary explanation of one-shot cooperation** by extending the logic of repeated interactions to one-shot settings.

What is the paper's main aim?

The suggested rewriting of the abstract helped us to make to overall aim of our paper clearer. The abstract now makes it transparent that our findings challenge two prominent **evolutionary** theories of cooperation **in one-shot settings**, namely theory based on repeated interactions and theory based on group competition.

The **repeated-interactions approach** explains the evolution of cooperation in one-shot interactions with two basic ingredients. First, it claims that repeated interactions provide an **evolutionary rationale for cooperation in repeated settings**. Second, it assumes that an evolutionary history of repeated interactions led to a psychological predisposition to **interpret all one-shot interactions as if they are repeated** or have a high chance of being repeated. By combining these two ideas, the repeated-interactions approach provides an evolutionary rationalization for why people cooperate in one-shot settings, and in doing so it has also been used to rationalize the existence of social (cooperative)

preferences. In other words, social preferences are viewed as the **proximate mechanism** through which evolution has implemented cooperation in one-shot settings.

Note that the first claim (i) represents a very prominent, if not dominant, view across the evolutionary sciences, and in this sense, it is the more fundamental of the two claims. **Our modelling results show, however, that repeated interactions alone cannot explain the evolution cooperation in repeated settings. This means that our results challenge both the repeated-interactions account of cooperation in repeated settings and its capacity to explain cooperation in one-shot settings.**

We now make this clear in the revised abstract by adding the following sentence.

This finding challenges repeated interactions as an evolutionary explanation for cooperation in repeated settings. It also undermines evolutionary theories that use repeated interactions in the past to explain one-shot cooperation in the present.

p. 1

In addition, the Discussion section of the revised version of our paper opens with the following direct summary (see page 26):

Repeated interactions alone cannot explain the evolution of one-shot cooperation because they cannot explain the evolution of repeated cooperation.

p. 26

Apart from a thorough challenge to repeated interactions, **our modelling results also challenge group competition** as a **stand-alone** evolutionary explanation for one-shot cooperation among ingroup members. In particular, group competition favours such cooperation only under very special, unlikely conditions. We also mention this now in the revised abstract.

Intergroup competitions also do not reliably support the evolution of cooperation because groups quickly become extremely similar, which leaves little scope for selection between groups.

p. 1

In addition, the Discussion section of the revised version now summarizes these findings as follows:

Intergroup competitions also do not reliably support the evolution of one-shot cooperation, and we have examined key subtleties suggesting the limitations of group competition could be even more serious than previously imagined.

pp. 28 - 29

Finally, our **third key result, and perhaps the most surprising, is that combining the two evolutionary mechanisms – both of which are insufficient when considered in isolation – creates strong synergies. These synergies lead to the evolution of cooperation much more reliably than each of the two mechanisms by themselves.** This result appears particularly striking in view of the enormous hostility between the proponents of the two constituent theories. Indeed, this animosity may have hindered scientific integration in the past. Nonetheless, we show that bringing the two mechanisms together does something special. This is one of our key messages. **Integration leads to outcomes that cannot be reduced to the sum of the underlying parts. In other words, integration leads to outcomes that are “super-additive”.** To make this idea explicit, we have expanded the Introduction to prepare the reader for this key idea.

With our jointly theoretical and experimental approach, we show that neither hypothesis about the evolution of one-shot cooperation actually works. Neither repeated interactions alone nor intergroup competitions alone support ingroup cooperation in a meaningful way, and neither mechanism leads to ingroup and outgroup predictions consistent with the one-shot behaviour observed in Papua New Guinea. Nonetheless, even though the discussion of the two hypotheses often seems to treat them as strict alternatives, they are not. Repeated interactions within groups and competitions between groups can coincide²¹. We also show that combining the two mechanisms generates strong positive interactions. The result is the evolution of cooperative reciprocity with ingroup members, which amplifies cooperation within groups, and uncooperative reciprocity with outgroup members, which erodes cooperation between groups. This mix in which all equilibrium strategies are reciprocal, but not all reciprocal strategies are cooperative, is exactly what we observed among our participants in Papua New Guinea.

p. 5

How generalizable are the empirical results?

This is a fundamental question. Thank you for reminding us to address it in the revision. Also, thank you for pointing us towards a recent and prominently published paper by Romano *et al.* (2021) in *Nature Communications*. We have added this key citation, as well as Baillet *et al.* (2014), to our revision in the final sentence of the paragraph below. Both references include important findings that allow us to say something about the generalizability of our own empirical results.

This super-additive mix produces the generic pattern observed in empirical studies, namely more cooperation with ingroup partners and less cooperation with outgroup partners⁷. More precisely, super-additive outcomes predict the specific nuanced pattern we observed in Papua New Guinea. Although second movers exhibited positive reciprocity with both ingroup and outgroup partners, the differences are telling. With ingroup partners, reciprocity was escalating and thus relatively cooperative, while with outgroup partners it was de-escalating and thus relatively uncooperative. Our joint scenario, and only our joint scenario, predicts these differential forms of reciprocity under a wide variety of conditions. Future empirical research could examine how widely this mix of ingroup-outgroup reciprocity holds across societies, but we already know that ingroup favouritism in social dilemmas is extraordinarily widespread^{10,50}.

p. 31

Could the meta-analytic results¹⁰ in Baillet *et al.* (2014) inform our study?

We are grateful to you for pointing us toward the meta-analysis¹⁰ by Baillet *et al.* (2014). For our purposes, the key insights from this meta-analysis are two-fold. First, they show that on average people are more cooperative with ingroup members. Second, intergroup discrimination is the result of ingroup favouritism rather than outgroup derogation. Our findings from Papua New Guinea are fully compatible with these results and suggest that these behavioural inclinations have deep roots as they occur even in almost stateless societies like those in Papua New Guinea. In this context, our finding that the subjects in Papua New Guinea are still somewhat prosocial to outgroup members by exhibiting de-escalating (“noncooperative”) reciprocity, while they display escalating (“cooperative”) reciprocity towards ingroup members, appears particularly interesting.

Aside from citing Baillet *et al.* (2014) in the passage quoted immediately above, we now cite this important paper in the Introduction to motivate the idea that *relative* selfishness towards outgroup

members does not require an interest in outgroup derogation. Rather, it may simply mean that the ingroup psychology is inactive.

When interacting with someone in an outgroup category, the ancestral psychology should remain relatively idle, and one should behave selfishly. This kind of selfish behaviour does not necessarily require an interest in derogating the outgroup. Rather, the cooperative ingroup psychology may simply be inactive¹⁰.

p. 3

What about proximate mechanisms that make no reference to evolutionary mechanisms?

You asked us why we did not consider explanations such as the existence of social norms, socialization practices, institutions, or social preferences. We agree that these are all plausible **proximate** reasons for human cooperation. In our paper, however, we are interested in the **ultimate** (i.e. **evolutionary**) reasons for human cooperation. In other words, we aim to identify the conditions that lead to behavioural strategies that enable the **evolution** of cooperation. Thanks to your comments, we have now made this point much more transparent in the paper, and as explained above we start doing so in the revised abstract.

Behavioural strategies are, of course, implemented via various potential proximate mechanisms. Regardless, no matter how thorough our understanding of proximate mechanisms, this understanding does not tell us about the conditions that might lead to the evolution of cooperative behavioural strategies and the proximate mechanisms on which they rest. These conditions are the principal motivation for our study.

To keep evolutionary models tractable, we often need to jump straight to the evolution of behavioural strategies without going into the details of the proximate mechanisms through which the strategies are implemented. In our simulations, we follow this tradition in evolutionary modelling.

Were modeling choices (e.g. parameter choices) potentially biased, even if unconsciously, in favor of specific results?

When developing the models, we were not blind to the experimental results. However, the modelling project is extraordinarily comprehensive, and thus we think it's unlikely our choices were biased in any meaningful way. Our models systematically manipulate a large number of mechanisms that are typically treated in isolation. In addition, we set up our simulations with the intention to blanket the parameter space. To illustrate, when modulating how sensitive the outcomes of group competitions are to variation in resources, we consider λ values ranging from 0 to 100. This is extreme variation. $\lambda = 0$ means that outcomes have nothing to do with variation in resources, and thus group selection is not possible. In contrast, $\lambda = 100$ nearly generates a step function such that the group with more resources is almost certain to win, i.e. group selection is almost as strong as possible.

To provide just one more example, when we consider cancellation effects at the group level, we simulate evolution for settings that range from one extreme to the other. When $\Xi = 40$, group-level cancellation effects are as weak as possible, and we examine this setting. At the other extreme, when $\Xi = 0$, group-level cancellation effects are as strong as possible. We also examine this setting, as well as a range of intermediate values.

These examples capture the essence of our approach, and they reveal why we do not think our modelling choices biased our results. In effect, subject to the constraints imposed by limited computing power, our modelling philosophy was to run the gamut for each and every parameter. Your question, however, revealed to us that we should be more explicit about this point, and we'd like to thank you for bringing this issue to our attention. Accordingly, the revision now includes the following expanded motivation and description of the modelling project.

Many details about human social life in the evolutionary past are unknown and likely to remain that way⁶. For example, we know little about how often people moved between ancestral groups, who exactly moved when someone did move, and perhaps most importantly how these characteristics varied across ancestral populations. Extrapolating from contemporary foragers is often the best we can do^{28, 31}. These limitations emphasise the importance of examining a wide range of conditions to identify settings that robustly support cooperation without being acutely sensitive to the details. To develop a set of models that can do this, we systematically manipulate the six model characteristics below [pp. 9 – 15]. Together with our three scenarios [pp. 7 – 9], the overall result is a framework that handles uncertainty about ancestral conditions by considering an unusually comprehensive set of possibilities.

p. 9

Finally, as mentioned in the opening of this letter, we dramatically expanded the modelling project for the revision. In particular, we tripled the length of all simulations. We then added implementation errors and repeated the entire simulation project with these errors in place. The upshot is that, with these new extensions, our modelling project presents an unusually comprehensive analysis of the evolution of cooperation. While we imagine – indeed hope – that additional research will follow, we feel confident that we have made an extreme effort to understand how our model system behaves under an extraordinary range of conditions.

Do our results inform the debate about group selection? Is group selection needed?

Our results are very relevant for the debate about the relevance of group selection for the evolution of cooperation. **We show, in particular, that group selection does not work as a stand-alone theory. In this context, we provide even additional reasons for the insufficiency of group selection as a stand-alone explanation:** First, the timing of life events. Second, cancellation effects at the group level. And third, the inherent uncertainties involved (i.e., the stochastic nature) in the outcomes of conflicts. We discuss these factors now explicitly on pages 28-30 of the paper:

Intergroup competitions also do not reliably support the evolution of one-shot cooperation, and we have examined key subtleties suggesting the limitations of group competition could be even more serious than imagined. In particular, group selection does not necessarily occur just because groups compete; cooperative groups must also win^{18, 45}. We have considered three reasons this may or may not happen. First, the timing of life events can affect the link between a group's productivity and its ability to win intergroup competitions. Under our coupled life cycle, a group with many cooperators produces large gains that remain in the group to help win competitions against other groups. Under our decoupled life cycle, however, migration exports the gains from cooperation before such competitions occur, which attenuates the link between productivity and winning. Migration does not hinder cooperation simply by making groups similar; migration makes groups similar at the worst possible time.

Second, cancellation effects at the group level⁸ undermine cooperation that would otherwise evolve. Intuitively, if a cooperative highly productive group ends up competing against its descendant groups, it competes against other cooperative highly productive groups. Cooperative groups enjoy little relative advantage because they compete against each other, while the uncooperative groups pair off to fight their own battles. Third, even if a productive group competes with an unproductive group, the outcome is not certain. The unproductive group may win, for example because it just happens to have a uniquely talented strategist. We manipulated the probability of such outcomes (λ) as a way of capturing all other forces that undercut the association between group competition and group selection.

We find that many pieces have to come together for group competition to support cooperation as a stand-alone mechanism. As we know, groups must be different from each other for the selection of groups to be meaningful^{18, 45}. Equally critical, the gains from cooperation must stay in the group until group competitions occur, cooperative groups must compete specifically with uncooperative groups, and the outcomes of competitions must be sensitive to the differences between cooperative and uncooperative groups. The existence of so many necessary conditions highlights a fundamental point. Estimating the frequency and lethality of ancestral wars⁴⁶⁻⁴⁸, to take a contentious example, is not by itself particularly conclusive when evaluating the role of group selection. We would also need to know which specific groups fought against each other and who exactly died when wars occurred.

p. 28-30

However, **despite the insufficiency of group selection as a stand-alone evolutionary theory, it is nevertheless needed for explaining the evolution of cooperation.** This follows from the fact that repeated interaction theory is also insufficient as a stand-alone theory, and that **only the combination of both mechanisms facilitates the evolution of cooperation under a wide variety of conditions.** The reason for this is that, when combined, the two evolutionary mechanisms generate strong synergistic effects. We summarize this now on pages 23-24.

Intergroup competitions are essential. Repeated interactions without group competitions lead to the evolution of uncooperative reciprocity (Fig. 5a – 5b), just like group competitions without repeated interactions (Fig. 5c – 5d). When the two mechanisms are combined, however, the result is often super-additive. Outcomes take an entirely different form compared to when the mechanisms operate in isolation. When outcomes are super-additive, ingroup strategies evolve so that cooperative forms of reciprocity prevail (Fig. 5e – 5f). Evolved outgroup strategies, in contrast, consist of uncooperative forms of reciprocity characterised by low initial transfers and de-escalation (Supplementary Information § 2.1.17). The joint scenario is the only scenario that generates this ingroup-outgroup pattern with any meaningful regularity.

p. 23-24

Is the assumption that “people migrate with their earnings” fair to the intergroup competition mechanism?

Precisely because we do not want our conclusions to depend on arbitrary assumptions, we have examined the evolution of cooperation under (i) a coupled life cycle and (ii) a decoupled life cycle.

Under the **coupled life cycle**, the order of events is birth, migration, game play, group competition when relevant, and individual selection within groups. This order means that the **cooperative gains from the social dilemma game stay within the group**. These gains are not dispersed throughout the population via migration because **migration occurs before game play**. As a result, the coupled life cycle favors cooperative groups in intergroup competitions.¹

Under the **decoupled life cycle**, the order of events is birth, game play, migration, group competition when relevant, and finally individual selection within groups. Here, **cooperative gains from the social dilemma game are dispersed throughout the overall population via migration because migration occurs after game play**. This feature of the life cycle weakens the influence of ingroup cooperation on the probability of winning a group conflict.

Because many details about human social life in the evolutionary past are unknown, it is important to study both life cycles. Moreover, **one can make arguments in favour of both life cycles**. For example, if cooperation within groups produces large physical infrastructure projects that cannot be moved to other groups, then the coupled life cycle assumption is closer to the reality. If, in contrast, cooperation mainly involves the production of human capital, then the decoupled life cycle assumption makes more sense because human capital can be easily moved across groups; the individual just has to walk from one group to another.

Thus, taken together, our approach provides a balanced assessment of the different evolutionary approaches precisely because we examine the consequences of an exceptionally wide range of different assumptions.

Thank you.

We'd like to close by thanking you again for your extremely helpful and thought-provoking report on our original submission. We believe that responding to your comments has improved the paper greatly. We hope you will agree and support publication of the revision in *Nature*.

Sincerely,

Charles Efferson and Ernst Fehr

¹ The term “coupled” follows from the fact that under the coupled life cycle game play and individual selection are coupled, while under the decoupled life cycle they are decoupled because migration occurs after game play but before group competition. Under the coupled life cycle, cooperators are more likely to compete with each other at the selection stage, which generates cancellation effects at the individual level. The decoupled life cycle mitigates these cancellation effects at the individual level.

References from Main Paper

1. Hagen, E. H. & Hammerstein, P. Game theory and human evolution: A critique of some recent interpretations of experimental games. *Theoretical Population Biology* 69, 339–348 (2006).
2. Zefferman, M. R. Direct reciprocity under uncertainty does not explain one-shot cooperation, but demonstrates the benefits of a norm psychology. *Evolution and Human Behavior* 35, 358–367 (2014).
3. Jagau, S. & van Veelen, M. A general evolutionary framework for the role of intuition and deliberation in cooperation. *Nature Human Behaviour* 1, 1–6 (2017).
4. Choi, J.-K. & Bowles, S. The coevolution of parochial altruism and war. *Science* 318, 636–640 (2007).
5. Richerson, P. et al. Cultural group selection plays an essential role in explaining human cooperation: A sketch of the evidence. *Behavioral and Brain Sciences* 39 (2016).
6. Boyd, R. & Richerson, P. J. Large-scale cooperation in small-scale foraging societies. *Evolutionary Anthropology: Issues, News, and Reviews* (2022).
7. De Dreu, C. K., Fariña, A., Gross, J. & Romano, A. Prosociality as a foundation for intergroup conflict. *Current Opinion in Psychology* 44, 112–116 (2022).
8. Akdeniz, A. & van Veelen, M. The cancellation effect at the group level. *Evolution* (2020).
9. Haselton, M. G., Nettle, D. & Murray, D. R. The evolution of cognitive bias. *The Handbook of Evolutionary Psychology* 1–20 (2015).
10. Balliet, D., Wu, J. & De Dreu, C. K. Ingroup favoritism in cooperation: a meta-analysis. *Psychological Bulletin* 140, 1556 (2014).
11. Yamagishi, T., Jin, N. & Kiyonari, T. Bounded generalized reciprocity: ingroup boasting and ingroup favoritism. *Advances in Group Processes* 16, 161–197 (1999).
12. Haley, K. J. & Fessler, D. M. T. Nobody's watching?: subtle cues affect generosity in an anonymous economic game. *Evolution and Human Behavior* 26, 245–256 (2005).
13. Yamagishi, T. & Mifune, N. Parochial altruism: does it explain modern human group psychology? *Current Opinion in Psychology* 7, 39–43 (2016).
14. Alger, I., Weibull, J. W. & Lehmann, L. Evolution of preferences in structured populations: genes, guns, and culture. *Journal of Economic Theory* 185, 104951 (2020).
15. Handley, C. & Mathew, S. Human large-scale cooperation as a product of competition between cultural groups. *Nature Communications* 11, 1–9 (2020).
16. Wilson, D. S. & Wilson, E. O. Rethinking the theoretical foundation of sociobiology. *The Quarterly Review of Biology* 82, 327–348 (2007).
17. Gross, J. & De Dreu, C. K. The rise and fall of cooperation through reputation and group polarization. *Nature Communications* 10, 1–10 (2019).
18. Henrich, J. Cultural group selection, coevolutionary processes and large-scale cooperation. *Journal of Economic Behavior and Organization* 53, 3–35 (2004).

19. Muthukrishna, M. & Henrich, J. A problem in theory. *Nature Human Behaviour* 3, 221–229 (2019).
20. Akdeniz, A. & van Veelen, M. The evolution of morality and the role of commitment. *Evolutionary Human Sciences* 3 (2021).
21. Henrich, N. & Henrich, J. *Why Humans Cooperate: A Cultural and Evolutionary Explanation* (Oxford: Oxford University Press, 2007).
22. Wahl, L. M. & Nowak, M. A. The continuous prisoner's dilemma: I. linear reactive strategies. *Journal of Theoretical Biology* 200, 307–321 (1999).
23. Le, S. & Boyd, R. Evolutionary dynamics of the continuous iterated prisoner's dilemma. *Journal of Theoretical Biology* 245, 258–267 (2007).
24. Gurven, M. To give and to give not: the behavioral ecology of human food transfers. *Behavioral and Brain Sciences* 27, 543–583 (2004).
25. Kaplan, H. & Gurven, M. The natural history of human food sharing and cooperation: a review and a new multi-individual approach to the negotiation norms. In Gintis, H., Bowles, S., Boyd, R. & Fehr, E. (eds.) *Moral Sentiments and Material Interests: The Foundation of Cooperation in Economic Life*, 75–113 (Cambridge: The MIT Press, 2005).
26. Hrdy, S. B. *Mothers and Others: The Evolutionary Origins of Mutual Understanding* (Cambridge, MA: Harvard University Press, 2011).
27. Axelrod, R. M. *The Evolution of Cooperation* (New York: Basic Books, 1984).
28. Hill, K. R. et al. Co-residence patterns in hunter-gatherer societies show unique human social structure. *Science* 331, 1286–1289 (2011).
29. Henrich, J. & Muthukrishna, M. The origins and psychology of human cooperation. *Annual Review of Psychology* 72, 207–240 (2021).
30. Rusch, H. The evolutionary interplay of intergroup conflict and altruism in humans: a review of parochial altruism theory and prospects for its extension. *Proceedings of the Royal Society B: Biological Sciences* 281, 20141539 (2014).
31. Apicella, C. L., Marlowe, F. W., Fowler, J. H. & Christakis, N. A. Social networks and cooperation in hunter-gatherers. *Nature* 481, 497 (2012).
32. Lehmann, L. & Rousset, F. How life history and demography promote or inhibit the evolution of helping behaviours. *Philosophical Transactions of the Royal Society B: Biological Sciences* 365, 2599–2617 (2010).
33. Taylor, P. D. Altruism in viscous populations: an inclusive fitness model. *Evolutionary Ecology* 6, 352–356 (1992).
34. Kaplan, H., Lancaster, J. & Robson, A. Embodied capital and the evolutionary economics of the human life span. *Population and Development Review* 29, 152–182 (2003).
35. Fehr, E. & Schmidt, K. M. A theory of fairness, competition, and cooperation. *Quarterly Journal of Economics* 114, 817–868 (1999).
36. Bowles, S. *Microeconomics: Behavior, Institutions, and Evolution* (New York: Russell Sage,

2004).

37. Chudek, M. & Henrich, J. Culture–gene coevolution, norm-psychology and the emergence of human prosociality. *Trends in Cognitive Sciences* 15, 218–226 (2011).
38. Gibbons, R. S. *Game Theory for Applied Economists* (Princeton University Press, 1992).
39. Bell, A. V., Richerson, P. J. & McElreath, R. Culture rather than genes provides greater scope for the evolution of large-scale prosociality. *Proceedings of the National Academy of Sciences* **106**, 17671–17674 (2009).
40. Brandts, J. & Charness, G. The strategy versus the direct-response method: a first survey of experimental comparisons. *Experimental Economics* 14, 375–398 (2011).
41. [A note. Not a citation.]
42. Boyd, R. & Lorberbaum, J. P. No pure strategy is evolutionarily stable in the repeated prisoner's dilemma game. *Nature* 327, 58–59 (1987).
43. van Veelen, M., García, J., Rand, D. G. & Nowak, M. A. Direct reciprocity in structured populations. *Proceedings of the National Academy of Sciences* 109, 9929–9934 (2012).
44. Boyd, R. Mistakes allow evolutionary stability in the repeated prisoner's dilemma game. *Journal of Theoretical Biology* 136, 47–56 (1989).
45. Price, G. R. Extension of covariance selection mathematics. *Annals of Human Genetics* 35, 485–490 (1972).
46. Gat, A. *War in Human Civilization* (Oxford: Oxford University Press, 2006).
47. Bowles, S. Did warfare among ancestral hunter-gatherers affect the evolution of human social behaviors? *Science* 324, 1293–1298 (2009).
48. Fry, D. P. & Söderberg, P. Lethal aggression in mobile forager bands and implications for the origins of war. *Science* 341, 270–273 (2013).
49. Romano, A., Sutter, M., Liu, J. H., Yamagishi, T. & Balliet, D. National parochialism is ubiquitous across 42 nations around the world. *Nature Communications* 12, 4456 (2021).
50. Axelrod, R. & Hamilton, W. D. The evolution of cooperation. *Science* 211, 1390–1396 (1981).

Response to Referee #3

Thank you for your excellent and insightful report on our original submission, “Super-additive cooperation.” We were extremely pleased to read that you are “*thrilled that this study critically examines the robustness of repetition as a solution to cooperation.*” We were equally delighted to learn that you see super-additivity as holding “*key implications for understanding the origin of human cooperation.*”

You offered a number of crucial ideas about how to expand the reach of the paper. In particular, you indicated the work would have even greater significance if we added choice mistakes to the model. As we explain shortly, we have done so and found that, ***if anything, choice mistakes expand the scope for super-additivity.*** You also suggested some ways in which we could revise the paper to make its significance more explicit and accessible than before. We happily incorporated all of these suggestions.

We believe the revision is a vast improvement over the original submission due to the insightful comments and questions from you, your fellow referees, and the editor. We hope you will agree and support publication in *Nature*. For your convenience, we now summarize the revisions we made specifically in response to your comments.

Add choice mistakes to the model.

We are extremely grateful to you for making this suggestion. Aside from your recommendation to add choice mistakes, Referee #1 also encouraged us to run our simulations for much longer than we originally did. Together, these two important recommendations implied a new simulation project at least six times as large as our original project. Given that the original project took roughly three months to run on multiple computers, we were concerned about computing resources.

Nonetheless, with some excellent assistance from the high-performance computing centre at the University of Lausanne, we were able to implement both of these recommendations by distributing the new simulations across as many as 144 computers at any given time. In particular, we increased the simulation time from 50,000 generations per simulated population to 150,000 generations without reducing the total number of simulated populations. We then added choice mistakes based on your comments, and we repeated the entire project, again simulating 150,000 generations per population. This means that, although your comments were mainly focused on choice mistakes for the repeated interactions scenario, we actually implemented them for all three scenarios.

After the new results were in, we made several crucial revisions. First, we had previously argued in the main paper that both dyadic interactions and decision making without mistakes support the evolution of reciprocity. You pointed out that the second point is not true in general. Thank you for raising this issue. In response, we have revised the relevant part of the main paper to mention only dyadic interactions.

Importantly, the weakness of repeated interactions holds even though we limit attention to dyads, which are relatively conducive to the evolution of reciprocity^{6, 18}.

p. 16

Later, when presenting super-additivity results for the joint scenario, we have also revised the main paper to mention choice mistakes for the first time. In particular, we mention briefly that model results are robust to settings when players make mistakes.

Instead, super-additivity and high levels of ingroup cooperation evolve under a wide range of conditions that involve, among other sources of variation, (i) three- and four-dimensional strategy spaces (Supplementary Figures 138 – 166), (ii) high migration rates (e.g. Figs. 3 – 4, panels d; Supplementary Information § 3), (iii) λ values well below the maximum (e.g. Fig. 3b; Supplementary Information § 3), and (iv) the decoupled life cycle (e.g. Fig. 3a; Supplementary Information § 3). Finally, cancellation effects at the group level can be quite detrimental to the evolution of ingroup cooperation. The joint scenario, however, leads to the evolution of ingroup cooperation far more robustly than group competition alone (Supplementary Information § 2.1.18), even when group-level cancellation effects reach their maximum value (e.g. Fig. 4d; Supplementary Information § 3). Finally, extensions of the model incorporate weak selection and individuals who make mistakes by deviating from the transfers their strategies specify. Results from both extensions are extremely similar to the results presented here (Supplementary Information § 4 – 5).

p. 21

Finally, we have expanded the Discussion section of the main paper to address two key points about choice mistakes. These additions draw heavily from your recommendations, and we'd like to thank you for engaging us with these ideas. First, we briefly summarize how and why mistakes are important for repeated play of the traditional prisoner's dilemma with a binary action space. This provides the basic motivation, as you pointed out, for us to consider mistakes when examining repeated play of a sequential prisoner's dilemma with a continuous action space. Second, we summarize what we find after adding mistakes. ***Namely, we find no meaningful effects for three- and four-dimensional models. When strategy spaces are two-dimensional, we find that mistakes slow down the invasion of cooperative strategies under repeated interactions in isolation. This actually increases the importance of super-additive interactions relative to the models without mistakes.*** The revised Discussion section now presents these conclusions as follows.

As an important caveat, the results we present here are based on models without mistakes; individuals transfer exactly the amounts their strategies specify. Theory based on repeated play of the standard prisoner's dilemma suggests this may not be an innocent assumption. Without mistakes, different cooperative strategies can drift in and out of the population because the strategies in question lead to identical choices^{43, 44}. The population eventually drifts towards some mix of cooperative strategies that is vulnerable to invasion by an uncooperative strategy, and cooperation collapses⁴³. With mistakes, however, these same cooperative strategies no longer generate identical choices. Drift accordingly plays a reduced role, and mistakes can stabilise a specific cooperative strategy from among a glut of cooperative strategies⁴⁴.

Because of the potential importance of mistakes, we added mistakes and repeated our entire simulation study to see what role they might play in our models (Supplementary Information § 5). For three-dimensional and four-dimensional strategies, results remain, in effect, exactly the same. In the two-dimensional case, mistakes dramatically slow down the invasion of cooperative strategies relative to otherwise identical situations without mistakes. As a result, over long but finite time scales, mistakes open the door for group competitions to interact positively with repeated interactions, which is exactly what happens (Supplementary Information § 5.3). Mistakes thus expand the range of conditions in which super-additive cooperation appears. Future research could vary the structure of mistakes when actions are continuous to see how robust this conclusion is.

p. 28

Sections 5.1 – 5.2 of the revised supplement (updated at github.com/cmefferson) present extensive results from the three-dimensional and four-dimensional simulations. The conclusion is, in both cases, that model results with mistakes are effectively identical to results from models without mistakes.

Interestingly, however, the two-dimensional case is different over finite time scales. Section 5.3 of the revised supplement explains how (emphasis added for clarity).

When the strategy space is two-dimensional and agents do not make implementation errors, repeated interactions as a stand-alone mechanism are sufficient to support the evolution of cooperative reciprocity (see Fig. 2 of the main paper). This is true regardless of initial conditions. In particular, this means that 150,000 simulated generations are more than enough for escalating strategies to invade and persist when the initial conditions are ancestral in the sense that all agents in the initial generation are unconditionally selfish.

Because repeated interactions are sufficient under these conditions for the evolution of cooperation, super-additivity is not a meaningful idea. If repeated interactions can do all the work alone, then intergroup competition can add nothing. **However, when we add implementation errors, an interesting and subtle difference appears. Specifically, mean cooperation rates from the final interactions between pairs show a large difference in the two-dimensional case that depends on initial conditions.** When initial conditions are favourable for cooperation, cooperation reaches more or less the maximum possible value within 150,000 generations. When initial conditions are unfavourable, however, cooperation levels are much lower (Supplementary Figure 200). An inspection of the dynamics clarifies the mechanism. As in the parallel simulations without implementation errors, escalating strategies invade. The difference is that **invasion is much slower with implementation errors** (Supplementary Figures 201–202) than without.

Consequently, the invasion of escalating strategies is far from complete after 150,000 generations, and thus the potential for intergroup competition to add something now exists. Interestingly, this is exactly what happens. Super-additivity is common with implementation errors in a way that is simply not possible without implementation errors (Supplementary Figures 203–205). Altogether, these results show that **implementation errors actually increase the scope for super-additivity in the two-dimensional case.** More specifically, implementation errors dramatically slow down the invasion of escalating strategies from ancestral initial conditions. Consequently, over time scales that are long but finite, intergroup competition can speed up invasion in a way that effectively creates super-additivity over the time scales in question.

Supplementary Information, pp. 283-284

Under repeated interactions as a stand-alone mechanism, traffic from the uncooperative equilibrium to the cooperative equilibrium has an extremely low probability per generation when choice mistakes are present compared to equivalent situations when choice mistakes are absent. As a result, choice mistakes make super-additive gains over lengthy but finite time scales possible, and such gains are precisely what we observe.

Supplementary Figure 203 | The gains from ingroup cooperation under different scenarios when strategies are two-dimensional and implementation errors occur. The mean surplus per ingroup interaction per agent is shown under the repeated interactions scenario (RI, $N = 100$), the group competition scenario (GC(1), $N = 1$ and $\lambda \in \{0, 10, 25, 100\}$), and the joint scenario (GC(100), $N = 100$ and $\lambda \in \{0, 10, 25, 100\}$). Under the joint scenario, we compare the mean surplus to an additive combination of the mean surplus in the repeated interactions scenario and the relevant group competition scenario (§ 3). If the mean surplus under the joint scenario is greater than this combination, the effect of combining repeated interactions within groups and competition between groups yields a surplus that is super-additive on average. If the mean surplus is less than the additive combination, the combined effect under the joint scenario is sub-additive. Panels differ in terms of whether or not the life cycle couples game play and individual selection and in terms of migration rates (m_j). Ξ controls group-level cancellation effects. Error bars denote 95% confidence intervals based on a bootstrapping algorithm clustered at the population level. We omit these confidence intervals when extremely narrow.

Discuss the larger significance in terms of group competition as a mechanism for selecting an equilibrium.

This is an extremely important recommendation, and we'd like to sincerely thank you for encouraging us to examine this idea more explicitly than in our original submission. Accordingly, we have expanded the relevant section of the main paper to do exactly this. The revised section now reads as follows.

Why do repeated interactions and group competition interact positively? As explained above, under repeated interactions alone, a finite population typically drifts to regions of strategy space that are susceptible to invasion by ambiguous forms of reciprocity, with the collapse of ingroup cooperation the inevitable result. Before this collapse, however, most individuals have cooperative strategies characterised by high initial transfers and escalating reciprocity (Supplementary Information § 1.2.12 and § 2.1.12). In this sense, repeated interactions support a kind of cooperative attractor that exists but is exceedingly fragile in finite populations. In the joint scenario, we augment repeated interactions with intergroup competitions, but these competitions do not create a cooperative equilibrium out of thin air. Rather, they help stabilise a finite population of cooperative escalating reciprocators against the corrosive effects of drift and ambiguous reciprocity.

Put differently, intergroup competitions function as a kind of equilibrium selection device. Several mechanisms can transform a social dilemma into some other game with multiple equilibria. An aversion to inequitable outcomes³⁵, for example, is a proximate psychological mechanism that can render mutual cooperation an equilibrium in a one-shot prisoner's dilemma³⁶. A psychology prone to internalise social norms and motivate people to punish norm violations can do the same^{29,37}. Repeated interactions famously support many equilibria³⁸, and in our setting evolutionary dynamics under repeated interactions support two general classes of equilibrium. One class is based on escalating reciprocity, and the other is based on de-escalating reciprocity. The cooperative escalating equilibria generate high payoffs. In the absence of group competitions, however, the fragility of these equilibria dominates evolutionary dynamics, and the most likely outcome in the long run is an uncooperative de-escalating equilibrium.

Group competitions, however, can change the balance of forces by adding a mechanism that favours relatively cooperative groups. The higher payoffs associated with escalation can now dominate the fragility of escalation, with the final outcome a cooperative escalating equilibrium. Importantly, when group competitions shift the balance of forces in this way, the cooperative outcome does not require large differences between groups. Indeed, the differences between groups in our simulated populations are limited³⁹, with the variation in strategies between groups constituting around 4 – 7% of the total variation in the population on average (Supplementary Information § 2.1.19). Any mechanism that increases variation among groups, with special forms of cultural transmission an obvious example¹⁸, would presumably increase the scope for super-additive gains when repeated interactions and group competitions coincide.

pp. 22-23

Why does de-escalating reciprocity evolve for outgroup interactions instead of unconditional defection?

This is an excellent question. The answer is the following. Both the dynamics of outgroup strategies (Supplementary Information § 2.3.4) and the final distributions over outgroups strategies (Supplementary Information § 2.1.15 – 2.1.17) reveal that drift is playing an outsize role for the evolution of right intercept values. We have revised the main paper to explain this.

When outcomes are super-additive, ingroup strategies evolve so that cooperative forms of reciprocity prevail (Fig. 5e – 5f). Evolved outgroup strategies, in contrast, consist of uncooperative forms of reciprocity characterised by low initial transfers and de-escalation (Supplementary Information § 2.1.17). Interestingly, de-escalation is at least partially due to the fact that selection on the right

intercepts of outgroup response functions should be especially weak in the joint scenario. Outgroup choices contribute little, proportionally speaking, to fitness under the joint scenario, and so selection on outgroup strategies should be weak in general. In particular, when initial transfers and left intercepts evolve to be low, as they do, selection on the right intercepts of response functions should be especially weak. Right intercepts can then drift, which leads to intermediate values and de-escalating reciprocity as a result. Importantly, however, these outgroup strategies are more cooperative than the unconditional defection that represents the upper limit of feasible generosity towards outgroup interactions in many models³⁰, but they are less cooperative than the reciprocal escalation that evolves to manage ingroup interactions in the joint scenario. The joint scenario is the only scenario that generates this ingroup-outgroup pattern with any meaningful regularity.

pp. 23-24

Why does the link to the original question about one-shot cooperation become so weak as the paper progresses?

We really owe you for this comment. We realized that we had indeed, somewhat bizarrely, failed to make a strong connection between our original question on one-shot interactions and our model results for repeated interactions. The basic argument is the following. Because interactions that are actually repeated do not reliably support the evolution of cooperation, we should not expect repeated interactions in the ancestral past to explain one-shot interactions in the present, even if these one-shot interactions are implicitly perceived to be repeated because some ancestral psychology is active. The Discussion section of the revision opens, as before, with the following direct summary.

Repeated interactions alone cannot explain the evolution of one-shot cooperation because they cannot explain the evolution of repeated cooperation.

p. 26

After this, we have revised the Discussion to really drive the point home with the following paragraph. In particular, we now explicitly argue that observing reciprocity empirically does not allow us to infer that repeated interactions by themselves are the evolutionary reason for reciprocity.

Because neither repeated interactions nor intergroup competitions support the evolution of cooperation by themselves, repeated interactions merit just as much infamy as group selection. Repeated interactions may seem more palatable because the effects operate via individual selection^{1, 12}, but this is irrelevant if cooperation is not the result. In this way, the claim that we explain one-shot cooperation with an ancestral psychology based on repeated interactions is not theoretically viable. Empirically, people may be reciprocators who care about their reputations, and this may even be true in anonymous one-shot settings, but repeated interactions do not explain the evolution of such a psychology. Across taxa, in fact, and consistent with the theoretical weaknesses of repeated interactions, empirical examples of conventional reciprocal altruism in animal societies are surprisingly rare⁴⁹.

p. 30

Again, we hope that our paper is making a number of contributions about repeated interactions, but this particular point is an important one. This is why we chose to open the paper with the focus on one-shot cooperation. All of this would be for naught, however, without looping back around to make the connection at the end. Thank you for identifying this weakness in our original submission and pointing the way toward a solution.

We'd like to close by reiterating our gratitude for your insightful and constructive report on our original submission. We were quite happy to learn of your enthusiasm for this project, but we were even happier to expand the project in light of your excellent comments. We think the paper is much improved as a result. We hope you will agree and support publication of the revised version in *Nature*.

Sincerely,

Charles Efferson and Ernst Fehr

References from Main Paper

1. Hagen, E. H. & Hammerstein, P. Game theory and human evolution: A critique of some recent interpretations of experimental games. *Theoretical Population Biology* 69, 339–348 (2006).
2. Zefferman, M. R. Direct reciprocity under uncertainty does not explain one-shot cooperation, but demonstrates the benefits of a norm psychology. *Evolution and Human Behavior* 35, 358–367 (2014).
3. Jagau, S. & van Veelen, M. A general evolutionary framework for the role of intuition and deliberation in cooperation. *Nature Human Behaviour* 1, 1–6 (2017).
4. Choi, J.-K. & Bowles, S. The coevolution of parochial altruism and war. *Science* 318, 636–640 (2007).
5. Richerson, P. et al. Cultural group selection plays an essential role in explaining human cooperation: A sketch of the evidence. *Behavioral and Brain Sciences* 39 (2016).
6. Boyd, R. & Richerson, P. J. Large-scale cooperation in small-scale foraging societies. *Evolutionary Anthropology: Issues, News, and Reviews* (2022).
7. De Dreu, C. K., Fariña, A., Gross, J. & Romano, A. Prosociality as a foundation for intergroup conflict. *Current Opinion in Psychology* 44, 112–116 (2022).
8. Akdeniz, A. & van Veelen, M. The cancellation effect at the group level. *Evolution* (2020).
9. Haselton, M. G., Nettle, D. & Murray, D. R. The evolution of cognitive bias. *The Handbook of Evolutionary Psychology* 1–20 (2015).
10. Balliet, D., Wu, J. & De Dreu, C. K. Ingroup favoritism in cooperation: a meta-analysis. *Psychological Bulletin* 140, 1556 (2014).
11. Yamagishi, T., Jin, N. & Kiyonari, T. Bounded generalized reciprocity: ingroup boasting and ingroup favoritism. *Advances in Group Processes* 16, 161–197 (1999).
12. Haley, K. J. & Fessler, D. M. T. Nobody's watching?: subtle cues affect generosity in an anonymous economic game. *Evolution and Human Behavior* 26, 245–256 (2005).
13. Yamagishi, T. & Mifune, N. Parochial altruism: does it explain modern human group psychology? *Current Opinion in Psychology* 7, 39–43 (2016).
14. Alger, I., Weibull, J. W. & Lehmann, L. Evolution of preferences in structured populations: genes, guns, and culture. *Journal of Economic Theory* 185, 104951 (2020).

15. Handley, C. & Mathew, S. Human large-scale cooperation as a product of competition between cultural groups. *Nature Communications* 11, 1–9 (2020).
16. Wilson, D. S. & Wilson, E. O. Rethinking the theoretical foundation of sociobiology. *The Quarterly Review of Biology* 82, 327–348 (2007).
17. Gross, J. & De Dreu, C. K. The rise and fall of cooperation through reputation and group polarization. *Nature Communications* 10, 1–10 (2019).
18. Henrich, J. Cultural group selection, coevolutionary processes and large-scale cooperation. *Journal of Economic Behavior and Organization* 53, 3–35 (2004).
19. Muthukrishna, M. & Henrich, J. A problem in theory. *Nature Human Behaviour* 3, 221–229 (2019).
20. Akdeniz, A. & van Veelen, M. The evolution of morality and the role of commitment. *Evolutionary Human Sciences* 3 (2021).
21. Henrich, N. & Henrich, J. *Why Humans Cooperate: A Cultural and Evolutionary Explanation* (Oxford: Oxford University Press, 2007).
22. Wahl, L. M. & Nowak, M. A. The continuous prisoner's dilemma: I. linear reactive strategies. *Journal of Theoretical Biology* 200, 307–321 (1999).
23. Le, S. & Boyd, R. Evolutionary dynamics of the continuous iterated prisoner's dilemma. *Journal of Theoretical Biology* 245, 258–267 (2007).
24. Gurven, M. To give and to give not: the behavioral ecology of human food transfers. *Behavioral and Brain Sciences* 27, 543–583 (2004).
25. Kaplan, H. & Gurven, M. The natural history of human food sharing and cooperation: a review and a new multi-individual approach to the negotiation norms. In Gintis, H., Bowles, S., Boyd, R. & Fehr, E. (eds.) *Moral Sentiments and Material Interests: The Foundation of Cooperation in Economic Life*, 75–113 (Cambridge: The MIT Press, 2005).
26. Hrdy, S. B. *Mothers and Others: The Evolutionary Origins of Mutual Understanding* (Cambridge, MA: Harvard University Press, 2011).
27. Axelrod, R. M. *The Evolution of Cooperation* (New York: Basic Books, 1984).
28. Hill, K. R. et al. Co-residence patterns in hunter-gatherer societies show unique human social structure. *Science* 331, 1286–1289 (2011).
29. Henrich, J. & Muthukrishna, M. The origins and psychology of human cooperation. *Annual Review of Psychology* 72, 207–240 (2021).
30. Rusch, H. The evolutionary interplay of intergroup conflict and altruism in humans: a review of parochial altruism theory and prospects for its extension. *Proceedings of the Royal Society B: Biological Sciences* 281, 20141539 (2014).
31. Apicella, C. L., Marlowe, F. W., Fowler, J. H. & Christakis, N. A. Social networks and cooperation in hunter-gatherers. *Nature* 481, 497 (2012).
32. Lehmann, L. & Rousset, F. How life history and demography promote or inhibit the evolution of helping behaviours. *Philosophical Transactions of the Royal Society B: Biological Sciences*

365, 2599–2617 (2010).

33. Taylor, P. D. Altruism in viscous populations: an inclusive fitness model. *Evolutionary Ecology* 6, 352–356 (1992).

34. Kaplan, H., Lancaster, J. & Robson, A. Embodied capital and the evolutionary economics of the human life span. *Population and Development Review* 29, 152–182 (2003).

35. Fehr, E. & Schmidt, K. M. A theory of fairness, competition, and cooperation. *Quarterly Journal of Economics* 114, 817–868 (1999).

36. Bowles, S. *Microeconomics: Behavior, Institutions, and Evolution* (New York: Russell Sage, 2004).

37. Chudek, M. & Henrich, J. Culture–gene coevolution, norm-psychology and the emergence of human prosociality. *Trends in Cognitive Sciences* 15, 218–226 (2011).

38. Gibbons, R. S. *Game Theory for Applied Economists* (Princeton University Press, 1992).

39. Bell, A. V., Richerson, P. J. & McElreath, R. Culture rather than genes provides greater scope for the evolution of large-scale prosociality. *Proceedings of the National Academy of Sciences* 106, 17671–17674 (2009).

40. Brandts, J. & Charness, G. The strategy versus the direct-response method: a first survey of experimental comparisons. *Experimental Economics* 14, 375–398 (2011).

41. [A note. Not a citation.]

42. Boyd, R. & Lorberbaum, J. P. No pure strategy is evolutionarily stable in the repeated prisoner's dilemma game. *Nature* 327, 58–59 (1987).

43. van Veelen, M., García, J., Rand, D. G. & Nowak, M. A. Direct reciprocity in structured populations. *Proceedings of the National Academy of Sciences* 109, 9929–9934 (2012).

44. Boyd, R. Mistakes allow evolutionary stability in the repeated prisoner's dilemma game. *Journal of Theoretical Biology* 136, 47–56 (1989).

45. Price, G. R. Extension of covariance selection mathematics. *Annals of Human Genetics* 35, 485–490 (1972).

46. Gat, A. *War in Human Civilization* (Oxford: Oxford University Press, 2006).

47. Bowles, S. Did warfare among ancestral hunter-gatherers affect the evolution of human social behaviors? *Science* 324, 1293–1298 (2009).

48. Fry, D. P. & Söderberg, P. Lethal aggression in mobile forager bands and implications for the origins of war. *Science* 341, 270–273 (2013).

49. Romano, A., Sutter, M., Liu, J. H., Yamagishi, T. & Balliet, D. National parochialism is ubiquitous across 42 nations around the world. *Nature Communications* 12, 4456 (2021).

50. Axelrod, R. & Hamilton, W. D. The evolution of cooperation. *Science* 211, 1390–1396 (1981).

Reviewer Reports on the First Revision:

Referees' comments:

Referee #1 (Remarks to the Author):

Super-additive cooperation – Revision 1

By Charles Efferson, Helen Bernhard, Urs Fischbacher, & Ernst Fehr

I am very happy with the revision. The biggest reward in the review process is not the nice words in the replies, but the fact that the authors take the content of the remarks seriously and engage with what is in the report. That regularly does not happen, and therefore it is much, much appreciated. I am particularly happy that the authors reran the simulations, ran them longer, and found that the initial state of the population did indeed not matter.¹ I was already enthusiastic about the paper, and now I am even more positive.

I do have some more detailed comments. These are just suggestions.

Abstract

I totally understand that the format requires the authors to squeeze information and explanation into a few sentences, at the risk of being (marginally) inaccurate. This is from the abstract:

Intergroup competitions also do not reliably support the evolution of cooperation because groups quickly become extremely similar, which leaves little scope for selection between groups. Moreover, even if groups vary, cooperative groups may often lose competitions, which weakens the link between group competition and group selection⁸

This describes the cancellation effect at the group level, which I appreciate, but I would say that groups becoming similar is not exactly the point, and partly orthogonal to the effect that the authors try to describe. It is not so much groups overall becoming similar that matters, because that can happen with or without the cancellation effect at the group level. Instead, what matters it that neighbouring groups become *more* similar than randomly chosen pairs of groups in the population of groups. Also, I am not 100% sure what the second sentence tries to convey. If this is there to make sure the reader understands that mentioning in the abstract. If it is there for another reason, then that needs clarification.

¹ Rerunning equally many runs, just longer, was not necessary; the authors could have saved a lot on computer time and capacity by choosing far fewer runs, and let them run just a little longer than they have now. But that is water under the bridge.

Introduction

Of course I totally understand that the authors are not erring on the side of underselling the importance of the interaction between repetition and population structure, but I have some small reservations when the authors write, for instance:

Only the joint influence of the two mechanisms can provide a sufficient explanation for the evolution of one-shot cooperation

The “only” is a bit more negative about alternatives than I would be, because I think that there are other, complementary candidates, such as commitment. It is also a tiny bit more positive about the super-additive interaction than I would be, in the sense that this still needs additional work to explain why humans cooperate so much more than other primates. Other primates have similar repetition rates and population structures, and for them the same super-additivity could also kick in – while it seems that it has not. All of this is of course within the acceptable margin of enthusiasm about the finding.

Repeated interactions and intergroup competition in all combinations

The game is a theoretical version of the social dilemma we used in Papua New Guinea, and so both our models and experiment rest on the same stage game.

Not sure if the “and” and the “so” should both be there.

After game play, paired groups compete with each other with relatively low probabilities (Supplementary Information § 2.1.7) that decrease as the groups become more similar (Supplementary Information § 2.1.5). This approach reflects the idea that paired groups assess each other, and they both avoid competing when they have trouble identifying the likely winner.

Not sure if this detail is essential for the outcome, and therefore I could imagine that a more minimalistic setup would have worked just as well in order to illustrate the effect that the authors find. I understand though that at this stage, it is not worth changing this.

A framework for examining a wide range of conditions

... when someone did move, and perhaps most importantly how these characteristics varied across ancestral populations.

Not sure if commas around “perhaps most importantly” would be better or worse.

Game play and individual selection are decoupled.

In the previous report, I complained a little about “coupled” and “decoupled” being relative, not absolute. I am happy that a remark is added to the main text to emphasize this, and I am happy that the authors pointed out this was already there in the SI. I now would like to add (and sorry for not doing this before) that I am not sure if the coupling/decoupling of game play and individual selection is a 100% match with what the change of order does. I would think that migrating after game play (rather than before) both makes individual selection happen less between individuals that just played the game with each other, and it makes your fellow group members, and yourself, profit less in the between-group competition from having cooperated in the game. The change in order therefore moves more than just one part, I would think, unless I am overlooking some important detail.

Point 6 / Figure 2

I interpret this as the afterglow of previously having found that the seeding of the population does matter for the outcome. Given that it now turns out not to matter, just a remark would suffice in the main text, I would think, and the elaborate description at point 6 as well as Figure 2 are superfluous.

Super-additive cooperation when combining repeated interactions and group competition

At multiple places in the paper, for instance on page 23, the authors use “*group competitions*”. Now my English is limited, so it’s probably caused by my limitations, but the plural feels a bit odd to me. For a sentence like

Group competitions, however, can change the balance of forces by adding a mechanism that favours relatively cooperative groups.

I would think that just writing

Group competition, however, can change the balance of forces by adding a mechanism that favours relatively cooperative groups.

would be a perfectly acceptable alternative – unless the authors want to use a word that sets between-group competition apart from group selection in general more emphatically. But then again, I defer to the native speakers here, and I understand that in other instances, this is more efficient than writing “instances of between-group competition”.

Put differently ... coincide

I think the authors may have over-interpreted my curiosity about the heterogeneity and “traffic” between different equilibria a little, but that does not have to be bad. I was just technically curious about whether the population transitions between different (more or less) stable uniform population states, or if there is more constant heterogeneity.

Overall

I am at least as positive as I was earlier, and I hope the remarks above are useful input to get the message across even better.

Referee #2 (Remarks to the Author):

Let me begin by thanking the authors for their thorough and constructive dealings with the earlier round of feedback, and making important revisions to their paper that, in many cases, clarified their work and contributions. The paper improved as a result, and the core message is clearer now.

My reading of the revision raised several (new) issues and questions. I suspect much of these can be dealt with in a revision.

Abstract

Despite improvement, especially the spelled-out focus on evolutionary theory, some areas remain difficult. Some suggestions:

(1) Perhaps more explicitly state that 'we show *with evolutionary agent-based simulations*...' (or something similar).

(2) Without having read the paper, and even then (see below), it is unclear what 'ambiguous reciprocity' means and why one should care. It would be helpful if this were defined up-front already here.

(3) [and this pertains to the Main Text also, see below]. Ambiguous reciprocity tends to disappear from the abstract when effects of group competition are discussed, begging the question whether the concept is needed here at all.

Main Text

The opening paragraphs are much clearer now and provide a strong entry into the paper. Thank you!

On the top of page 6, [...only the influence ... one-shot interaction]. Here I would have liked to see a bit more before having to dive into the model. The material on p.22-23 actually contains much of what could be summarized here too. In a way, the remainder of the paper then serves to provide the details on methods and results needed to understand and appreciate these general take-homes. By slight reformatting, there then also is the opportunity to provide more technical info relevant to follow the simulations performed (also see my final comment here below).

Section Repeated interactions ...

p. 6: It might be helpful to specify 'any amount' as operationalized in the simulations

p. 7: what is meant with 'individuals only have *ingroup* strategies'? Also, the next sentence on predictions for the experiment are unclear, and perhaps at this point may not be needed (somehow it

may suggest that it is assumed that within groups there is higher likelihood of repeated interactions than between groups, but this then requires some introduction / discussion).

p. 8: the section of (modelling) group competition needs work in my view. The sentence “After game play, paired groups compete with each other...” is confusing (a.o. because the reader may think individuals played another game and I had to go to the SI to find out this is not the case). In the modelling, as far as I understand, after game play, groups are compared on the basis of group welfare earned from game play, and the wealthier group lives and replaces the poorer group. Cooperation is thus defined as a behavioural strategy but competition is not (perhaps describing it as a winner-takes-all contest with its success function scaled on (relative) endowments of antagonist groups).

What is puzzling is the next sentence: ...with relatively low probabilities that decrease as groups become more similar. This could be read as the likelihood that groups compete is the endogenous result of how similar groups were in their cooperation during game play? If true, the scenarios are not independent and at least one wants to see the prevalence of competition under the various strategies. It is possible that two completely uncooperative groups hence do not compete, like two completely cooperative groups do not compete. Only asymmetrical cases compete with non-zero probability. If this is correct, what are the implications then for the evolution of cooperation strategies?

I can imagine that analytically and for simulations it doesn't matter whether the likelihood of competition is conditioned on welfare (differences), or decided by coinflip in case of a (welfare) tie. Some commentary and justification is needed to help the reader (at least this one) understand the dynamics here. Can we exclude that super-additivity is the result of one mechanism (cooperation in repeated play) conditioning the other mechanism (presence/likelihood of competition)?

p. 10: As noted, I struggled with the ambiguous reciprocity strategy. First, it isn't clear whether the ambiguous strategy discussed here is the only one from 3D that is entered into the simulation, or just one of several (and if so, please specify, including the mixture in the simulations). Second, I had a hard time understanding the functionality of this (particular) ambiguous reciprocity strategy – being generous with a stingy partner and stingy with a generous one. Why would such strategy be played? As the authors remark later in the paper, this form of ambiguous reciprocity may not be observed empirically and indeed, the strategy also doesn't seem to emerge in the reported experiment, which revealed generous (with in-group) and stingy reciprocation (with out-group) only. That ‘humans may have the social cognition capacities to implement such strategies’ felt like a rather weak justification for including it.

In the end, the precise form of 3D and 4D strategies seems to be less relevant than that they can be played in theory, once action space is scaled rather than dichotomous. As soon as ambiguous strategies are allowed, cooperation cannot evolve (p. 16ff). I take this as applying to *any* form 3D or 4D strategy

can take. If I understand correctly, it would be helpful when this is specified earlier on in the paper – when in populations (some) agents can invest other than as matchers (perfect reciprocation), being stingy (always defect), or being generous (always cooperate), cooperation cannot evolve. If, however, this is ‘limited’ to the ambiguous reciprocation strategy as defined and understood here, this needs to be specified and the reader would benefit from knowing about the viability/functionality of such strategies existing in the first place.

p. 14: The description of competition and how it occurs doesn’t seem to match one-on-one with the description on p. 8 on when and how competition emerges (i.e., groups are paired and have a competition with positive probability – but the probability depends on the comparative welfare from game-play right? Please check and update if necessary.

One final thought: Sometimes relevant descriptive information is lacking or hard to find. Initial population structures/mixtures, for example, are not always clear or specified. More methods detail here and there would be helpful (perhaps in a separate section towards the end of the paper?).

Referee #3 (Remarks to the Author):

The manuscript has substantively improved with the revisions. The write-up is much clearer and conveys the logic and broader implications of the paper more effectively. It is good that the authors were able to expand the analyses to include choice mistakes, even though it was an intensive computational operation. The finding's robustness to the addition errors (and to initial conditions) is very compelling, and suggests that the result stems from what is likely to be a fundamental feature of the evolution of reciprocal cooperation.

The following two suggestions should be minor fixes:

1. The presentation of information in the last paragraph on page 7, starting with "The repeated interactions scenario consists of models of populations sub-divided into 40 groups of 24 individuals..." can be improved. The paragraph seems to do two things --- introduce specifically the "repeated interactions scenario" of the model, while also bringing up how the model results generally will map on to inform the predictions of the experimental conditions. Having the latter information thrown into this paragraph is confusing. Rather, consider including a paragraph (after describing both the repeated interactions and group competition model scenarios) that delineates and justifies the mapping from model condition to experimental conditions. The connection between these is one of the strengths of

the paper, and I'm afraid the reader still has to do some mental gymnastics to make the connection between the model and the experiment.

2. The Supplemental Information figures 69-106 are currently lacking an explanation for what the axes and bins represent, and what the dark colors in the bins are showing. I finally found an explanation of this figure type in the caption of figure 107, which was indeed helpful...but it was 37 figures too late! Additionally, while the explanation in figure 107 was a good start, I think providing more guidance to the reader on how to interpret this class of graphs will be helpful.

In summary, this is an impressive amount of work that has resulted in a meaningful contribution to understanding how reciprocity-based cooperation may have evolved in humans. The part that I find most exciting is that population structure conducive to between-group competition is essential for reciprocal cooperation to evolve. It provides a plausible platform for understanding the discrepancy in levels of reciprocity between humans and other animals. Several authors have alluded to cultural transmission generating population structure uniquely in humans, and how competition between cultural groups can serve as an equilibrium selection mechanism on norms and institutions. This paper opens the door for additionally thinking about how culture-based population structure could have shaped the genetic evolution of the psychology for reciprocal cooperation.

Author Rebuttals to First Revision:

Response to Referee #1

Thank you again for your ongoing contribution as a referee for our paper. We are glad that you are “very happy” with the first revision. We are especially pleased that, although you were “already enthusiastic about the paper”, you are now “even more positive”.

You had a few remaining comments on our last revision. In general, these comments implied minor rewrites to help with clarity and precision. Some of them did not imply the need for any changes at all. In any case, your comments, as before, were helpful, insightful, and constructive. We have revised the paper accordingly, and we detail the changes below. We feel the paper is much improved due to your commitment to a thoughtful exchange between referee and authors. We hope you agree and support publication in *Nature*.

Clarify cancellation effects at the group level in the abstract, in particular the importance of neighbouring groups being “more similar than randomly chosen pairs of groups”.

You aptly argued that cancellation effects at the group level are not really a matter of groups becoming similar in general, but rather a matter of similar groups being especially likely to compete against each other. We completely agree with this point, and your comment helped us to realise that the relevant sentences in our abstract were unclear.

The structure we actually had in mind is the following. One reason that group competition does not support ingroup cooperation by itself is that groups tend to be similar under reasonable migration rates. A second reason is that the link between group competition and group selection is weak, which can be true for several reasons. We view cancellation effects at the group level as falling into this second category. When cancellation effects are high, cooperative groups are disproportionately likely to compete against each other. Thus, cooperative groups lose competitions with a relatively high probability, which weakens group selection all else equal. This is not really about the variation between groups so much as what happens given the variation that does exist. We did not communicate this effectively in our last abstract, and we have revised accordingly.

Intergroup competition also does not reliably support the evolution of cooperation because groups quickly become extremely similar, which leaves little scope for selection between groups. Moreover, even if groups vary, group competitions may generate little group selection for multiple reasons. Cooperative groups, for example, may tend to compete against each other⁸.

p. 1

When explaining the positive interaction between repeated interactions and group competition, do not say that “only” this combination works because doing so is a bit too “negative about alternatives”.

Thank you for encouraging us to be more even-handed in an entirely appropriate way. In addition to the sentence you specifically highlighted, we have actually removed this kind of language throughout the paper. As one example, the final sentences of the revised abstract now read,

Results from a one-shot behavioural experiment in Papua New Guinea fit exactly this pattern. They thus suggest neither an evolutionary history of repeated interactions without group competition nor a

history of group competition without repeated interactions. Our results rather suggest social motives that evolved under the joint influence of both mechanisms.

p. 2

We don't need both the "and" and the "so".

With respect to the sentence, "The game is a theoretical version of the social dilemma we used in Papua New Guinea, and so both our models and experiment rest on the same stage game," you suggested that we don't need both the "and" and the "so". We cut the "so":

The game is a theoretical version of the social dilemma we used in Papua New Guinea, and both our models and experiment rest on the same stage game.

p. 6

A more "minimalistic setup" controlling the occurrence of group competitions would probably work just as well as what we chose.

We completely agree with this point. Indeed, imagine a more minimalistic setting in which group competition occurs according to some fixed probability. In this case, groups that are relatively similar would be more likely to compete than they are under our set-up. However, one could compensate for this by simply choosing a slightly higher λ value to yield exactly the same degree of group selection. For this reason, we share your opinion that alternative approaches would work, and we strongly believe that the exact choice of protocol is not crucial.

That said, the advantage of the protocol we used is that it's both consistent with past modelling efforts, especially Choi and Bowles (2007)⁴, and with ethnographic accounts. Christopher Boehm's book on blood feuds in Montenegro²⁹, in particular, describes a process strikingly similar to the protocol we implemented in our simulation models. To highlight these points, we have revised the paper to note these similarities.

After game play, we model the occurrence of group competitions by assuming that paired groups compete against each other with relatively low probabilities (Supplementary Information § 2.1.7) that decrease as the groups become more similar (Supplementary Information § 2.1.5). Although we do not explicitly model decision making at the group level, our approach reflects the idea that paired groups assess each other and avoid competing when they have trouble identifying the likely winner, which is consistent with both past modelling work and ethnographic evidence^{4,29}.

pp. 8-9

Variation in the life cycles actually has two effects; it "moves more than just one part".

In response to our description of cancellation effects at the individual level, you argue that varying the life cycles as we do actually has two effects. First, it modulates cancellation effects at the individual level, which is why we chose to vary the life cycles in the first place. Second, it modulates the extent to which group competition translates into group selection. As you astutely point out, "[the decoupled life cycle] makes your fellow group members, and yourself, profit less in the between-group competition from having cooperated in the game."

We could not agree more. Although we did not anticipate this result, the effect you describe holds unambiguously, and we discuss it at length later in the paper. We do so for the first time when detailing the “limits of group competition”.

Recall that under repeated interactions the decoupled life cycle is relatively favourable for cooperation because it reduces cancellation effects at the individual level (Supplementary Figures 15 - 16). We find exactly the opposite pattern for the group competition scenario. The decoupled life cycle is unfavourable for cooperation because it separates, relative to the coupled life cycle, a group's productivity from the group's ability to win intergroup competitions. Specifically, the decoupled sequence includes game play, migration, and then intergroup competition. During game play, a group with many cooperative individuals enjoys large gains because many group members cooperate. Immediately after game play, however, group members migrate, and they carry the gains from cooperation with them. This idea is relevant, for example, in situations where the beneficiaries of cooperation accumulate embodied capital in the form of knowledge, skills, health, and physical strength³⁵. More broadly, when individuals carry the benefits of cooperation with them, any movement of individuals after game play but before group competition redistributes resources across groups in a way that must attenuate, all else equal, the bite of intergroup competition as a mechanism. The effect is weak if the migration rate is low and strong if high.

pp. 20-21

We elaborate further in the Discussion.

Intergroup competition also does not reliably support the evolution of one-shot cooperation, and we have examined key subtleties suggesting the limitations of group competition could be even more serious than imagined. In particular, group selection does not necessarily occur just because groups compete; cooperative groups must also win^{18,46}. We have considered three reasons this may or may not happen. First, the timing of life events can affect the link between a group's productivity and its ability to win intergroup competitions. Under our coupled life cycle, a group with many cooperators produces large gains that remain in the group to help win competitions against other groups. Under our decoupled life cycle, however, migration exports the gains from cooperation before such competitions occur, which attenuates the link between productivity and winning. Migration does not hinder cooperation simply by making groups similar; migration makes groups similar at the worst possible time.

pp. 30-31

Given that initial conditions do not matter, would it suffice to simply say this briefly in lieu of including Point 6 and Fig. 2?

Thanks to your previous comments, we established in the last revision that initial conditions do not affect long-run outcomes. Because of this, we were able to cut some of the figures that appeared in our initial submission.

However, we have continued to explain that we did, in fact, vary initial conditions. We have also retained Fig. 2, which shows results for the repeated interactions scenario under different initial conditions. In an ideal world, we would agree with you. Saying far less about this would suffice, and we could cut Fig. 2. However, given the contentious nature of research on the evolution of cooperation, and given the widespread faith in repeated interactions among evolutionary researchers, we suspect that we really need Fig. 2.

Fig. 2 answers two fundamental questions. Do cooperative strategies resist invasion when common, and do they invade when rare? The figure shows that the answer to both questions is “yes” when the strategy space is two-dimensional and “no” when three- and four-dimensional. We worry that, given the commitment to repeated interactions among many researchers, sceptical readers will dismiss this important result unless we show it to them . . . plain, clear, and unequivocal. For this reason, we would like to retain Fig. 2.

Intergroup/group “competition” versus intergroup/group “competitions”

Thank you for sharing the fact that you find “intergroup competitions” strange in some instances, and that you think the singular form would be perfectly acceptable. In light of your comment, we thought about this point quite a bit and eventually concluded the following.

We think both the singular and plural work, but they do have a different feel. The singular form invites the reader to think more about intergroup competition as an abstract or generic mechanism, while the plural form invites the reader to think about the competitions themselves.¹ We actually want to evoke both meanings at various points in the paper. Nonetheless, we agree that the last revision included instances where we meant the abstract mechanism, but we used the plural form. For this reason, we have gone through the entire paper and replaced several plural forms (intergroup competitions, group competitions) with their singular equivalents (intergroup competition, group competition).

Thank you!

We’d like to thank you again for the time and energy you devoted to this review process. The paper is really much better, in particular, because of the extensions and additional simulations you suggested. We also hope you will support publication of “Super-additive cooperation” in *Nature*.

Sincerely,

Charles Efferson and Ernst Fehr

References from Main Paper

1. Hagen, E. H. & Hammerstein, P. Game theory and human evolution: A critique of some recent interpretations of experimental games. *Theoretical Population Biology* 69, 339–348 (2006).
2. Zefferman, M. R. Direct reciprocity under uncertainty does not explain one-shot cooperation, but demonstrates the benefits of a norm psychology. *Evolution and Human Behavior* 35, 358–367 (2014).
3. Jagau, S. & van Veelen, M. A general evolutionary framework for the role of intuition and deliberation in cooperation. *Nature Human Behaviour* 1, 1–6 (2017).

¹ In addition, given that “repeated interactions” is plural, the use of “intergroup competitions” is sometimes useful for constructing a parallel compound subject, e.g. “Repeated interactions and intergroup competitions both fail to support ingroup cooperation by themselves.”

4. Choi, J.-K. & Bowles, S. The coevolution of parochial altruism and war. *Science* 318, 636–640 (2007).
5. Richerson, P. et al. Cultural group selection plays an essential role in explaining human cooperation: A sketch of the evidence. *Behavioral and Brain Sciences* 39 (2016).
6. Boyd, R. & Richerson, P. J. Large-scale cooperation in small-scale foraging societies. *Evolutionary Anthropology: Issues, News, and Reviews* (2022).
7. De Dreu, C. K., Fariña, A., Gross, J. & Romano, A. Prosociality as a foundation for intergroup conflict. *Current Opinion in Psychology* 44, 112–116 (2022).
8. Akdeniz, A. & van Veelen, M. The cancellation effect at the group level. *Evolution* (2020).
9. Haselton, M. G., Nettle, D. & Murray, D. R. The evolution of cognitive bias. *The Handbook of Evolutionary Psychology* 1–20 (2015).
10. Balliet, D., Wu, J. & De Dreu, C. K. Ingroup favoritism in cooperation: a meta-analysis. *Psychological Bulletin* 140, 1556 (2014).
11. Yamagishi, T., Jin, N. & Kiyonari, T. Bounded generalized reciprocity: ingroup boasting and ingroup favoritism. *Advances in Group Processes* 16, 161–197 (1999).
12. Haley, K. J. & Fessler, D. M. T. Nobody's watching?: subtle cues affect generosity in an anonymous economic game. *Evolution and Human Behavior* 26, 245–256 (2005).
13. Yamagishi, T. & Mifune, N. Parochial altruism: does it explain modern human group psychology? *Current Opinion in Psychology* 7, 39–43 (2016).
14. Alger, I., Weibull, J. W. & Lehmann, L. Evolution of preferences in structured populations: genes, guns, and culture. *Journal of Economic Theory* 185, 104951 (2020).
15. Handley, C. & Mathew, S. Human large-scale cooperation as a product of competition between cultural groups. *Nature Communications* 11, 1–9 (2020).
16. Wilson, D. S. & Wilson, E. O. Rethinking the theoretical foundation of sociobiology. *The Quarterly Review of Biology* 82, 327–348 (2007).
17. Gross, J. & De Dreu, C. K. The rise and fall of cooperation through reputation and group polarization. *Nature Communications* 10, 1–10 (2019).
18. Henrich, J. Cultural group selection, coevolutionary processes and large-scale cooperation. *Journal of Economic Behavior and Organization* 53, 3–35 (2004).
19. Muthukrishna, M. & Henrich, J. A problem in theory. *Nature Human Behaviour* 3, 221–229 (2019).
20. Akdeniz, A. & van Veelen, M. The evolution of morality and the role of commitment. *Evolutionary Human Sciences* 3 (2021).

21. Henrich, N. & Henrich, J. *Why Humans Cooperate: A Cultural and Evolutionary Explanation* (Oxford: Oxford University Press, 2007).
22. Wahl, L. M. & Nowak, M. A. The continuous prisoner's dilemma: I. linear reactive strategies. *Journal of Theoretical Biology* 200, 307–321 (1999).
23. Le, S. & Boyd, R. Evolutionary dynamics of the continuous iterated prisoner's dilemma. *Journal of Theoretical Biology* 245, 258–267 (2007).
24. Gurven, M. To give and to give not: the behavioral ecology of human food transfers. *Behavioral and Brain Sciences* 27, 543–583 (2004).
25. Kaplan, H. & Gurven, M. The natural history of human food sharing and cooperation: a review and a new multi-individual approach to the negotiation norms. In Gintis, H., Bowles, S., Boyd, R. & Fehr, E. (eds.) *Moral Sentiments and Material Interests: The Foundation of Cooperation in Economic Life*, 75–113 (Cambridge: The MIT Press, 2005).
26. Hrdy, S. B. *Mothers and Others: The Evolutionary Origins of Mutual Understanding* (Cambridge, MA: Harvard University Press, 2011).
27. Axelrod, R. M. *The Evolution of Cooperation* (New York: Basic Books, 1984).
28. Hill, K. R. et al. Co-residence patterns in hunter-gatherer societies show unique human social structure. *Science* 331, 1286–1289 (2011).
29. Boehm, C. *Blood Revenge: The Enactment and Management of Conflict in Montenegro and Other Tribal Societies* (Lawrence, KS: University Press of Kansas, 1984).
30. Henrich, J. & Muthukrishna, M. The origins and psychology of human cooperation. *Annual Review of Psychology* 72, 207–240 (2021).
31. Rusch, H. The evolutionary interplay of intergroup conflict and altruism in humans: a review of parochial altruism theory and prospects for its extension. *Proceedings of the Royal Society B: Biological Sciences* 281, 20141539 (2014).
32. Apicella, C. L., Marlowe, F. W., Fowler, J. H. & Christakis, N. A. Social networks and cooperation in hunter-gatherers. *Nature* 481, 497 (2012).
33. Lehmann, L. & Rousset, F. How life history and demography promote or inhibit the evolution of helping behaviours. *Philosophical Transactions of the Royal Society B: Biological Sciences* 365, 2599–2617 (2010).
34. Taylor, P. D. Altruism in viscous populations: an inclusive fitness model. *Evolutionary Ecology* 6, 352–356 (1992).
35. Kaplan, H., Lancaster, J. & Robson, A. Embodied capital and the evolutionary economics of the human life span. *Population and Development Review* 29, 152–182 (2003).

36. Fehr, E. & Schmidt, K. M. A theory of fairness, competition, and cooperation. *Quarterly Journal of Economics* **114**, 817–868 (1999).
37. Bowles, S. *Microeconomics: Behavior, Institutions, and Evolution* (New York: Russell Sage, 2004).
38. Chudek, M. & Henrich, J. Culture–gene coevolution, norm-psychology and the emergence of human prosociality. *Trends in Cognitive Sciences* **15**, 218–226 (2011).
39. Gibbons, R. S. *Game Theory for Applied Economists* (Princeton University Press, 1992).
40. Bell, A. V., Richerson, P. J. & McElreath, R. Culture rather than genes provides greater scope for the evolution of large-scale prosociality. *Proceedings of the National Academy of Sciences* **106**, 17671–17674 (2009).
41. Brandts, J. & Charness, G. The strategy versus the direct-response method: a first survey of experimental comparisons. *Experimental Economics* **14**, 375–398 (2011).
42. [A note. Not a citation.]
43. Boyd, R. & Lorberbaum, J. P. No pure strategy is evolutionarily stable in the repeated prisoner's dilemma game. *Nature* **327**, 58–59 (1987).
44. van Veelen, M., García, J., Rand, D. G. & Nowak, M. A. Direct reciprocity in structured populations. *Proceedings of the National Academy of Sciences* **109**, 9929–9934 (2012).
45. Boyd, R. Mistakes allow evolutionary stability in the repeated prisoner's dilemma game. *Journal of Theoretical Biology* **136**, 47–56 (1989).
46. Price, G. R. Extension of covariance selection mathematics. *Annals of Human Genetics* **35**, 485–490 (1972).
47. Gat, A. *War in Human Civilization* (Oxford: Oxford University Press, 2006).
48. Bowles, S. Did warfare among ancestral hunter-gatherers affect the evolution of human social behaviors? *Science* **324**, 1293–1298 (2009).
49. Fry, D. P. & Söderberg, P. Lethal aggression in mobile forager bands and implications for the origins of war. *Science* **341**, 270–273 (2013).
50. Clutton-Brock, T. Cooperation between non-kin in animal societies. *Nature* **462**, 51–57 (2009).
51. Romano, A., Sutter, M., Liu, J. H., Yamagishi, T. & Balliet, D. National parochialism is ubiquitous across 42 nations around the world. *Nature Communications* **12**, 4456 (2021).
52. Axelrod, R. & Hamilton, W. D. The evolution of cooperation. *Science* **211**, 1390–1396 (1981).

Response to Referee #2

Thank you again for your ongoing commitment to serving as a referee for our paper, “Super-additive cooperation”. Your contribution has been very useful. We were especially pleased to learn that you found our initial round of revisions “thorough and constructive”, that the “core message is clearer”, and that the introductory paragraphs provide a “strong entry into the paper”.

After reading our first revision, you had a number of additional suggestions for how to improve clarity on key points. Thank you for these suggestions. We have taken all of them on board and revised accordingly. We feel the paper is much improved due to the feedback from you, your fellow referees, and the editor. We hope you will agree and support publication of the current revision in *Nature*.

For your convenience, we now detail the changes we made in response to your comments. Our responses below are self-contained in the sense that we explain in detail how we changed the paper as a result of your comments.

State methods (e.g. “evolutionary agent-based simulations”) more explicitly in the abstract.

Thank you for encouraging us to do this. It will surely help readers to understand quickly what kind of study this is. You recommended using the phrase “we show with evolutionary simulations”. Although we do rely heavily on evolutionary simulations, we use three different kinds of methods altogether, namely analytical modelling, simulation modelling, and a behavioural experiment. All of these methods feed into our overall set of findings, and we see this triangulation of methods as a strength of the paper. Thus, we have revised the abstract to reflect the larger set of methods.

Using analytical and simulation models and a behavioural experiment, we show that neither mechanism reliably supports the evolution of cooperation when actions vary continuously.

p. 1

Define ambiguous reciprocity “up-front” in the abstract. Relatedly, why does ambiguous reciprocity tend “to disappear from the abstract” when the discussion turns to group competition?

We agree that the exact meaning of ambiguous reciprocity is not clear in the abstract. Unfortunately, however, our abstract is already well beyond the recommended length, and we do not have the space to go into further details in the abstract. For this reason, we simply signal that it’s a new type of strategy by saying that it’s a strategy “generally ignored in models of reciprocal altruism”. It also has big effects in the sense that it “completely undermines cooperation under repeated interactions”. Our intention is to pique interest in this way until we can present the details later in the paper.

You are also correct that, when the abstract turns from repeated interactions to intergroup competition, ambiguous reciprocity fades into the background. This is appropriate because, as we explain later in the paper, whether or not ambiguous strategies can arise via mutation has no consequences when group competition is present. Instead, the evolutionary effects of intergroup competition depend on other considerations; they do not depend on the dimensionality of the strategy space. The dimensionality of

strategy space, and thus whether or not ambiguous strategies can arise via mutation, is only important under repeated interactions as a stand-alone mechanism.

That said, your overall set of comments about ambiguous reciprocity showed us that we should be clearer about what it means to allow ambiguous strategies in an evolutionary model. We take up this issue later in this letter.

Summarise key messages before we “dive into” into the details of the model. Specifically, consider moving abbreviated versions of the material on pp. 22-23 (Revision 1) to the top of p. 6 (Revision 1).

This will surely help the reader see where the paper is headed. We have taken your advice and incorporated brief summaries of the key findings just before turning to the details of the model. The revised paper now reads,

With our jointly theoretical and experimental approach, we show that neither hypothesis about the evolution of one-shot cooperation actually works. Neither repeated interactions alone nor intergroup competitions alone support ingroup cooperation in a meaningful way, and neither mechanism leads to ingroup and outgroup predictions consistent with the one-shot behaviour observed in Papua New Guinea. Specifically, repeated interactions generate a cooperative equilibrium, but this equilibrium is exceedingly vulnerable to invasion by a class of naturally occurring mutations we call “ambiguous reciprocity”. Gratuitously assuming that such mutations are impossible eliminates the vulnerability, but this approach has no biological justification. Group competition does not support ingroup cooperation by itself because several mechanisms reduce both the variation between groups and the extent to which group selection can occur given the limited variation that does persist.

Crucially, even though the discussion of the two hypotheses often seems to treat them as strict alternatives, they are not. Repeated interactions within groups and competitions between groups can coincide²¹. We also show that combining the two mechanisms generates strong positive interactions. Positive interactions occur because intergroup competition can stabilise ingroup cooperation against mutations that repeatedly introduce ambiguous reciprocity into the population, and intergroup competitions often do this even when they do not support cooperation on their own. When the two mechanisms interact in this way, the result is the evolution of cooperative reciprocity with ingroup members, which amplifies cooperation within groups, and uncooperative reciprocity with outgroup members, which erodes cooperation between groups. This mix in which all equilibrium strategies are reciprocal, but not all reciprocal strategies are cooperative, is exactly what we observed among our participants in Papua New Guinea. Thus, an evolved psychology based on repeated interactions in the ancestral past may be necessary to explain contemporary one-shot cooperation with ingroup partners, but such an evolved psychology is definitely not sufficient. Intergroup competitions are also necessary but not sufficient. The joint influence of the two mechanisms, in contrast, can provide a sufficient explanation for the evolution of one-shot cooperation.

pp. 5-6

Explain better what it means to transfer “any amount”, specifically “as operationalized in the simulations”.

In previous versions of the paper, when describing the basic framework for the models, we said that the players “transfer any amount” to their partners. You pointed out that this was not clear, and thank you for doing so. What we really wanted to say is that players “can transfer any amount”. We have revised the paper with this in mind.

For the stage game (Supplementary Information § 1), each player in a pair has an endowment normalised to one. The first mover can transfer any amount from her endowment to the second mover, and the transfer is doubled. Conditional on the first mover's transfer, the second mover can also transfer any amount from her endowment to the first mover, and this transfer is also doubled.

p. 7

What do we mean by saying that individuals “only have ingroup strategies” in the repeated interactions scenario? In addition, better explain how this claim relates to the “next sentence on predictions for the experiment”.

Thank you for finding these instances of ineffective communication. What we were trying to say is that individuals only play the social dilemma, whether one-shot or repeated, with ingroup partners. It's much more effective, of course, to just say that! We have revised accordingly.

The **repeated interactions scenario** consists of models of populations sub-divided into 40 groups of 24 individuals each without any competition between groups. Individuals within groups pair off randomly to play the game. Individuals only play the social dilemma with ingroup partners, and we consider both one-shot games and repeated interactions (Supplementary Information § 2.1.4). Because individuals only play with ingroup partners, the repeated interactions scenario isolates the effects of repeated interactions and the reputational concerns they create from the effects of intergroup competition and more generally outgroup interactions of all sorts.

p. 8

Regarding the link between the description of the scenarios and the predictions for the experiment, you and Referee #3 both felt this could be improved. Referee #3 suggested a restructuring that we have implemented in the revision. The idea is the following. Instead of presenting the three modelling scenarios and associated predictions together, we first present the three scenarios and then present the logic behind deriving predictions. This results in the following passage on predictions in the revised paper.

The strategies that evolve under the three scenarios provide predictions for our experiment. For the repeated interactions scenario, the strategies that evolve under repeated interactions provide predictions for the ingroup treatment in our experiment, while the strategies that evolve under one-shot play provide predictions for the outgroup treatment. This captures the hypothesis that ingroup interactions activate an ancestral psychology based on repeated play, while outgroup interactions, assumed to be rare and typically one-shot in the ancestral past, leave this psychology dormant¹⁰⁻¹³.

Thus, if we want to predict how people play with ingroup partners, we need to identify the strategies that evolve in theory under repeated play. If we want to predict how people play with outgroup partners, we need to know the strategies that evolve in theory under one-shot play.

For the group competition scenario, individuals play the social dilemma with both ingroup partners and outgroup partners, and thus strategies for doing so are explicitly conditional on group affiliation. Our experiment involved both ingroup and outgroup pairings, and so we can directly derive ingroup and outgroup predictions for the experiment from the ingroup and outgroup strategies that evolve in the model. The opportunity to cooperate with outgroup partners in our models is different from most evolutionary models of parochialism because most models limit attention to outgroup strategies that range from defection to outright aggression³¹. Defection is thus the most generous feasible option for an outgroup interaction. In general, the group competition scenario isolates the effects of intergroup competition from the effects of repeated interactions and associated reputational concerns within groups. Finally, for the joint scenario, individuals also play the social dilemma with ingroup and outgroup partners, and predictions for the experiment follow directly from the ingroup and outgroup strategies that evolve in the model. However, because the joint scenario combines repeated interactions within groups and competition between groups into a novel selective regime, it can potentially support the evolution of strategies that differ from those that evolve when the two mechanisms operate in isolation.

pp. 9-10

Clarify how ambiguous strategies appear and why they are important. In particular, “[w]hy would such [a] strategy be played”, and how is it important to that “they can be played in theory”?

You pinpointed some potential for confusion about ambiguous reciprocity. Specifically, you argued that we were not clear about how ambiguous strategies appear in the model, and indeed some readers might incorrectly infer that we force ambiguous strategies into the simulations. Thank you for pointing this out because the issue is indeed quite important. In short, we do not force ambiguous strategies to appear in any way. Instead, we simply allow them to arise (3-D, 4-D) or not (2-D) via random mutations.

An analogy with a standard prisoner’s dilemma illustrates the basic idea. The simplest model of reciprocal altruism under repeated play of the prisoner’s dilemma examines the evolutionary stability of two possible strategies, namely unconditional defection (ALLD) and tit-for-tat (TFT). The strategy space is {ALLD, TFT}. This means that, if a mutation occurs, the mutation can only produce one of these strategies. If an ALLD individual replicates and a mutation occurs, the offspring is TFT. If a TFT replicates and a mutation occurs, the offspring is ALLD. With this strategy space, provided the probability of continuing to play with one’s partner is high enough, both strategies are evolutionary stable strategies. The evolutionary stability of TFT, in particular, provides the basis for the idea that repeated play supports the evolution of reciprocal strategies.

What happens if we change the strategy space to include unconditional cooperation (ALLC)? The strategy space becomes {ALLD, TFT, ALLC}, and mutations now produce one of three possible strategies. In this case, ALLC is never an evolutionary stable strategy, which is not surprising. However, as first pointed out by Reinhard Selten and Peter Hammerstein (McElreath and Boyd, 2007, *Mathematical Models of Social Evolution*, p. 133), including it in the strategy space can also undermine the evolutionary

stability of TFT, which is surprising. This is important because there's no biological reason to exclude ALLC, and the mere fact that it appears and then disappears can have strong effects on evolutionary outcomes.

Our approach to ambiguous strategies is analogous. In our two-dimensional strategy space, if a mutation occurs, it cannot produce an ambiguous strategy because, mathematically, two dimensions are not enough to represent ambiguous strategies. This, however, has nothing to do with biology. So, what happens if we work with higher-dimensional spaces in our model? We accept a higher degree of mathematical complexity, but we can also better represent the potential complexity of real biological systems. In our specific case, three- and four-dimensional strategy spaces allow all sorts of strategies to appear via mutation that are not possible in the two-dimensional case. This does not mean we force the strategies into a simulated population. Instead, we simply allow them to appear via mutation. You settled on this interpretation in your referee report when you concluded that the important point is that three and four dimensions mean that more complex strategies "can be played in theory". This is absolutely correct.

That said, if ambiguous strategies are allowed, they readily appear for the following reasons. If a population has converged on a state in which all individuals are either escalators or de-escalators, ambiguous strategies will naturally appear when mutations occur. Mutations to the left intercepts of de-escalating response functions produce ambiguous strategies, and mutations to the right intercepts of escalating response functions also produce ambiguous strategies. For this reason, the only way to exclude ambiguous strategies is to simply assume, as a modeller, that they are not allowed. This is what we do in our two-dimensional models. Such an approach, however, has no biological basis. It's like excluding ALLC in our example above. ALLC may not be a good strategy from the individual's perspective, but that's no reason to exclude it theoretically.

A similar logic applies in our case. Moreover, as we find, when cooperation levels vary continuously, one has to make extreme restrictions for repeated interactions to support the evolution of cooperation. One has to restrict the strategy space in a way that minimizes the role of continuous choices and makes the game, in effect, as close as possible to the standard PD with binary actions. As we explain in the revision,

[i]f we only allow escalating and de-escalating strategies, as with our two-dimensional models, we make the continuous game similar to the standard prisoner's dilemma. Specifically, as escalating or de-escalating reciprocity become common, interacting players tend to converge quickly on full cooperation or full defection respectively. Players do not converge on intermediate levels of cooperation, which minimizes the role of continuous actions by making the continuous game similar to the standard game. Without this restriction, repeated interactions do not reliably support the evolution of cooperation, and our repeated interactions scenario paints an even more dismal picture than studies relying on the standard prisoner's dilemma^{43,44}. In our case, a simple three-dimensional strategy that conditions only on the partner's most recent move is already enough to destabilize cooperative strategies, with little to no cooperation the final outcome.

p. 29

Importantly, your comments showed to us that we need to be more explicit about two key points. First, we do not force ambiguous strategies to appear. Instead, we either allow them to appear via mutation or not. Second, however, if we do allow ambiguous strategies, they readily appear precisely because

ambiguous strategies require only small mutations to escalating and de-escalating strategies, which is exactly how we think of mutations, i.e., small changes to resident strategies. We have made several rewrites throughout the paper to emphasize these two points. Here we quote some representative passages from the revision for your convenience.

[R]epeated interactions generate a cooperative equilibrium, but this equilibrium is exceedingly vulnerable to invasion by a class of naturally occurring mutations we call “ambiguous reciprocity”. Gratuitously assuming that such mutations are impossible eliminates the vulnerability, but this approach has no biological justification.

p. 5

The number of dimensions used to specify a strategy controls which strategies can and cannot arise via mutation.

p. 11

Ambiguous strategies, if allowed, arise readily from mutations of de-escalating and escalating strategies (Supplementary Information § 1.2.12).

Fig. 1 caption

Explain group competitions better to avoid confusion and apparent inconsistencies. In particular, the description of whether group competition occurs is such that “the reader may think individuals played another game” after the social dilemma, even though this is not the case. Additionally, the description of group competitions on p. 14 (Revision 1) “doesn’t seem to match one-on-one with the description on p. 8 [Revision 1]”.

Intergroup competition involves two basic components. First, do specific groups in a specific generation actually compete? Second, conditional on a competition occurring, who wins? The answers to these two questions jointly shape the intensity of group selection in the model.

Your comments helped us to realise that we should do a better job explaining how these two components work in our models. With respect to the occurrence of competitions, you pointed out that our description of intergroup competition sounded as if agents play a second game in which they decide if they engage in a group competition. As you correctly inferred, however, agents do not play a second game, and we have clarified this point in the revision in light of your observation.

After game play, we model the occurrence of group competitions by assuming that paired groups compete against each other with relatively low probabilities¹ (Supplementary Information § 2.1.7) that decrease as the groups become more similar (Supplementary Information § 2.1.5). Although we do not explicitly model decision making at the group level, our approach reflects the idea that paired groups assess each other and avoid competing when they have trouble identifying the likely winner, which is consistent with both past modelling work and ethnographic evidence^{4,29}.

pp. 8-9

¹ The competition rates emerging in our simulations are consistent with, but slightly below, the rigorous empirical estimates of Bowles (2009, cited in endnote 48). We adopted parameter values that produce a slightly conservative competition rate because some researchers (Frey & Söderberg 2013, cited in endnote 49) consider Bowles’ mortality estimate to be a bit high. Note that the empirical literature uses mortality data as the main proxy for the prevalence of group competitions.

Later in the paper, we discuss the details of intergroup competition again, but this time we focus on the second half of the process, namely who wins given a competition. You mentioned that this second passage (p. 14, Revision 1) could seem inconsistent with respect to the first (p. 8, Revision 1), and we'd like to thank you for pointing out this potential source of confusion. To clarify that we are talking about two different stages of intergroup competition, with the second conditional on the outcome of the first, we have modified our description of the second component to read as follows.

Conditional on a group competition occurring between paired groups, as explained above, the group with more resources wins the competition with a probability more or less sensitive to the difference in total resources between the two groups. We consider four levels of sensitivity (Supplementary Information §2.1.5) controlled by the parameter $\lambda \in \{0,10,25,100\}$. If $\lambda = 0$, which group wins is unrelated to the difference in total resources. Groups compete in this case, but outcomes are unsystematic. Therefore, group selection cannot occur, and in this sense $\lambda = 0$ is like the repeated interactions scenario. As λ values increase, the group with more resources is increasingly likely to win, and the group competition and joint scenarios are increasingly different from the repeated interactions scenario.

p. 16

What are the implications for the “evolution of cooperat[ive] strategies” if only “asymmetrical” groups with different resources compete with “non-zero probability”?

Importantly, the assumption that competitions do not occur between groups with identical payoffs cannot have any evolutionary implications. To see the logic, imagine that we add competitions between groups with identical payoffs. These added competitions would not generate selection at the group level because groups with identical payoffs would also have identical chances of winning the competition. The winners of the competitions would thus be completely random and unsystematic, which is equivalent to saying the competitions in question would not generate group selection.

We are thus confident that our choice of protocol, which excludes competitions between symmetric groups, is not crucial. Our protocol does, however, have some key advantages. It's consistent with other well-known models of group competition, especially Choi and Bowles (2007), and it's consistent with ethnographic evidence (see quoted passage from revision at the end of the preceding page of this letter). Christopher Boehm's book on blood feuds in Montenegro²⁹, for example, describes a process strikingly similar to our approach. By Boehm's account, groups assess each other, and they avoid conflict when they seem evenly matched. Viewed the other way around, as the asymmetry between groups increases, the likelihood of a conflict also increases, which is exactly what our model does.

What is the “prevalence of competition” between groups under different conditions and scenarios?

With respect to the occurrence of intergroup competitions, i.e., the first component, you also correctly pointed out that our approach means the probability paired groups have a competition varies. As a result, you recommended showing the rates of intergroup competition.

The supplement now provides extensive information on rates of group competition, and in our revised version we point the reader to this information when we first introduce the group competition scenario (p. 8). For example, as we explain in the Supplementary Information § 2.1.7, our three-dimensional simulations produced an overall competition rate of 0.227 between paired groups, which is equivalent to a mortality rate due to group competition of 0.114². This is consistent with, but slightly below, the rigorous empirical estimates of Bowles⁴⁸. We adopted parameter values that produce a slightly conservative competition rate because some researchers⁴⁹ consider Bowles’ mortality estimate to be a bit high.

Note, however, that the actual competition rate is not by itself crucial for the occurrence of group *selection*. This is true because the extent to which group selection occurs depends on how strongly the resource differences between competing groups translate into winning versus losing the competition. As described above, we captured this with the parameter λ .

Supplementary Figures 17 – 28 provide complete information about mortality rates due to group competition, where mortality rates are simply half of the group competition rates (see Supplementary Information § 2.1.7). These graphs show that group competition rates vary slightly across scenarios and parameter values, but the variation is extremely limited and relatively unsystematic. We provide two representative examples to illustrate this in the figures below. They show the rate of group competitions for the group competition scenario ($N = 1$) and the joint scenario ($N = 100$).

As the figures show, the variation across scenarios is limited and relatively unsystematic. Sometimes the competition rates are higher in the joint scenario than in the group competition scenario (e.g., Supplementary Figure 17a), while sometimes they’re the same (e.g. Supplementary Figure 28d) or even slightly lower (e.g. Supplementary Figure 17d). Sometimes the competition rate increases with cancellation effects at the group level (e.g. Supplementary Figure 28a, $N = 1$), sometimes it decreases (e.g. Supplementary Figure 28a, $N = 100$), and often it does not vary at all with cancellation effects at the group level (e.g. Supplementary Figure 28b – 28d). Moreover, the variation that does exist is limited.

² Recall that the empirical literature uses mortality data as the main proxy for the prevalence of group competition. The mortality rate due to intergroup competition is one half the competition rate as measured at the group level (Supplementary Information § 2.1.7).

Supplementary Figure 17 | Agent mortality due to inter-group competition under three-dimensional strategies. The graphs show mortality due to inter-group competition averaged over simulations given specific parameter values. Initial conditions consist of seeding the population with unconditionally selfish strategies. Group competitions have unsystematic outcomes ($\lambda = 0$). Each panel shows both the group competition ($N = 1$) and joint scenarios ($N = 100$). Life cycles either decouple game play and individual selection (**a**, **c**), which attenuates cancellation effects at the individual level, or they couple game play and individual selection (**b**, **d**), which increases cancellation effects at the individual level. Migration (m_j) and the intensity of cancellation effects at the group level (Ξ) both vary, with group-level cancellation effects declining as Ξ increases.

Supplementary Figure 28 | Agent mortality due to inter-group competition under three-dimensional strategies. The graphs show mortality due to inter-group competition averaged over simulations given specific parameter values. Initial conditions consist of seeding the population with perfectly reciprocal strategies. Group competitions have systematic outcomes based on $\lambda = 100$. Each panel shows both the group competition ($N = 1$) and joint scenarios ($N = 100$). Life cycles either decouple game play and individual selection (**a**, **c**), which attenuates cancellation effects at the individual level, or they couple game play and individual selection (**b**, **d**), which increases cancellation effects at the individual level. Migration (m_j) and the intensity of cancellation effects at the group level (Ξ) both vary, with group-level cancellation effects declining as Ξ increases.

The limited variation in competition rates across scenarios is a straightforward consequence of another pattern explained elsewhere in the supplement. Namely, we find (Supplementary Information § 2.1.19) that one and only one mechanism *systematically* affects the differences between groups that arise in our models. This mechanism is the migration rate. If the migration rate is low, approximately 7% of the total strategy variation in simulated populations is between groups (compared to within groups). If the migration rate is high, just under 5% of the total strategy variation is between groups.

The upshot is the following. Variation in migration rates has small but consistent effects in terms of differences between groups. Because we manipulate the migration rate in the same way across all scenarios, the group-level differences that follow are effectively the same across all scenarios. By extension, scenarios do not differ in terms of the extent to which group-level differences in strategies emerge.

Thank you!

We'd like to thank you again for your insightful and constructive approach to our paper. Your contribution has helped to improve the paper greatly, and we are grateful for this. We also hope you will support publication of "Super-additive cooperation" in *Nature*.

Sincerely,

Charles Efferson and Ernst Fehr

References from Main Paper

1. Hagen, E. H. & Hammerstein, P. Game theory and human evolution: A critique of some recent interpretations of experimental games. *Theoretical Population Biology* 69, 339–348 (2006).
2. Zefferman, M. R. Direct reciprocity under uncertainty does not explain one-shot cooperation, but demonstrates the benefits of a norm psychology. *Evolution and Human Behavior* 35, 358–367 (2014).
3. Jagau, S. & van Veelen, M. A general evolutionary framework for the role of intuition and deliberation in cooperation. *Nature Human Behaviour* 1, 1–6 (2017).
4. Choi, J.-K. & Bowles, S. The coevolution of parochial altruism and war. *Science* 318, 636–640 (2007).
5. Richerson, P. et al. Cultural group selection plays an essential role in explaining human cooperation: A sketch of the evidence. *Behavioral and Brain Sciences* 39 (2016).
6. Boyd, R. & Richerson, P. J. Large-scale cooperation in small-scale foraging societies. *Evolutionary Anthropology: Issues, News, and Reviews* (2022).
7. De Dreu, C. K., Fariña, A., Gross, J. & Romano, A. Prosociality as a foundation for intergroup conflict. *Current Opinion in Psychology* 44, 112–116 (2022).
8. Akdeniz, A. & van Veelen, M. The cancellation effect at the group level. *Evolution* (2020).

9. Haselton, M. G., Nettle, D. & Murray, D. R. The evolution of cognitive bias. *The Handbook of Evolutionary Psychology* 1–20 (2015).
10. Balliet, D., Wu, J. & De Dreu, C. K. Ingroup favoritism in cooperation: a meta-analysis. *Psychological Bulletin* 140, 1556 (2014).
11. Yamagishi, T., Jin, N. & Kiyonari, T. Bounded generalized reciprocity: ingroup boasting and ingroup favoritism. *Advances in Group Processes* 16, 161–197 (1999).
12. Haley, K. J. & Fessler, D. M. T. Nobody's watching?: subtle cues affect generosity in an anonymous economic game. *Evolution and Human Behavior* 26, 245–256 (2005).
13. Yamagishi, T. & Mifune, N. Parochial altruism: does it explain modern human group psychology? *Current Opinion in Psychology* 7, 39–43 (2016).
14. Alger, I., Weibull, J. W. & Lehmann, L. Evolution of preferences in structured populations: genes, guns, and culture. *Journal of Economic Theory* 185, 104951 (2020).
15. Handley, C. & Mathew, S. Human large-scale cooperation as a product of competition between cultural groups. *Nature Communications* 11, 1–9 (2020).
16. Wilson, D. S. & Wilson, E. O. Rethinking the theoretical foundation of sociobiology. *The Quarterly Review of Biology* 82, 327–348 (2007).
17. Gross, J. & De Dreu, C. K. The rise and fall of cooperation through reputation and group polarization. *Nature Communications* 10, 1–10 (2019).
18. Henrich, J. Cultural group selection, coevolutionary processes and large-scale cooperation. *Journal of Economic Behavior and Organization* 53, 3–35 (2004).
19. Muthukrishna, M. & Henrich, J. A problem in theory. *Nature Human Behaviour* 3, 221–229 (2019).
20. Akdeniz, A. & van Veelen, M. The evolution of morality and the role of commitment. *Evolutionary Human Sciences* 3 (2021).
21. Henrich, N. & Henrich, J. *Why Humans Cooperate: A Cultural and Evolutionary Explanation* (Oxford: Oxford University Press, 2007).
22. Wahl, L. M. & Nowak, M. A. The continuous prisoner's dilemma: I. linear reactive strategies. *Journal of Theoretical Biology* 200, 307–321 (1999).
23. Le, S. & Boyd, R. Evolutionary dynamics of the continuous iterated prisoner's dilemma. *Journal of Theoretical Biology* 245, 258–267 (2007).
24. Gurven, M. To give and to give not: the behavioral ecology of human food transfers. *Behavioral and Brain Sciences* 27, 543–583 (2004).
25. Kaplan, H. & Gurven, M. The natural history of human food sharing and cooperation: a review and a new multi-individual approach to the negotiation norms. In Gintis, H., Bowles, S., Boyd, R. & Fehr, E. (eds.) *Moral Sentiments and Material Interests: The Foundation of Cooperation in Economic Life*, 75–113 (Cambridge: The MIT Press, 2005).
26. Hrdy, S. B. *Mothers and Others: The Evolutionary Origins of Mutual Understanding* (Cambridge, MA: Harvard University Press, 2011).

27. Axelrod, R. M. *The Evolution of Cooperation* (New York: Basic Books, 1984).
28. Hill, K. R. et al. Co-residence patterns in hunter-gatherer societies show unique human social structure. *Science* 331, 1286–1289 (2011).
29. Boehm, C. *Blood Revenge: The Enactment and Management of Conflict in Montenegro and Other Tribal Societies* (Lawrence, KS: University Press of Kansas, 1984).
30. Henrich, J. & Muthukrishna, M. The origins and psychology of human cooperation. *Annual Review of Psychology* 72, 207–240 (2021).
31. Rusch, H. The evolutionary interplay of intergroup conflict and altruism in humans: a review of parochial altruism theory and prospects for its extension. *Proceedings of the Royal Society B: Biological Sciences* 281, 20141539 (2014).
32. Apicella, C. L., Marlowe, F. W., Fowler, J. H. & Christakis, N. A. Social networks and cooperation in hunter-gatherers. *Nature* 481, 497 (2012).
33. Lehmann, L. & Rousset, F. How life history and demography promote or inhibit the evolution of helping behaviours. *Philosophical Transactions of the Royal Society B: Biological Sciences* 365, 2599–2617 (2010).
34. Taylor, P. D. Altruism in viscous populations: an inclusive fitness model. *Evolutionary Ecology* 6, 352–356 (1992).
35. Kaplan, H., Lancaster, J. & Robson, A. Embodied capital and the evolutionary economics of the human life span. *Population and Development Review* 29, 152–182 (2003).
36. Fehr, E. & Schmidt, K. M. A theory of fairness, competition, and cooperation. *Quarterly Journal of Economics* 114, 817–868 (1999).
37. Bowles, S. *Microeconomics: Behavior, Institutions, and Evolution* (New York: Russell Sage, 2004).
38. Chudek, M. & Henrich, J. Culture–gene coevolution, norm-psychology and the emergence of human prosociality. *Trends in Cognitive Sciences* 15, 218–226 (2011).
39. Gibbons, R. S. *Game Theory for Applied Economists* (Princeton University Press, 1992).
40. Bell, A. V., Richerson, P. J. & McElreath, R. Culture rather than genes provides greater scope for the evolution of large-scale prosociality. *Proceedings of the National Academy of Sciences* 106, 17671–17674 (2009).
41. Brandts, J. & Charness, G. The strategy versus the direct-response method: a first survey of experimental comparisons. *Experimental Economics* 14, 375–398 (2011).
42. [A note. Not a citation.]
43. Boyd, R. & Lorberbaum, J. P. No pure strategy is evolutionarily stable in the repeated prisoner's dilemma game. *Nature* 327, 58–59 (1987).
44. van Veelen, M., García, J., Rand, D. G. & Nowak, M. A. Direct reciprocity in structured populations. *Proceedings of the National Academy of Sciences* 109, 9929–9934 (2012).
45. Boyd, R. Mistakes allow evolutionary stability in the repeated prisoner's

- dilemma game. *Journal of Theoretical Biology* 136, 47–56 (1989).
46. Price, G. R. Extension of covariance selection mathematics. *Annals of Human Genetics* 35, 485–490 (1972).
47. Gat, A. *War in Human Civilization* (Oxford: Oxford University Press, 2006).
48. Bowles, S. Did warfare among ancestral hunter-gatherers affect the evolution of human social behaviors? *Science* 324, 1293–1298 (2009).
49. Fry, D. P. & Söderberg, P. Lethal aggression in mobile forager bands and implications for the origins of war. *Science* 341, 270–273 (2013).
50. Clutton-Brock, T. Cooperation between non-kin in animal societies. *Nature* **462**, 51-57 (2009).
51. Romano, A., Sutter, M., Liu, J. H., Yamagishi, T. & Balliet, D. National parochialism is ubiquitous across 42 nations around the world. *Nature Communications* 12, 4456 (2021).
52. Axelrod, R. & Hamilton, W. D. The evolution of cooperation. *Science* 211, 1390–1396 (1981).

Response to Referee #3

Thank you once more for serving as a referee for our paper, “Super-additive cooperation”. Your comments have led to several significant improvements, and we are grateful for your ongoing contribution. We are very glad to learn that you think super-additivity “stems from what is likely to be a fundamental feature of the evolution of reciprocal cooperation”, and that you “enthusiastically support publication” of our paper in *Nature*.

You identified two additional revisions that would, if implemented, help the busy reader. We implemented both of your recommendations, and we hope you find the improvements compelling. We detail these changes below.

First present the three scenarios (repeated interactions, group competition, joint) and then present how we use these scenarios to derive predictions for the experiment.

This was a very useful suggestion. By mixing both our explanations of what the scenarios are and our explanations of how we use them to derive predictions, we may overburden a busy reader. We have followed your advice exactly. For your convenience, we describe the revised descriptions of the three scenarios in the first two paragraphs below. The third and the fourth paragraphs then provide the predictions.

The **repeated interactions scenario** consists of models of populations sub-divided into 40 groups of 24 individuals each without any competition between groups. Individuals within groups pair off randomly to play the game. Individuals only play the social dilemma with ingroup partners, and we consider both one-shot games and repeated interactions (Supplementary Information § 2.1.4). Because individuals only play with ingroup partners, the repeated interactions scenario isolates the effects of repeated interactions and the reputational concerns they create from the effects of intergroup competition and more generally outgroup interactions of all sorts. We ignore uncertainty about whether a game is one-shot or repeated^{2,3}, which maximises the scope for repeated interactions to support cooperation when relationships actually do last a long time.

The **group competition scenario** also consists of models in sub-divided populations. In this scenario, however, groups compete, and games are always one-shot. Groups are paired within a generation (Supplementary Information § 2.1.5). Each individual plays both a one-shot social dilemma with a randomly selected ingroup partner and a one-shot social dilemma with a randomly selected outgroup partner from the paired group. The individual has separate strategies for these two interactions. After game play, we model the occurrence of group competitions by assuming that paired groups compete against each other with relatively low probabilities (Supplementary Information § 2.1.7) that decrease as the groups become more similar (Supplementary Information § 2.1.5). Although we do not explicitly model decision making at the group level, this approach reflects the idea that paired groups assess each other and avoid competing when they have trouble identifying the likely winner, which is consistent with both past modelling work and ethnographic evidence^{4,29}. We can think of a competition as a violent conflict, a competition for some limited resource, or a process where the culture of one group displaces the culture of another group³⁰. The **joint scenario** combines both repeated interactions within groups and competition between groups (Supplementary Information § 2.1.6). It is identical to the group competition scenario with one exception; ingroup interactions are always repeated.

The strategies that evolve under the three scenarios provide predictions for our experiment. For the repeated interactions scenario, the strategies that evolve under repeated interactions provide predictions for the ingroup treatment in our experiment, while the strategies that evolve under one-shot play provide predictions for the outgroup treatment. This captures the hypothesis that ingroup interactions activate an ancestral psychology based on repeated play, while outgroup interactions, assumed to be rare and typically one-shot in the ancestral past, leave this psychology dormant¹⁰⁻¹³. Thus, if we want to predict how people actually play with ingroup partners, we need to identify the strategies that evolve in theory under repeated play. If we want to predict how people actually play with outgroup partners, we need to know the strategies that evolve in theory under one-shot play.

For the group competition scenario, individuals play the social dilemma with both ingroup partners and outgroup partners, and thus strategies for doing so are explicitly conditional on group affiliation. Our experiment involved both ingroup and outgroup pairings, and so we can directly derive ingroup and outgroup predictions for the experiment from the ingroup and outgroup strategies that evolve in the model. The opportunity to cooperate with outgroup partners in our models is different from most evolutionary models of parochialism because most models limit attention to outgroup strategies that range from defection to outright aggression³¹. Defection is thus the most generous feasible option for an outgroup interaction. In general, the group competition scenario isolates the effects of intergroup competition from the effects of repeated interactions and associated reputational concerns within groups. Finally, for the joint scenario, individuals also play the social dilemma with ingroup and outgroup partners, and predictions for the experiment follow directly from the ingroup and outgroup strategies that evolve in the model. However, because the joint scenario combines repeated interactions within groups and competition between groups into a novel selective regime, it can potentially support the evolution of strategies that differ from those that evolve when the two mechanisms operate in isolation.

pp. 8-10

The reader needs to know how to interpret the histograms over three-dimensional strategy space that appear in the Supplementary Information (e.g. Supplementary Figures 69-106).

Thank you for pointing this out. Because we have so many of these histograms (heat maps specifically), we do not want to repeat the information over and over. Doing so, in particular, would force us to make the figures on each page much smaller and harder to read.

As a solution, we have a section of the supplement (§ 2.1.9) that explains how to read these histograms. To avoid the problem you discovered, we have added to every relevant figure caption, “See § 2.1.9 for an explanation of how to read this type of graph”.

Incidentally, your comment also led us to take analogous approaches for the graphs showing dynamics under a two-dimensional strategy space (§ 2.2.1, Supplementary Figures 119-124) and a four-dimensional strategy space (§ 2.3.1, Supplementary Figures 125-148).

Thank you!

We’d like to thank you again for your constructive and insightful comments on our paper. Your recommendation to add choice mistakes, in particular, dramatically improved the scope of the research and the confidence one can have in the results. We hope you agree and will continue to support publication of “Super-additive cooperation” in *Nature*.

Sincerely,

Charles Efferson and Ernst Fehr

References from Main Paper

1. Hagen, E. H. & Hammerstein, P. Game theory and human evolution: A critique of some recent interpretations of experimental games. *Theoretical Population Biology* 69, 339–348 (2006).
2. Zefferman, M. R. Direct reciprocity under uncertainty does not explain one-shot cooperation, but demonstrates the benefits of a norm psychology. *Evolution and Human Behavior* 35, 358–367 (2014).
3. Jagau, S. & van Veelen, M. A general evolutionary framework for the role of intuition and deliberation in cooperation. *Nature Human Behaviour* 1, 1–6 (2017).
4. Choi, J.-K. & Bowles, S. The coevolution of parochial altruism and war. *Science* 318, 636–640 (2007).
5. Richerson, P. et al. Cultural group selection plays an essential role in explaining human cooperation: A sketch of the evidence. *Behavioral and Brain Sciences* 39 (2016).
6. Boyd, R. & Richerson, P. J. Large-scale cooperation in small-scale foraging societies. *Evolutionary Anthropology: Issues, News, and Reviews* (2022).
7. De Dreu, C. K., Fariña, A., Gross, J. & Romano, A. Prosociality as a foundation for intergroup conflict. *Current Opinion in Psychology* 44, 112–116 (2022).
8. Akdeniz, A. & van Veelen, M. The cancellation effect at the group level. *Evolution* (2020).
9. Haselton, M. G., Nettle, D. & Murray, D. R. The evolution of cognitive bias. *The Handbook of Evolutionary Psychology* 1–20 (2015).
10. Balliet, D., Wu, J. & De Dreu, C. K. Ingroup favoritism in cooperation: a meta-analysis. *Psychological Bulletin* 140, 1556 (2014).
11. Yamagishi, T., Jin, N. & Kiyonari, T. Bounded generalized reciprocity: ingroup boasting and ingroup favoritism. *Advances in Group Processes* 16, 161–197 (1999).
12. Haley, K. J. & Fessler, D. M. T. Nobody's watching?: subtle cues affect generosity in an anonymous economic game. *Evolution and Human Behavior* 26, 245–256 (2005).
13. Yamagishi, T. & Mifune, N. Parochial altruism: does it explain modern human group psychology? *Current Opinion in Psychology* 7, 39–43 (2016).
14. Alger, I., Weibull, J. W. & Lehmann, L. Evolution of preferences in structured populations: genes, guns, and culture. *Journal of Economic Theory* 185, 104951 (2020).
15. Handley, C. & Mathew, S. Human large-scale cooperation as a product of competition between

- cultural groups. *Nature Communications* 11, 1–9 (2020).
16. Wilson, D. S. & Wilson, E. O. Rethinking the theoretical foundation of sociobiology. *The Quarterly Review of Biology* 82, 327–348 (2007).
 17. Gross, J. & De Dreu, C. K. The rise and fall of cooperation through reputation and group polarization. *Nature Communications* 10, 1–10 (2019).
 18. Henrich, J. Cultural group selection, coevolutionary processes and large-scale cooperation. *Journal of Economic Behavior and Organization* 53, 3–35 (2004).
 19. Muthukrishna, M. & Henrich, J. A problem in theory. *Nature Human Behaviour* 3, 221–229 (2019).
 20. Akdeniz, A. & van Veelen, M. The evolution of morality and the role of commitment. *Evolutionary Human Sciences* 3 (2021).
 21. Henrich, N. & Henrich, J. *Why Humans Cooperate: A Cultural and Evolutionary Explanation* (Oxford: Oxford University Press, 2007).
 22. Wahl, L. M. & Nowak, M. A. The continuous prisoner's dilemma: I. linear reactive strategies. *Journal of Theoretical Biology* 200, 307–321 (1999).
 23. Le, S. & Boyd, R. Evolutionary dynamics of the continuous iterated prisoner's dilemma. *Journal of Theoretical Biology* 245, 258–267 (2007).
 24. Gurven, M. To give and to give not: the behavioral ecology of human food transfers. *Behavioral and Brain Sciences* 27, 543–583 (2004).
 25. Kaplan, H. & Gurven, M. The natural history of human food sharing and cooperation: a review and a new multi-individual approach to the negotiation norms. In Gintis, H., Bowles, S., Boyd, R. & Fehr, E. (eds.) *Moral Sentiments and Material Interests: The Foundation of Cooperation in Economic Life*, 75–113 (Cambridge: The MIT Press, 2005).
 26. Hrdy, S. B. *Mothers and Others: The Evolutionary Origins of Mutual Understanding* (Cambridge, MA: Harvard University Press, 2011).
 27. Axelrod, R. M. *The Evolution of Cooperation* (New York: Basic Books, 1984).
 28. Hill, K. R. et al. Co-residence patterns in hunter-gatherer societies show unique human social structure. *Science* 331, 1286–1289 (2011).
 29. Boehm, C. *Blood Revenge: The Enactment and Management of Conflict in Montenegro and Other Tribal Societies* (Lawrence, KS: University Press of Kansas, 1984).
 30. Henrich, J. & Muthukrishna, M. The origins and psychology of human cooperation. *Annual Review of Psychology* 72, 207–240 (2021).
 31. Rusch, H. The evolutionary interplay of intergroup conflict and altruism in humans: a review

- of parochial altruism theory and prospects for its extension. *Proceedings of the Royal Society B: Biological Sciences* 281, 20141539 (2014).
32. Apicella, C. L., Marlowe, F. W., Fowler, J. H. & Christakis, N. A. Social networks and cooperation in hunter-gatherers. *Nature* 481, 497 (2012).
33. Lehmann, L. & Rousset, F. How life history and demography promote or inhibit the evolution of helping behaviours. *Philosophical Transactions of the Royal Society B: Biological Sciences* 365, 2599–2617 (2010).
34. Taylor, P. D. Altruism in viscous populations: an inclusive fitness model. *Evolutionary Ecology* 6, 352–356 (1992).
35. Kaplan, H., Lancaster, J. & Robson, A. Embodied capital and the evolutionary economics of the human life span. *Population and Development Review* 29, 152–182 (2003).
36. Fehr, E. & Schmidt, K. M. A theory of fairness, competition, and cooperation. *Quarterly Journal of Economics* 114, 817–868 (1999).
37. Bowles, S. *Microeconomics: Behavior, Institutions, and Evolution* (New York: Russell Sage, 2004).
38. Chudek, M. & Henrich, J. Culture–gene coevolution, norm-psychology and the emergence of human prosociality. *Trends in Cognitive Sciences* 15, 218–226 (2011).
39. Gibbons, R. S. *Game Theory for Applied Economists* (Princeton University Press, 1992).
40. Bell, A. V., Richerson, P. J. & McElreath, R. Culture rather than genes provides greater scope for the evolution of large-scale prosociality. *Proceedings of the National Academy of Sciences* 106, 17671–17674 (2009).
41. Brandts, J. & Charness, G. The strategy versus the direct-response method: a first survey of experimental comparisons. *Experimental Economics* 14, 375–398 (2011).
42. [A note. Not a citation.]
43. Boyd, R. & Lorberbaum, J. P. No pure strategy is evolutionarily stable in the repeated prisoner's dilemma game. *Nature* 327, 58–59 (1987).
44. van Veelen, M., García, J., Rand, D. G. & Nowak, M. A. Direct reciprocity in structured populations. *Proceedings of the National Academy of Sciences* 109, 9929–9934 (2012).
45. Boyd, R. Mistakes allow evolutionary stability in the repeated prisoner's dilemma game. *Journal of Theoretical Biology* 136, 47–56 (1989).
46. Price, G. R. Extension of covariance selection mathematics. *Annals of Human Genetics* 35, 485–490 (1972).

47. Gat, A. *War in Human Civilization* (Oxford: Oxford University Press, 2006).
48. Bowles, S. Did warfare among ancestral hunter-gatherers affect the evolution of human social behaviors? *Science* 324, 1293–1298 (2009).
49. Fry, D. P. & Söderberg, P. Lethal aggression in mobile forager bands and implications for the origins of war. *Science* 341, 270–273 (2013).
50. Clutton-Brock, T. Cooperation between non-kin in animal societies. *Nature* **462**, 51-57 (2009).
51. Romano, A., Sutter, M., Liu, J. H., Yamagishi, T. & Balliet, D. National parochialism is ubiquitous across 42 nations around the world. *Nature Communications* 12, 4456 (2021).
52. Axelrod, R. & Hamilton, W. D. The evolution of cooperation. *Science* 211, 1390–1396 (1981).

Reviewer Reports on the Second Revision:

Referees' comments:

Referee #1 (Remarks to the Author):

Super-additive cooperation – Revision 2

By Charles Efferson, Helen Bernhard, Urs Fischbacher, & Ernst Fehr

I am very positive about this paper, as I have been with earlier versions. Some minor differences in preferences remain, but in my previous report I said that I had only a few suggestions, so I will not reiterate things that I left up to the authors to use or not use. I like the paper, and I appreciate the responsiveness in general, and their constructive response to the essential suggestions from the first round in particular. A detail is that the paper was already long, and I have the impression that it may have become even longer in the process, but that is between the journal and the authors; I can live with all choices for how long it should be.

There is one small thing that I do attach some weight to. For the cancellation effect at the individual level, the authors cite Taylor (1992). The cancellation effect at the individual level, however, was discovered by Wilson, Pollock and Dugatkin (1992). Peter Taylor's paper is a response to theirs, redoing what they found in inclusive fitness terms.

Now the inclusive fitness crowd tends to cite only Taylor (1992), implicitly or explicitly suggesting that the cancellation effect was discovered by Peter Taylor. They have a hard time positively acknowledging anything that DS Wilson did, because they hate group selection, and DS Wilson is a big proponent of group selection. This resulted in the weird situation that the paper by Taylor has 674 citations, and the actual paper that discovered the cancellation effect has 457, and a whole bunch of people now think that it is Peter Taylor who discovered it. If the authors want to cite one paper only for the cancellation effect at the individual level, then my suggestion would be to cite the paper by the trio that actually discovered it, or to cite both papers, but I would suggest not to follow the bad tradition that the inclusive fitness crowd started.

Referee #2 (Remarks to the Author):

I have carefully read the Author's (excellent) responses to my last round of comments, and also re-read the paper. I believe this is a strong and clear paper that opens up deep new questions about the evolution of cooperation. I thank the authors for being so very responsive and for producing excellent work.

Referee #3 (Remarks to the Author):

The authors have seriously engaged with all of the reviewers' comments, adequately responded to the few minor concerns, and modified the writing to improve clarity. I support publication of the paper in Nature.

Sarah Mathew

Author Rebuttals to Second Revision:

Response to one remaining referee comment, which is from Referee #1

Referee #1 recommended that we cite Wilson et al. (1992) when discussing cancellation effects at the individual level. Indeed, we were unaware of this paper, and the referee is correct that Taylor (1992) has somehow become established, perhaps unfairly, as the standard reference. We now cite both papers.